# ZeroSiam: An Efficient Asymmetry for Test-Time Entropy Optimization without Collapse

**Guohao Chen**[1*], **Shuaicheng Niu**[12*], **Deyu Chen**[3], **Jiahao Yang**[3], **Zitian Zhang**[3]
**Mingkui Tan**[3], **Pengcheng Wu**[1], **Zhiqi Shen**[1†]
{guohao.chen, shuaicheng.niu, zqshen}@ntu.edu.sg
Nanyang Technological University[1]  Joint WeBank-NTU Research Institute on Fintech[2]
South China University of Technology[3]

## Abstract

Test-time entropy minimization helps adapt a model to novel environments and incentivize its reasoning capability, unleashing the model's potential during inference by allowing it to evolve and improve in real-time using its own predictions, achieving promising performance. However, pure entropy minimization can favor non-generalizable shortcuts, such as inflating the logit norm and driving all predictions to a dominant class to reduce entropy, risking collapsed solutions (*e.g.*, constant one-hot outputs) that trivially minimize the objective without meaningful learning. In this paper, we reveal asymmetry as a key mechanism for collapse prevention and introduce ZeroSiam–an efficient asymmetric Siamese architecture tailored for test-time entropy minimization. ZeroSiam prevents collapse through asymmetric divergence alignment, efficiently achieved by a learnable predictor and a stop-gradient operator before the classifier. We provide empirical and theoretical evidence that ZeroSiam not only prevents collapse, but also regularizes biased learning signals, enhancing performance even when no collapse occurs. Despite its simplicity, extensive results show that ZeroSiam performs more stably over prior methods using negligible overhead, demonstrating efficacy on both vision adaptation and large language model reasoning tasks across challenging test scenarios and diverse models, including particularly collapse-prone tiny models.

## 1 Introduction

Entropy measures the uncertainty of a model's predictions. Since its emergence, entropy optimization has been widely adopted as an auxiliary objective alongside supervised, reinforcement signals, *etc.* To be specific, in semi-supervised learning it enforces low-entropy predictions on unlabeled data to refine decision boundaries (Grandvalet & Bengio, 2004; Lee et al., 2013; Sajjadi et al., 2016); in domain adaptation it mitigates distribution shifts by promoting confident outputs (Jiang & Zhai, 2007; Long et al., 2016; Morerio et al., 2017); and in reinforcement learning it is often maximized to encourage exploration or minimized to ensure deterministic policies (Ziebart et al., 2008; Ahmed et al., 2019), among others, showing its remarkable success in boosting the learning effectiveness.

Of late, **test-time** entropy minimization has attracted growing interest, as it operates purely unsupervised during inference, without relying on any ground-truth supervision. For example, in test-time adaptation (TTA) (Wang et al., 2021; Lee et al., 2024) it encourages confident predictions to mitigate domain shifts on unseen data streams; in large language models, it helps calibrate uncertainty and improve prediction consistency for alignment (Hu et al., 2025a; Jang et al., 2025). These advances highlight the potential of entropy minimization in unleashing model power during inference, enabling adaptation to novel or out-of-distribution domains and moving toward greater intelligence.

However, entropy minimization naturally drives the model to increase the maximum predicted logit, regardless of whether it aligns with the true label. In practice, testing often involves noisy real-world data, domain shifts, or novel environments, where models often tend to be highly sensitive and uncertain, and the predicted maximum class is frequently incorrect. In this sense, minimizing

---

*Equal contribution. †Corresponding author. Source code at: https://github.com/Cascol-Chen/ZeroSiam.

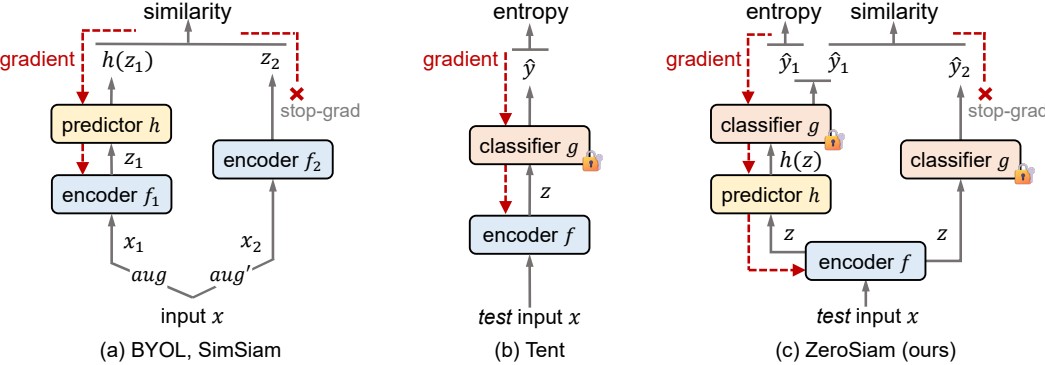

Figure 1: Comparisons on architectures. (a) Alignment-oriented SSL methods (BYOL (Grill et al., 2020), SimSiam (Chen & He, 2021)). (b) Test-time entropy minimization (Tent) (Wang et al., 2021). (c) Our ZeroSiam, which designs a minimal asymmetry for entropy minimization with a lightweight predictor and a stop-gradient branch—without augmentations, extra encoder passes, or teacher models—to substantially enhance learning stability and boost performance while retaining efficiency.

entropy alone does not ensure that the corresponding supervised loss (*e.g.*, cross-entropy) on the target task is also reduced. Instead, the model can easily exploit a shortcut solution by producing near one-hot outputs for all inputs, which trivially minimizes entropy but fails to capture meaningful predictions, leading to degraded performance. For instance, after TTA, models can deteriorate to predicting all samples to a single class under challenging wild testing scenarios (Niu et al., 2023).

Various TTA methods have been proposed to tackle the above issue, such as filtering unreliable gradients using predefined thresholds (Niu et al., 2022; Lee et al., 2024; Hu et al., 2025a), or mitigating disturbances through optimization of a sharpness-aware loss surface (Niu et al., 2023). However, the use of heuristic thresholds for gradient selection remains problematic, as such thresholds are difficult to define and generalize across domains. In this sense, filtering or suppressing noisy gradients can only be partial, and the model remains exposed to trivial solutions by optimizing with the remaining gradients, for which entropy can still be minimized by collapsing into constant one-hot outputs regardless of the inputs. As a result, it is unavoidable that these methods still risk collapse during deployment, particularly under prolonged or more challenging testing scenarios, as in Tables 2 & 6.

In this paper, we reveal asymmetry as a key mechanism for collapse prevention, which was introduced in the negative-free self-supervised learning (SSL) (Chen & He, 2021; Zhang et al., 2022a) that seeks to learn meaningful representations by maximizing the similarity between two augmented views of the same image. Negative-free SSL uses an *asymmetric structure* to prevent the outputs of *two branches* from collapsing toward the same constant, resolving collapse from architectural design. However, asymmetric structure is tailored for representation learning during pre-training, and naively applying it to our context is infeasible or inferior, as: 1) test-time entropy minimization typically has only one prediction branch and optimizes entropy instead of similarity; 2) traditional Siamese design requires extra backbone passes, which impairs efficiency for our test-time learning.

Therefore, we design ZeroSiam, demonstrating how asymmetry can be efficiently implemented in test-time entropy minimization without using augmentations, extra backbone passes, and teacher models. To embed asymmetry within a single backbone pass, ZeroSiam decouples a prediction into two asymmetric outputs based on the same feature: an online branch with a learnable predictor and a target branch with stop-gradient. As shown in Figure 1, we minimize entropy on the online branch to learn discriminative features, while performing asymmetric predictor–target alignment to prevent collapsed constants. We investigate ZeroSiam's behaviors empirically and theoretically during TTA in Section 3.2, demonstrating that our architecture avoids collapse and is also meaningful in regularizing biased shortcut learning signals at testing, which improves the performance of entropy optimization even when no collapse occurs. ZeroSiam achieves notable stability and effectiveness with a single extra predictor, introducing negligible overhead (see Table 1). Despite its simplicity, ZeroSiam yields superior robustness compared to prior methods across diverse architectures and a wide range of test scenarios, *e.g.*, adapting reliably even with *incorrect* pseudo labels (see Table 6).

**Main Novelty and Contributions**: 1) We are the first to study asymmetric structure in TTA and propose ZeroSiam, revealing how asymmetry can be efficiently implemented in test-time entropy min-

imization for collapse prevention without using augmentations, extra backbone passes, and teacher models. 2) We provide empirical and theoretical insights into ZeroSiam's behaviors (Section 3.2), demonstrating that ZeroSiam also helps absorb and regulate non-generalizable shortcut learning signals at testing, which enhances test-time learning performance even when no collapse occurs. 3) Extensive experiments on both language and vision tasks across both transformer and CNN models with varying sizes and a wide range of challenging test scenarios verify our efficacy.

## 2 PRELIMINARY AND PROBLEM STATEMENT

We briefly revisit test-time entropy minimization in this section for the convenience of our method presentation and put detailed related work discussions into Appendix A due to page limits.

**Test-Time Entropy Minimization** Formally, given any model with its encoder $f(\cdot; \theta_f)$ and classifier $g(\cdot; \theta_g)$ trained on source data $\mathcal{D}_{train}$, test-time entropy minimization conducts model learning directly from the testing data $\mathcal{D}_{test} = \{\mathbf{x}_j\}_{j=1}^M$ by optimizing

$$\min_{\tilde{\theta} \subset \{\theta_f, \theta_g\}} \mathcal{L}(\mathbf{x}; \theta_f, \theta_g) = -\sum_c p(\hat{y}_c) \log p(\hat{y}_c), \quad \text{where} \quad p(\hat{y}_c) = g(f(\mathbf{x}; \theta_f); \theta_g)[c]. \quad (1)$$

Here, $p(\hat{y}_c)$ denotes the predicted probability for class $c$, and $\tilde{\theta}$ represents the learnable parameters. Based on Eqn. (1), test-time adaptation methods (Wang et al., 2021; Zhang et al., 2022b; Marsden et al., 2024; Zhang et al., 2025a) often leverage it to adapt a pretrained model to novel domains under potential distribution shifts, *known as* out-of-distribution generalization. In natural language processing, it has also been employed for large language model alignment and improving predictive performance (Hu et al., 2025a; Jang et al., 2025). By using the model's own predictive entropy as a self-supervised signal during inference, Eqn. (1) requires neither source data nor modifications to the training process, making it a practical and lightweight solution for real-world deployment.

**Problem Statement and Motivation** As shown in Figure 2 (c-d), model learning with unsupervised entropy minimization on test data may favor non-generalizable shortcuts, such as (i) inflating the logit norm to reduce entropy, or (ii) aligning all logits towards a dominant mode. Such updates converge toward collapsed trivial solutions (*e.g.*, constant one-hot outputs), causing performance degradation. This phenomenon becomes particularly severe in challenging test scenarios or when using weaker base models (*e.g.*, ConvNeXt-Tiny), where the lower source accuracy makes the resulting entropy gradients substantially less reliable, thereby increasing the risk of collapse. Existing methods mainly seek to improve TTA stability with heuristic thresholds to remove some unreliable updates (Niu et al., 2023; Lee et al., 2024), but *the objective is yet trivially minimized by collapsed solutions*. Consequently, prior methods remain exposed to collapse and are sensitive to architectures and domains (see Tables 3-6). This gap motivates us to ask: *How can we design an efficient, theoretically grounded entropy optimization mechanism that inherently avoids undesired trivial solutions?*

## 3 ZEROSIAM: STABLE ADAPTATION VIA MINIMAL ASYMMETRIC NETWORK

We achieve the goal of efficient test-time entropy optimization without collapse by designing a lightweight asymmetric Siamese architecture for entropy minimization, namely ZeroSiam. We draw inspiration from the traditional asymmetric design for similarity learning in SSL, but extend its compatibility to the single-branch entropy minimization via a single learnable predictor—without requiring augmentations and extra backbone passes. We depict the design choice of our ZeroSiam in Section 3.1, and provide further theoretical and empirical evidence of ZeroSiam's stability for TTA in Section 3.2. The overall details of ZeroSiam are illustrated in Figure 1 (c) and Algorithm 1.

### 3.1 MINIMAL ASYMMETRIC SIAMESE ARCHITECTURE FOR ENTROPY MINIMIZATION

Unlike cross-entropy training supervised by ground-truth labels, test-time entropy minimization relies on the model's own predicted labels for self-training. This, however, creates a shortcut solution, where the objective can be minimized simply by producing one-hot outputs irrespective of the input. As shown in Figure 2 (c-d), pure entropy minimization tends to inflate the logit norm and drive all predictions toward a dominant class to reduce the entropy naively, leading to model collapse into degenerate solutions (*i.e.*, constant one-hot outputs). Similar issues appear in negative-free SSL, where

---

**Algorithm 1** *ZeroSiam*: Test-Time Asymmetric Entropy Minimization.

---

**Input:** Test samples $\mathcal{D}_{test} = \{\mathbf{x}_j\}_{j=1}^M$, encoder $f(\cdot; \theta_f)$, classifier $g(\cdot; \theta_g)$, inserted predictor $h(\cdot; \theta_h)$
**Output:** Predictions $\{p_j^r\}_{j=1}^M$.
Initialize predictor $h(\cdot; \theta_h) = $ Identity ;          // warm start for fully test-time learning
**for** $\mathbf{x}_j \in \mathcal{D}_{test}$ **do**
    Compute encoder feature: $z = f(\mathbf{x}_j; \theta_f)$ ;          // augmentation-free, single-pass
    Compute target branch outputs $u^r = g(z; \theta_g), p^r = \text{softmax}(u^r)$ ;          // stop-gradient
    Compute online branch outputs $u^o = g(h(z; \theta_h); \theta_g), p^o = \text{softmax}(u^o)$ ;
    Update $\theta_h$ and normalization params $\tilde{\theta} \subset \theta_f$ via Eqn. (4).
**end**

---

the model learns trivial solutions via constant representations regardless of inputs to maximize representation similarity. SSL (Chen & He, 2021) prevents such collapse by introducing an asymmetry: with a non-identity predictor $h$ on one branch, and a stop-gradient operation on the other (see Figure 1). This ensures that degenerate constant outputs incur non-zero alignment loss, preventing the output of *two branches* from collapsing toward the same constant, showing a promising solution.

Motivated by this, we hypothesize that an asymmetric mechanism, which remains under-explored in entropy optimization, could also prevent undesired trivial solutions and stabilize self-training. However, constructing asymmetry in test-time entropy optimization remains an open question: (1) Unlike contrastive SSL, entropy-based learning relies on *one prediction branch*, making it non-trivial to introduce a Siamese asymmetric structure; (2) Its objective optimizes entropy rather than similarity, offering no pairwise stop-gradient alignment that enables asymmetry as in SSL; (3) Traditional Siamese designs require extra backbone passes, hindering the efficiency for our test-time learning.

To bridge these gaps, our key idea is to embed asymmetry within a single forward pass. We achieve this by inserting a predictor and the stop-gradient operator before the classifier, which decouples a prediction into two *asymmetric* views from the same feature: an online branch (through the predictor) for *optimizing entropy*, and a target branch (the original logits) for *asymmetric stop-gradient alignment*. In this way, it establishes an efficient asymmetric Siamese for entropy optimization, using no augmentations or extra backbone passes, yet still exploiting asymmetric predictor–target alignment to prevent collapsed constants and stabilize adaptation. Formally, let $f(\cdot; \theta_f)$ denote the encoder, $g(\cdot; \theta_g)$ the classifier, and $h(\cdot; \theta_h)$ a lightweight *linear* predictor. Given a test sample $\mathbf{x}$, ZeroSiam computes the encoder feature $z = f(\mathbf{x}; \theta_f)$ once, then define two asymmetric branches:

$$u^r = g(z; \theta_g), \qquad\qquad \text{(target branch, stop-gradient)} \qquad (2)$$
$$u^o = g(h(z; \theta_h); \theta_g), \qquad\qquad \text{(online branch).} \qquad\qquad (3)$$

Let $p^r = \text{softmax}(u^r)$ and $p^o = \text{softmax}(u^o)$. The ZeroSiam objective combines entropy minimization on the online branch with an alignment regularizer to the target branch:

$$\mathcal{L} = H(p^o) + \alpha\, D(p^o \,\|\, \text{sg}[p^r]), \qquad (4)$$

where $H(p) = -\sum_c p_c \log p_c$ is the prediction entropy, $D(\cdot\|\cdot)$ is a divergence (*e.g.*, $\ell_2$ or KL)[1], and $\text{sg}[\cdot]$ denotes stop-gradient. Here, $\theta_h$ is initialized as an identity to ensure a warm start, which quickly diverges during online learning, as shown in Figure 2 (a), creating the asymmetry necessary to prevent collapsed constant solutions. Even when using a *randomly initialized* predictor to introduce asymmetry, Table 8 shows that ZeroSiam helps prevent collapse in Tent (Wang et al., 2021), revealing asymmetry's value. In essence, the asymmetric predictor–stop-gradient alignment inherently rules out collapsed constant solutions as valid minima, thereby avoiding collapse and improving stability during online entropy minimization. Unlike BYOL and SimSiam, which rely on augmentations or extra backbone passes, ZeroSiam shows that asymmetry can also be instantiated within pure online entropy minimization in a minimal and efficient manner.

## 3.2 HOW ZEROSIAM RESISTS COLLAPSE: EMPIRICAL AND THEORETICAL INSIGHTS

Although asymmetric designs (*e.g.*, SimSiam for SSL) are known to prevent collapsed trivial solutions, test-time entropy minimization poses new challenges like logit norm inflation and single-class

---

[1]ZeroSiam uses symmetric KL loss as $D(\cdot\|\cdot)$ to penalize both under- and over-coverage of modes, *i.e.*, $D(p\|q) = D_{\text{KL}}(p\|q) + D_{\text{KL}}(q\|p)$, and $\alpha$ is fixed to 1. Results with KL and reverse KL are put in Table 10.

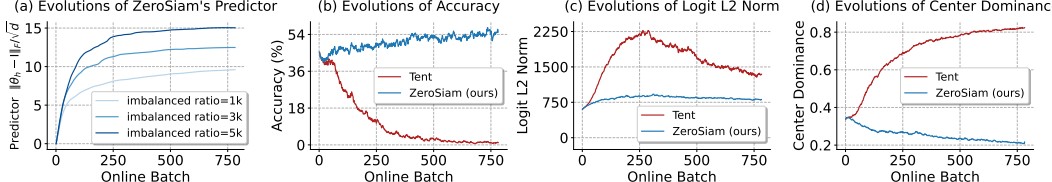

Figure 2: Empirical evidence of ZeroSiam's stabilization effects. (a) records the Frobenius distance between $\theta_h$ and the identity matrix under the non-i.i.d. streams with varying imbalance ratios (Niu et al., 2023). (b-d) record the ODD accuracy, logits $L_2$ norm, and center dominance in model predictions under a mild test scenario (Wang et al., 2021). Center dominance is calculated by $\|\overline{u}\|/\|u\|$ following (Zhang et al., 2022a). Experiments are conducted on ImageNet-C (Snow, level 5) with ResNet50-GN. For fair comparisons, ZeroSiam and Tent use the same learning rate configuration.

dominance. In this section, we rethink TTA stability from shortcuts, and find that ZeroSiam not only inherits the anti-collapse effect of asymmetry, but also *plays a broader role* in absorbing biased shortcuts and regularizing dynamics during entropy minimization. We present empirical evidence showing these stabilizing effects, and further explain ZeroSiam's behaviors with theoretical insights.

**Rethinking TTA Stability from Shortcuts** During unsupervised TTA, the model can exploit shortcuts (*e.g.*, assigning all predictions to a dominant class) to naively reduce the objective, resulting in model collapse, as shown in Figure 2. Therefore, to improve TTA stability, we seek to identify what shortcuts are and suppress the exploitation of shortcuts during unsupervised learning. Instead of identifying shortcuts specifically for each objective, as in Figures 2 & B, our key insight is that models with different sizes and architectures exploit similar shortcut patterns under the same objective. In this sense, we *define shortcuts in an objective as patterns learnable by an extremely simple network (*e.g., the predictor $h$*), which naively reduce the loss value without improving generalization.

**Empirical Evidence of ZeroSiam's Stabilization Effects** In Figure 2, we visualize the effects of our ZeroSiam under the unsupervised TTA. We conduct experiments with ResNet50-GN on ImageNet-C, and record the evolution of OOD accuracy, representative failure modes of entropy-based TTA, and changes in predictor parameters $\theta_h$ (*i.e.*, a simple FC layer, learning preliminarily non-generalizable modes due to capacity). From the results, we highlight two key observations:

*Observation 1: Predictor as an Effective Absorber of Biased Shortcut Signals.* Figure 2 (a) shows that the predictor parameters $\theta_h$ deviate more rapidly and substantially from the identity mapping when facing online data streams with a higher label imbalance ratio (Niu et al., 2023), which produces more shortcut signals for entropy minimization that may result in collapse. In this sense, this pronounced deviation thus suggests an active absorption of biased shortcut signals within predictor $h$ (an extremely simple network), adaptively posing stronger alignment regularization in Eqn. (4) to stabilize entropy adaptation under more collapse-prone TTA scenarios, as shown in Figure C.

*Observation 2: Asymmetry Suppresses Learning Non-Generalizable Collapse Modes.* In Figure 2 (c), the logit $L_2$ norm under Tent suffers from a rapid inflation, whereas ZeroSiam grows slowly and then stabilizes. In Figure 2 (d), Tent increasingly aligns logits toward a dominant mode (higher center dominance), while our ZeroSiam suppresses such dominance over time. Consequently, Tent collapses into non-generalizable shortcuts, whereas ZeroSiam mitigates them and achieves better generalization as shown in Figure 2 (b), similarly even when no collapse occurs (see Figure B).

Overall, Figure 2 shows that ZeroSiam absorbs and regulates non-generalizable shortcuts during TTA beyond collapse prevention. This reveals a distinct advantage of ZeroSiam, and provides TTA-specific evidence on the role of asymmetry in generalization that extends prior findings from SSL.

**Theoretical Insights of ZeroSiam's Mechanism** We provide a theoretical analysis to explain ZeroSiam's behaviors, showing that (i) the predictor enhances the online branch's dynamism in exploring the parameter space, thereby facilitating more efficient entropy optimization; (ii) the alignment term explicitly regulates the biased learning signals induced by entropy minimization, preventing the model from being exposed to collapsed trivial solutions and meanwhile prompting performance.

**Theorem 1.** *(Optimization and Stability of ZeroSiam) Consider the ZeroSiam objective $\mathcal{L} = H(p^o) + \alpha\, D(p^o \,\|\, \mathrm{sg}[p^r])$, where $H(\cdot)$ denotes the entropy loss, $D(\cdot)$ the alignment regularizer, and $p^o, p^r \in \Delta^{|\mathcal{C}|-1}$ are the probability distributions induced by the online and target branches.*

*Given the encoder $f$, classifier $g$ and predictor $h$. Under Assumptions 1 and 2, the following hold:*
*(1) For $\alpha = 0$, the entropy variation satisfies $|\Delta H(p^o)| > |\Delta H(p^r)|$, and the Hessian of $H(p^o)$ attains its minimal eigenvalue along collapse directions $\mathbf{v}$:*

$$\lambda_{\min}\big(\nabla^2 H(p^o)\big) = \mathbf{v}^\top \nabla^2 H(p^o)\mathbf{v}.$$

*(2) For $\alpha > 0$, the predictor $h$ serves as a filtering mechanism that suppresses gradient update directions corresponding to over-amplified logits, and the system converges to a stable equilibrium: there exists $h_{\min} > 0$ such that*

$$H(p^o) > h_{\min}, \ p^o \to p^r.$$

**Remark 1.** *Theorem 1 characterizes ZeroSiam's optimization dynamics and convergence behavior. Initially, the online and target branches coincide within the non-collapse region, so entropy minimization dominates. In this regime, the online branch exhibits a stronger tendency toward collapse than the target branch in both magnitude and directional dominance (**Conclusion (1)**). As optimization proceeds, the alignment term $D(\cdot)$ constrains the predictor $h$ to filter/suppress the gradient components that drive $p^o$ away from $p^r$, i.e., regulating the biased learning signals that may result in collapse, guiding the system toward a stable and non-collapsing equilibrium (**Conclusion (2)**). This show that the predictor in the online branch purposely converts biased signals (e.g., shortcuts) into explicit discrepancies, where these biased signals are then penalized by the alignment loss.*

## 4 COMPARISONS WITH STATE-OF-THE-ARTS

**Dataset and Methods** For image classification, we conduct experiments based on ImageNet-C (Hendrycks & Dietterich, 2019), a large-scaled benchmark for out-of-distribution generalization. It contains 15 types of 4 main categories (noise, blur, weather, digital) corrupted images and each type has 5 severity levels. For natural language reasoning, we assess logical reasoning capabilities on Math-500 (Lightman et al., 2023), CollegeMath (Tang et al., 2024), AIME24, and Minerva (Lewkowycz et al., 2022), ranging from high-school mathematics to advanced competition problems. We compare with the following methods. EATA (Niu et al., 2022), SAR (Niu et al., 2023), DeYO (Lee et al., 2024), and CETA (Yang et al., 2024) are entropy-based methods with sample selection or update reweighting. COME (Zhang et al., 2025a) predefines an uncertainty mass, implemented based on DeYO for comparisons. TLM (Hu et al., 2025a) refines the prediction entropy on input sequences.

Table 1: Efficiency comparison for processing 50,000 images (Gaussian noise, level 5 on ImageNet-C) via an RTX 3090 GPU on ViT-Base.

| Method | GPU time |
|---|---|
| Tent (Wang et al., 2021) | 193 secs |
| SAR (Niu et al., 2023) | 382 secs |
| EATA (Niu et al., 2022) | 197 secs |
| COME (Zhang et al., 2025a) | 300 secs |
| DeYO (Lee et al., 2024) | 280 secs |
| ZeroSiam (ours) | 193 secs |

**Models and Implementation Details** We conduct experiments on ResNet50-GN, ViT-Base, ViT-Small, ConvNeXt-Tiny, and Swin-Tiny that are obtained from `timm` (Wightman, 2019) for image classification, and Llama3.1-8B-Instruct (Dubey et al., 2024) for natural language reasoning. We use symmetric KL as the divergence term in Eqn. (4), and $\alpha$ is fixed to 1 without tuning. We update predictor params $\theta_h$, and the affine parameters in norm layers for the vision task per Tent (Wang et al., 2021), and the LoRA (Hu et al., 2022) parameters for LLM. More details are put in Appendix C.

### 4.1 ROBUSTNESS TO CORRUPTION UNDER VARIOUS WILD TEST SETTINGS

In this section, we evaluate the robustness of ZeroSiam under the wild test scenarios as established by SAR (Niu et al., 2023), which simulate practical TTA scenarios in real-world applications, including 1) **online imbalanced label distribution shifts**: the samples in a non-i.i.d. data stream come in class order (**in Table 3**), 2) **mixed distribution shifts**: the samples in online data stream are obtained from a mixture of 15 corruption types (a total of $15 \times 50,000$ images) in ImageNet-C, i.e., a prolonged TTA scenario with multiple concurrent shifts (**in Table 2**), and 3) **batch size = 1**: the samples come one-by-one due to the scarcity of data in online streams (**in Table 4**). Compared to SAR, we extend the evaluations using more models, including ViT-Small, ConvNeXt-Tiny, and Swin-Tiny.

From the results, we have the following general observations: 1) *Sensitivity on model choice*: While sample selection methods like DeYO perform competitively on ResNet50-GN and ViT-Base, they remain unstable and *sensitive to model choices*, particularly with weaker models where pseudo labels are highly unreliable, e.g., the accuracy of 0.1% (DeYO) vs. 26.5% (ZeroSiam) on

Table 3: Comparisons with SOTAs on ImageNet-C (level 5) under **ONLINE LABEL SHIFTS** (imbalance ratio=∞) w.r.t. **Acc(%)**. "$\mathcal{N}/\mathcal{B}/\mathcal{W}/\mathcal{D}$" are short for *Noise/Blur/Weather/Digital* corruptions. RED marks results worse than NoAdapt. **Detailed results of each corruption are put in Appendix.**

| Method | ResNet50-GN | | | | ViT-Base | | | | ViT-Small | | | | ConvNeXt-Tiny | | | | Swin-Tiny | | | | Avg. |
| | $\mathcal{N}$ | $\mathcal{B}$ | $\mathcal{W}$ | $\mathcal{D}$ | $\mathcal{N}$ | $\mathcal{B}$ | $\mathcal{W}$ | $\mathcal{D}$ | $\mathcal{N}$ | $\mathcal{B}$ | $\mathcal{W}$ | $\mathcal{D}$ | $\mathcal{N}$ | $\mathcal{B}$ | $\mathcal{W}$ | $\mathcal{D}$ | $\mathcal{N}$ | $\mathcal{B}$ | $\mathcal{W}$ | $\mathcal{D}$ | |
|---|---|---|---|---|---|---|---|---|---|---|---|---|---|---|---|---|---|---|---|---|---|
| NoAdapt | 18.6 | 19.4 | 47.7 | 33.9 | 8.1 | 28.4 | 36.1 | 41.6 | 1.8 | 23.4 | 27.7 | 33.5 | 23.7 | 24.8 | 52.1 | 35.4 | 25.8 | 21.7 | 50.5 | 25.9 | 29.0 |
| Tent | 2.9 | 14.6 | 27.7 | 38.0 | 22.9 | 54.3 | 41.3 | 64.7 | 0.3 | 40.5 | 38.5 | 48.2 | 22.8 | 22.9 | 50.2 | 34.4 | 13.6 | 19.4 | 34.9 | 25.3 | 30.9 |
| SAR | 35.0 | 24.9 | 47.8 | 40.6 | 46.2 | 55.8 | 61.4 | 65.8 | 1.6 | 42.2 | 46.7 | 52.8 | 35.5 | 24.2 | 43.3 | 38.0 | 29.2 | 19.0 | 39.4 | 26.7 | 38.8 |
| EATA | 27.8 | 20.4 | 42.7 | 34.6 | 35.7 | 47.4 | 61.6 | 51.6 | 16.3 | 42.2 | 54.8 | 54.2 | 35.4 | 29.0 | 54.9 | 42.2 | 35.6 | 32.9 | 53.4 | 41.1 | 40.7 |
| COME | 18.4 | 19.2 | 47.7 | 33.4 | 49.8 | 59.2 | 69.8 | 67.5 | 0.2 | 39.5 | 56.0 | 59.6 | 43.1 | 24.4 | 62.9 | 44.9 | 42.9 | 23.9 | 59.7 | 27.7 | 42.5 |
| DeYO | 43.7 | 23.2 | 54.2 | 54.5 | 48.0 | 56.6 | 71.2 | 69.9 | 0.1 | 41.4 | 59.4 | 60.7 | 36.0 | 19.0 | 43.8 | 34.8 | 36.5 | 22.7 | 52.1 | 34.7 | 43.1 |
| *ZeroSiam* | 43.7 | 41.2 | 62.1 | 57.3 | 52.8 | 60.1 | 71.2 | 69.6 | 26.5 | 50.4 | 62.0 | 60.8 | 43.0 | 40.0 | 62.4 | 53.8 | 45.3 | 42.2 | 59.3 | 53.6 | **52.9** |

Table 4: Comparisons with SOTAs on ImageNet-C (level 5) with **BATCH SIZE=1** w.r.t. **Acc (%)**.

| Method | ResNet50-GN | | | | ViT-Base | | | | ViT-Small | | | | ConvNeXt-Tiny | | | | Swin-Tiny | | | | Avg. |
| | $\mathcal{N}$ | $\mathcal{B}$ | $\mathcal{W}$ | $\mathcal{D}$ | $\mathcal{N}$ | $\mathcal{B}$ | $\mathcal{W}$ | $\mathcal{D}$ | $\mathcal{N}$ | $\mathcal{B}$ | $\mathcal{W}$ | $\mathcal{D}$ | $\mathcal{N}$ | $\mathcal{B}$ | $\mathcal{W}$ | $\mathcal{D}$ | $\mathcal{N}$ | $\mathcal{B}$ | $\mathcal{W}$ | $\mathcal{D}$ | |
|---|---|---|---|---|---|---|---|---|---|---|---|---|---|---|---|---|---|---|---|---|---|
| NoAdapt | 18.6 | 19.4 | 47.7 | 33.9 | 8.1 | 28.4 | 36.1 | 41.7 | 1.8 | 23.3 | 27.8 | 33.6 | 23.7 | 24.6 | 52.2 | 35.3 | 25.9 | 21.6 | 50.6 | 25.8 | 29.0 |
| Tent | 2.6 | 13.3 | 27.5 | 37.9 | 28.8 | 51.7 | 43.7 | 61.9 | 23.7 | 24.6 | 52.2 | 35.3 | 32.0 | 23.9 | 51.9 | 36.5 | 24.2 | 23.0 | 42.1 | 26.7 | 33.2 |
| SAR | 24.3 | 23.2 | 46.9 | 40.6 | 39.6 | 53.7 | 63.1 | 64.6 | 25.9 | 21.6 | 50.6 | 25.8 | 31.2 | 25.3 | 51.0 | 35.8 | 25.2 | 22.8 | 47.0 | 26.3 | 37.2 |
| EATA | 26.3 | 23.3 | 50.3 | 43.5 | 29.8 | 43.7 | 52.3 | 56.4 | 2.2 | 30.7 | 36.9 | 43.6 | 28.0 | 25.1 | 51.8 | 36.4 | 33.0 | 25.3 | 50.6 | 30.8 | 36.0 |
| COME | 18.4 | 19.3 | 47.6 | 33.5 | 51.7 | 55.1 | 70.6 | 69.6 | 1.5 | 41.0 | 56.7 | 59.6 | 41.8 | 25.5 | 62.0 | 43.5 | 39.7 | 25.3 | 58.3 | 38.6 | 43.0 |
| DeYO | 43.2 | 28.4 | 50.2 | 55.5 | 53.7 | 59.0 | 71.7 | 70.4 | 0.5 | 42.4 | 59.1 | 60.4 | 37.7 | 21.3 | 44.2 | 35.3 | 36.4 | 24.6 | 50.1 | 35.3 | 44.0 |
| *ZeroSiam* | 42.6 | 40.8 | 62.9 | 58.6 | 52.5 | 59.9 | 70.9 | 69.6 | 27.1 | 50.1 | 61.4 | 60.5 | 42.2 | 39.3 | 61.7 | 52.8 | 44.9 | 41.5 | 58.8 | 52.9 | **52.5** |

*noise* corruptions with ViT-Small under the label shift scenario in Table 3, and SAR and COME also fail to perform TTA (worse than NoAdapt) on Swin-Tiny under mixed shifts in Table 2; 2) *Sensitivity on test scenario choice:* For example, compared to results in Table 3 under label shifts, EATA's performance drops significantly on Swin-Tiny under TTA with a single sample in Table 4, *e.g.*, 32.9%→25.3% regarding the accuracy on *blur* corruptions. This highlights the instability in prior methods and that

Table 2: Comparisons with state-of-the-arts on ImageNet-C (level 5) under **MIXTURE OF 15 CORRUPTION TYPES** w.r.t. **Acc (%)**.

| Method | R-50 | Vit-B | Vit-S | Con-T | Swi-T | Avg. |
|---|---|---|---|---|---|---|
| NoAdapt | 30.6 | 29.9 | 22.9 | 34.7 | 31.3 | 29.9 |
| Tent | 13.4 | 16.5 | 7.1 | 27.8 | 6.7 | 14.3 |
| SAR | 38.1 | 55.7 | 41.3 | 36.5 | 25.9 | 39.5 |
| EATA | 38.3 | 57.1 | 36.9 | 41.1 | 41.3 | 42.9 |
| COME | 30.0 | 58.5 | 40.9 | 23.7 | 19.1 | 34.4 |
| DeYO | 38.6 | 59.4 | 35.9 | 27.8 | 29.0 | 38.1 |
| *ZeroSiam* | 40.4 | 57.1 | 41.7 | 42.4 | 39.3 | **44.2** |

achieving *robust TTA across diverse architectures and test scenarios remains highly non-trivial*; 3) Our ZeroSiam is simple yet effective. ZeroSiam neither relies on careful sample selection strategies as in SAR or DeYO, nor requires source data as in EATA, and it achieves consistently stable and high performance across models and scenarios, markedly outperforming prior methods, *e.g.*, 52.9% (ZeroSiam) *vs.* 38.8% (SAR) in terms of average accuracy under label shifts in Table 3, and an average gain of +8.5% over DeYO under the batch size of 1 in Table 4, suggesting our effectiveness.

## 4.2 EFFECTIVENESS FOR INCENTIVIZING REASONING CAPABILITY ONLINE

A statistical reasoning model may fail to generalize to the diverse reasoning tasks in real-world applications, while retraining the model offline is both expensive and time-consuming. We further explore adaptively incentivizing the reasoning capability in a large language model *online*, by minimizing the entropy of its own predicted tokens. From Table 5, we have the following observations: 1) Entropy optimization demonstrates a great potential in enhancing the model's reasoning online, *i.e.*, all prior methods achieve an accuracy gain by +3.34% on AIME24. 2) However, for a comprehensive benchmark like CollegeMath, covering algebra, calculus, probability, differential equations, *etc*, existing methods may suffer from overfitting and poor generalization, *e.g.*, -0.83% on Tent and -0.75% on SAR. 3) ZeroSiam enhances generalization for online reasoning incentivization with its predictor and asymmetric alignment design, which regularizes non-generalizable shortcuts as discussed in Section 3.2, yielding significant further gains, *e.g.*, +10.00% on AIME24, +3.40% on Math-500, and +3.94% on average. This suggests the potential of our ZeroSiam to boost both the perception robustness and reasoning capability in a model during inference for greater intelligence.

## 4.3 ROBUSTNESS TO ADVERSARIAL ADAPTATION SCENARIOS (STRESS TEST)

**Robustness under Blind-Spot Adaptation** Existing entropy-based TTA methods largely rely on reliable sample selection strategies to stabilize adaptation (Niu et al., 2023; Lee et al., 2024). How-

Table 5: Comparisons with state-of-the-arts for **ONLINE ADAPTIVE REASONING** in mathematical reasoning benchmarks regarding **Accuracy (%)**.

| Model+Method | Math-500 | CollegeMath | AIME24 | Minerva | Average |
|---|---|---|---|---|---|
| Llama3.1-8B (Dubey et al., 2024) | 49.20 | 25.00 | 3.33 | 20.96 | 24.62 |
| • Tent (Wang et al., 2021) | $50.00_{(+0.80)}$ | $24.17_{(-0.83)}$ | $6.67_{(+3.34)}$ | $20.59_{(-0.37)}$ | $25.36_{(+0.74)}$ |
| • SAR (Niu et al., 2023) | $49.20_{(+0.00)}$ | $24.25_{(-0.75)}$ | $6.67_{(+3.34)}$ | $21.32_{(+0.36)}$ | $25.36_{(+0.74)}$ |
| • EATA (Niu et al., 2022) | $48.80_{(-0.40)}$ | $24.83_{(-0.17)}$ | $6.67_{(+3.34)}$ | $20.59_{(-0.37)}$ | $25.22_{(+0.60)}$ |
| • COME (Zhang et al., 2025a) | $49.80_{(+0.60)}$ | $25.42_{(+0.42)}$ | $6.67_{(+3.34)}$ | $22.74_{(+1.78)}$ | $26.16_{(+1.54)}$ |
| • TLM (Hu et al., 2025a) | $50.00_{(+0.80)}$ | $25.58_{(+0.58)}$ | $6.67_{(+3.34)}$ | $19.49_{(-1.47)}$ | $25.44_{(+0.82)}$ |
| • *ZeroSiam* (ours) | $52.60_{(+3.40)}$ | $26.25_{(+1.25)}$ | $13.33_{(+10.00)}$ | $22.06_{(+1.10)}$ | $\mathbf{28.56}_{(+3.94)}$ |

Table 6: Comparisons with SOTAs on ImageNet-C (level 5)'s **BLIND-SPOT SUBSET** (samples initially misclassified by the NoAdapt model) with **BATCH SIZE=1** regarding **Accuracy (%)**.

| | ResNet50-GN | | | | ViT-Base | | | | ViT-Small | | | | ConvNeXt-Tiny | | | | Swin-Tiny | | | | |
|---|---|---|---|---|---|---|---|---|---|---|---|---|---|---|---|---|---|---|---|---|---|
| Method | $\mathcal{N}$ | $\mathcal{B}$ | $\mathcal{W}$ | $\mathcal{D}$ | $\mathcal{N}$ | $\mathcal{B}$ | $\mathcal{W}$ | $\mathcal{D}$ | $\mathcal{N}$ | $\mathcal{B}$ | $\mathcal{W}$ | $\mathcal{D}$ | $\mathcal{N}$ | $\mathcal{B}$ | $\mathcal{W}$ | $\mathcal{D}$ | $\mathcal{N}$ | $\mathcal{B}$ | $\mathcal{W}$ | $\mathcal{D}$ | Avg. |
| NoAdapt | 18.6 | 19.4 | 47.7 | 33.9 | 8.1 | 28.4 | 36.1 | 41.7 | 1.8 | 23.3 | 27.8 | 33.6 | 23.7 | 24.6 | 52.2 | 35.3 | 25.9 | 21.6 | 50.6 | 25.8 | 29.0 |
| Tent | 0.2 | 5.2 | 16.8 | 28.1 | 0.2 | 42.9 | 20.0 | 52.0 | 0.1 | 40.0 | 36.3 | 42.8 | 17.9 | 19.7 | 36.3 | 31.0 | 0.2 | 8.0 | 22.2 | 13.0 | 21.7 |
| SAR | 17.9 | 18.4 | 35.6 | 31.6 | 31.1 | 55.0 | 54.0 | 56.4 | 2.0 | 30.4 | 42.0 | 43.3 | 26.4 | 24.6 | 50.7 | 35.1 | 27.0 | 22.2 | 49.3 | 23.8 | 33.8 |
| EATA | 19.5 | 19.0 | 44.2 | 40.2 | 26.3 | 43.5 | 53.2 | 57.5 | 2.0 | 26.5 | 32.8 | 41.7 | 25.8 | 24.7 | 51.6 | 35.7 | 27.4 | 22.4 | 49.1 | 26.8 | 33.5 |
| COME | 18.6 | 19.4 | 47.6 | 33.8 | 0.1 | 44.8 | 70.9 | 69.7 | 1.0 | 37.0 | 55.9 | 46.2 | 41.4 | 16.4 | 31.1 | 36.1 | 40.5 | 9.5 | 25.3 | 26.9 | 33.6 |
| DeYO | 0.5 | 9.4 | 21.0 | 40.0 | 0.2 | 47.2 | 72.6 | 71.4 | 0.2 | 38.2 | 60.2 | 61.7 | 35.2 | 12.2 | 35.0 | 33.0 | 0.2 | 8.8 | 25.7 | 23.9 | 29.8 |
| *ZeroSiam* | 39.9 | 36.6 | 53.9 | 52.7 | 53.0 | 60.4 | 71.6 | 70.2 | 26.1 | 49.8 | 61.8 | 61.1 | 43.3 | 40.3 | 61.9 | 55.0 | 46.5 | 42.1 | 58.8 | 54.7 | **52.0** |

ever, in many real-world scenarios, adaptation inevitably involves numerous erroneous pseudo labels that cannot be identified, especially when the model's initial accuracy is very low. This raises a critical concern: How safe is existing TTA in the real world? To stress-test this vulnerability, we construct a *blind-spot subset* consisting only of samples misclassified by the NoAdapt model for each domain and adapt exclusively on this subset, before evaluation on the full dataset. As shown in Table 6, prior methods suffer from severe instability, often collapse and underperform the NoAdapt baseline after TTA, *e.g.*, DeYO deteriorates accuracy on *12 out of 20 scenarios*. In contrast, ZeroSiam introduces an efficient asymmetry to prevent undesired trivial solutions from optimization, delivering consistent accuracy gains *even when adapting with incorrect labels e.g.*, the average accuracy of 29.0% (NoAdapt) *vs.* 52.0% (ZeroSiam), drastically breaking prior performance ceiling ($\approx$33.6%). This underscores ZeroSiam as a more principled and robust solution for real-world TTA.

**Resistance to Learning from Noise** In dynamic real-world applications, models may frequently encounter test data that are severely corrupted and *non-semantic*, such as extreme occluded frames and pure sensor noise where no valid label exists. Minimizing entropy on these data can then be misled to "confidently learn nonsense", risking model degeneration. To simulate this, we first pre-adapt models on varying amounts of pure Gaussian noise, then perform TTA on ImageNet-C (Fog, level 5) with ViT-Base under the mid scenario (Wang et al., 2021). As shown in Figure 3, existing entropy-based methods degrade sharply post-adaptation to noise, *e.g.*, 67.3%→27.2% on CETA, indicating an overfit to non-semantic patterns. In contrast, ZeroSiam absorbs

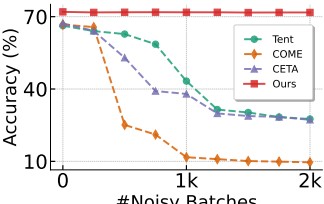

Figure 3: Resistance to learning from noise. Models pre-adapt on $N$ pure Gaussian noise, then run TTA on ImageNet-C (level 5).

and regularizes non-generalizable shortcuts through the asymmetric alignment (c.f. Section 3.2), maintaining consistently high and stable accuracy ($\approx 72\%$) regardless of the exposure to noisy inputs. This robustness implies that ZeroSiam can be safely deployed even when streams contain meaningless or corrupted inputs, a property crucial for reliable deployment in real-world scenarios.

## 4.4 ADDITIONAL DISCUSSIONS

**Efficacy under Mild Test Setting** We further evaluate ZeroSiam in the mild setting (Wang et al., 2021), where data comes with shuffled labels. As shown in Table 7, ZeroSiam delivers superior performance compared to prior method, *e.g.*, the average accuracy of 50.1% (ZeroSiam) *vs.* 43.0% (COME), suggesting our efficacy across a wide range of scenarios. Moreover, as in Table 1, ZeroSiam also achieves a lower latency compared to prior state-of-the-arts, *e.g.*, 193s (ZeroSiam) *vs.* 280s (DeYO), highlighting the advantages of ZeroSiam regarding both TTA stability and efficiency. We also offer more results under diverse adaptation epochs over the entire test set in Appendix E.7.

Table 7: Comparisons with state-of-the-art methods on ImageNet-C (severity level 5) under a **MILD SCENARIO** regarding **Accuracy (%)**. "$\mathcal{N}/\mathcal{B}/\mathcal{W}/\mathcal{D}$" denote *Noise/Blur/Weather/Digital* corruptions.

| Method | ResNet50-GN | | | | ViT-Base | | | | ViT-Small | | | | ConvNeXt-Tiny | | | | Swin-Tiny | | | | |
| | $\mathcal{N}$ | $\mathcal{B}$ | $\mathcal{W}$ | $\mathcal{D}$ | $\mathcal{N}$ | $\mathcal{B}$ | $\mathcal{W}$ | $\mathcal{D}$ | $\mathcal{N}$ | $\mathcal{B}$ | $\mathcal{W}$ | $\mathcal{D}$ | $\mathcal{N}$ | $\mathcal{B}$ | $\mathcal{W}$ | $\mathcal{D}$ | $\mathcal{N}$ | $\mathcal{B}$ | $\mathcal{W}$ | $\mathcal{D}$ | Avg. |
|---|---|---|---|---|---|---|---|---|---|---|---|---|---|---|---|---|---|---|---|---|---|
| NoAdapt | 18.6 | 19.4 | 47.7 | 33.9 | 8.1 | 28.4 | 36.1 | 41.6 | 1.8 | 23.4 | 27.7 | 33.5 | 23.7 | 24.8 | 52.1 | 35.4 | 25.8 | 21.7 | 50.5 | 25.9 | 29.0 |
| Tent | 5.5 | 17.7 | 33.1 | 39.0 | 29.0 | 51.3 | 44.7 | 62.1 | 0.5 | 37.2 | 41.4 | 47.2 | 32.1 | 24.0 | 51.9 | 36.6 | 24.2 | 23.1 | 42.6 | 26.9 | 33.5 |
| SAR | 29.9 | 24.6 | 41.2 | 41.8 | 34.7 | 52.4 | 62.5 | 62.6 | 1.5 | 38.6 | 44.1 | 49.0 | 34.2 | 25.0 | 45.4 | 37.8 | 32.4 | 25.1 | 47.3 | 29.0 | 38.0 |
| EATA | 38.6 | 32.8 | 56.9 | 51.4 | 50.1 | 57.4 | 68.2 | 67.2 | 22.2 | 44.7 | 56.4 | 56.2 | 36.2 | 30.6 | 55.5 | 42.6 | 41.8 | 37.5 | 56.5 | 47.1 | 47.5 |
| COME | 18.4 | 19.3 | 47.6 | 33.6 | 51.6 | 58.9 | 70.1 | 69.0 | 0.3 | 41.2 | 56.0 | 58.4 | 41.1 | 26.3 | 60.5 | 43.4 | 42.2 | 25.5 | 58.7 | 38.6 | 43.0 |
| DEYO | 40.8 | 30.4 | 44.0 | 51.4 | 53.3 | 55.0 | 71.1 | 69.6 | 0.2 | 44.7 | 57.8 | 59.0 | 37.3 | 22.1 | 44.3 | 36.2 | 36.1 | 24.6 | 51.0 | 41.1 | 43.5 |
| *ZeroSiam* | 41.6 | 37.1 | 59.7 | 54.2 | 51.0 | 58.4 | 69.6 | 68.3 | 24.2 | 48.1 | 59.6 | 58.8 | 40.2 | 35.4 | 59.5 | 49.0 | 43.1 | 38.2 | 56.6 | 49.8 | **50.1** |

Table 8: Ablation on predictor design in ZeroSiam on ImageNet-C (level 5) with ViT-Base under label shifts.

| Method | Predictor | Acc. |
|---|---|---|
| Tent | - | 47.3 |
| ZeroSiam | FC | **64.1** |
| • Exp1 | *fixed* random FC | 60.7 |
| • Exp2 | FC+ReLU+FC | 64.0 |

Table 9: ZeroSiam's efficacy when used with prior methods. Experiments are on ImageNet-C (level 5) under label shifts.

| Method | ViT-B | R-50 | Con-T | Avg. |
|---|---|---|---|---|
| EATA | 49.4 | 31.6 | 40.7 | 40.6 |
| + *ZeroSiam* | 57.1 | 48.3 | 41.7 | 49.0 |
| DeYO | 62.3 | 43.9 | 33.2 | 46.5 |
| + *ZeroSiam* | 65.0 | 50.7 | 46.1 | 53.9 |

Table 10: Generality of ZeroSiam across objective and divergence choices on ImageNet-C (level 5) with ViT-Base under label shifts.

| Loss | N/A | sKL | KL | rKL | JS | MSE |
|---|---|---|---|---|---|---|
| SLR | 44.0 | 62.2 | 63.2 | 61.0 | 63.4 | 61.5 |
| CE | 43.2 | 61.6 | 61.4 | 61.9 | 61.4 | 62.1 |
| $-p^2$ | 45.8 | 63.3 | 63.4 | 62.1 | 63.0 | 63.6 |
| Tent | 47.3 | 64.1 | 64.0 | 64.1 | 63.8 | 64.1 |

**Ablation on Predictor Design** We further examine other predictor designs in Table 8. Interestingly, replacing the learnable predictor with a fixed, randomly initialized predictor ($\theta_h = \mathbf{I} + 0.1\,\mathbf{W}$, $\mathbf{W} \sim \mathcal{N}(0, \mathbf{I})$) already improves upon Tent, *i.e.*, 47.3%→60.7% w.r.t. accuracy, confirming the value of breaking symmetry alone in collapse mitigation. However, making the predictor deeper and nonlinear brings no further gain (64.0% vs. 64.1%), since the predictor in ZeroSiam is to absorb biased shortcuts for regularization rather than learning rich representations, which do not need large representation capacity. Therefore, we use the single linear predictor for all other experiments.

**Integration with Prior Methods** We further evaluate the efficacy of our ZeroSiam when integrated with state-of-the-art methods. From Table 9, ZeroSiam consistently enhances the performance of prior methods, *e.g.*, improving the average accuracy by +8.4% on EATA and +7.4% on DeYO, suggesting the efficacy of our asymmetric architecture as a plug-and-play component. However, incorporating with EATA and DeYO does not ensure an improvement over ZeroSiam. This is because while EATA and DeYO aim to filter partial noisy samples, ZeroSiam adapts reliably even with incorrect pseudo labels, as verified in Table 6. Thus, ZeroSiam may not directly benefit from the traditional sample selection design and encourages more exploration. We leave it for future work.

**Sensitivity to Learning Rates** We further examine the sensitivity of ZeroSiam to learning rates. When the predictor learning rate is set to zero, the predictor becomes a frozen identity and ZeroSiam degenerates to Tent. From Figure 4, we have the following observations: 1) our ZeroSiam consistently outperforms Tent across a wide range of encoder learning rates, *i.e.*, by $11.3\%-28.2\%$ in accuracy, suggesting the benefit of our asymmetric design. 2) For a fixed encoder learning rate, varying $\eta_h \in \{1, 2.5, 5, 7.5, 10\} \times 0.001$ changes accuracy by only $\approx 1\%$, highlighting our effectiveness does not rely on careful hyperparameter tuning. 3) overall, ZeroSiam achieves both high accuracy and a broad performance plateau under diverse learning rate configurations, *e.g.*, above 62.3% when $\eta_f \in [2.5, 10] \times 10^{-3}$ and $\eta_h \in [1, 10] \times 10^{-3}$, consistently out-

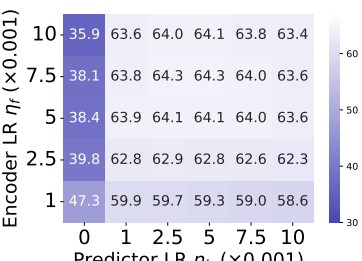

Figure 4: Sensitivity to learning rates. Results are reported on ImageNet-C (level 5) with ViT-Base under label shifts w.r.t. Accuracy.

performing the accuracy of 58.0% in SAR. These results suggest that ZeroSiam is not only effective but also robust to learning rate choices, making it practical and easy to use for online deployment. We also provide more sensitivity analysis of learning rates on Swin-Tiny and further introduce a data-free approach for determining predictor learning rate $\eta_h$ in Appendix E.5 and E.6, respectively.

**Sensitivity of $\alpha$ in ZeroSiam** To avoid heavy tuning in practical TTA use cases, we consistently set $\alpha = 1.0$ in ZeroSiam. From the perspective of learning objective, $\alpha = 1.0$ strikes a balanced and interpretable interaction between the two objectives: 1) comparable gradient scales:

both entropy loss and the divergence loss operate in *probability space* and are also *per-sample normalized*, which yield comparable gradient scales for TTA, without any one dominating optimization; 2) simplified and interpretable formulations: consider Eqn. (4) with $\alpha = 1.0$ detailed as $H(p^o) + D_{KL}(p^o||\text{sg}[p^r]) + D_{KL}(\text{sg}[p^r]||p^o)$. By expanding the terms, we derive an overall formulation of $-p^o \log \text{sg}[p^r] - \text{sg}[p^r] \log p^o$, which expands to a symmetric cross-entropy between online and target branches. This symmetry encourages the two branches to exchange reliable signals while preventing either from drifting too far—an effect we find important for stable adaptation.

Table 11: Sensitivity of $\alpha$ in ZeroSiam. We report **Accuracy (%)** on ImageNet-C across severities with ViT-Base under **ONLINE IMBALANCED LABEL SHIFTS**, where the imbalance ratio is $\infty$.

| Method | No adapt | Tent | $\alpha = 0.5$ | $\alpha = 1.0$ | $\alpha = 1.5$ | $\alpha = 2.0$ | $\alpha = 2.5$ |
|---|---|---|---|---|---|---|---|
| Severity Level=1 | 67.9 | 76.8 | 78.8 | 78.9 | 78.6 | 78.0 | 77.0 |
| Severity Level=3 | 53.8 | 69.5 | 73.9 | 74.1 | 73.8 | 73.2 | 71.8 |
| Severity Level=5 | 29.9 | 47.3 | 63.9 | 64.1 | 63.9 | 63.3 | 62.4 |

we provide a sensitivity analysis of $\alpha$ under different domain shift severities. As shown in Table 11, by aligning the gradient scale of both terms, ZeroSiam achieves consistently the best results with $\alpha = 1$ across severities of domain shift, outperforming Tent's accuracy by 4.6% (at level 3) and 16.8% (at level 5). Extensive results across various test scenarios (Tables 3-7) also validate the efficacy of this setup, making it a reasonable default to avoid careful hyperparameter tuning in TTA.

**Generality of ZeroSiam's architecture** ZeroSiam introduces a general asymmetric architecture without limiting the use of self-training objectives[2] and divergence terms. As shown in Table 10, ZeroSiam remains effective across a wide range of combinations and consistently outperforms the single-branch baselines, *e.g.*, 44.0% (SLR) *vs.* 63.2% (SLR+KL). This arises because collapsed solutions and non-generalizable shortcuts remain pervasive across many unsupervised objectives, suggesting a promising direction for extending ZeroSiam to a wider range of tasks in the future. For entropy minimization, we use symmetric KL as the divergence with $\alpha = 1$ as discussed in Table 11.

## 5 CONCLUSION

In this paper, we explore asymmetry as a new principled test-time entropy minimization mechanism that improves learning stability by inherently avoiding collapsed trivial solutions. To this end, we devise a lightweight Siamese architecture for asymmetric entropy minimization (*termed* ZeroSiam) to prevent the model from being exposed to collapsed solutions, using an injected learnable predictor. The predictor converts collapsed solutions into explicit discrepancies, which are then eliminated by an alignment term. We empirically and theoretically demonstrate that ZeroSiam not only prevents collapse but also absorbs and regularizes biased shortcut signals during learning, thereby boosting the performance even when collapse does not occur. Extensive experiments verify that ZeroSiam performs more stably over prior methods with negligible overhead, proving effective in both vision adaptation and LLM reasoning tasks across challenging models and test scenarios.

## ACKNOWLEDGMENT

This research was supported, in part, by the Joint WeBank-NTU Research Institute on Fintech, Nanyang Technological University, Singapore, and Ministry of Education, Singapore, under its Academic Research Fund Tier 1.

## ETHICS STATEMENT

This research focuses on improving the stability of test-time entropy minimization through an asymmetric Siamese architecture, to advance robustness and reliability in vision and language models for academic and socially beneficial purposes. The proposed approach is evaluated solely on widely used, publicly available models and benchmark datasets under their respective licenses. The study

---

[2]In Table 10, SLR denotes the soft likelihood ratio (Marsden et al., 2024), CE denotes cross-entropy loss with its own predicted labels, and $-p^2$ denotes the negative squared-probability loss, *i.e.*, $-\sum_k p_k^2$.

does not involve human subjects or sensitive personal data, and thus does not raise privacy, fairness, or discrimination concerns. No conflicts of interest or external sponsorship influence this work, and all experiments were conducted in accordance with established standards of research integrity.

## REPRODUCIBILITY STATEMENT

In this work, we implement all methods (all compared methods and our ZeroSiam) with different models (ResNet50-GN, ViT-Base, ViT-Small, ConvNeXt-Tiny, Swin-Tiny, and Llama3.1-8B-Instruct) on the ImageNet-C, Math-500, College-Math, AIME-24, and Minerva datasets. Reproducing all the results in our paper depends on the following three aspects:

1. **DATASETS.** The first paragraph of Section 4 and Appendix C.1 provide the details of the adopted datasets and the download URLs.

2. **MODELS.** All adopted models (with the pre-trained weights) for test-time adaptation are publicly available. Specifically, all vision models: {ResNet50-GN, ViT-Base, ViT-Small, ConvNeXt-Tiny, Swin-Tiny} are from `timm` repository (Wightman, 2019), and Llama3.1-8B-Instruct is from its official repository. Appendix C.2 provides the download URLs of them.

3. **PROTOCOLS OF EACH METHOD.** The second paragraph of Section 4 and Appendix C.2 provides the implementation details of all compared methods and our ZeroSiam. We reproduce all compared methods based on the code from their official GitHub, for which the download `url` is provided (in Appendix C.2) following each method introduction.

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

# APPENDIX

## CONTENTS

## A   RELATED WORK

In this section, we connect test-time entropy minimization and self-supervised learning through their shared challenge of collapse, and relate our approach to these areas.

**Test-Time Entropy minimization** seeks to promote confident predictions by minimizing the prediction entropy on test data. Tent (Wang et al., 2021) first exploits this scheme for test-time model adaptation to novel and out-of-distribution environments. Unlike *test-time training* approaches (Sun et al., 2020; Liu et al., 2021; Bartler et al., 2022; Gandelsman et al., 2022), which train an extra auxiliary self-supervised branch to produce learning signals from test data, test-time entropy minimization (Wang et al., 2021; Niu et al., 2022; Marsden et al., 2024; Lee et al., 2024; Zhang et al., 2025a; Hu et al., 2025b; Zhang et al., 2025b) removes reliance on the source training process and adapts an arbitrary model on-the-fly by using its own predicted labels for self-training, rendering it practical in real-world applications (Niu et al., 2024; Liang et al., 2025). Recently, test-time entropy optimization has attracted growing interest in various fields, such as incentivizing knowledge in large language models during inference (Jang et al., 2025; Zuo et al., 2025; Hu et al., 2025a), and adapting a fixed policy network to evolving environments in real-time (Xu et al., 2025).

Nevertheless, as highlighted by Niu et al. (2023), entropy minimization is inherently unstable and prone to collapse, *e.g.*, converging towards constant one-hot outputs that trivially minimize the entropy loss without meaningful predictions. To improve stability, subsequent methods explored strategies such as sample filtering (Niu et al., 2023; Lee et al., 2024; Hu et al., 2025a), uncertainty quantification (Zhang et al., 2025a), and parameter selection (Chen et al., 2024; Marsden et al., 2024; Zhang et al., 2025c). Despite these advances, collapsed constant solutions remain valid optima in prior methods, so stability often hinges on heuristics, thresholds, or extra compute, leaving methods sensitive to architectures and domains, where robust architectural design is largely underexplored. Our ZeroSiam fills this gap by introducing asymmetry that rules out trivial minima in test-time entropy minimization, while remaining efficient by avoiding augmentations and extra encoder passes.

**Self-supervised Learning** aims to learn discriminative and consistent representations by creating pretext tasks such as contrastive learning (Oord et al., 2018; Saunshi et al., 2019; He et al., 2020; Chen et al., 2020). A central challenge in SSL is trivial collapse, where all outputs degenerate to a constant to maximize representation similarity. Contrastive methods such as SimCLR (Chen et al., 2020) and MoCo (He et al., 2020) prevent collapse by pushing apart negative pairs, but rely on large batch sizes, memory banks, or careful negative mining strategies. To remove dependency on negative samples, BYOL (Grill et al., 2020) and SimSiam (Chen & He, 2021) introduce asymmetric architectures, preventing collapse with a *predictor–stop-gradient asymmetry* to break the stability of the collapsed constant solution. Although subsequent methods such as Barlow Twins (Zbontar et al., 2021) and VicReg (Bardes et al., 2022) achieve stability without symmetry-breaking design by maximizing feature diversity within a batch, they implicitly assume large batches and i.i.d. streams, which often fail during testing. Inspired by Simsiam, we revisit asymmetry in the context of single-branch, entropy-based TTA, and show how it can be instantiated in a minimal and efficient manner.

## B  THEORETICAL ANALYSIS

**Notations.** Let $\mathcal{L} = H(p^o) + \alpha\,D(p^o\,\|\,\text{sg}[p^r])$ be the optimization objective of ZeroSiam, where $H(\cdot)$ is the entropy loss, $D(\cdot)$ the alignment regularizer, and $p^o, p^r \in \Delta^{|\mathcal{C}|-1}$ are the probability distributions induced by the online and target branches. Let the encoder $f$ and classifier $g$ be pretrained, and $h$ a lightweight *linear* predictor. At initialization, we set $h$ as the identity mapping $\mathbf{I}$, so that the online and target branches coincide and both lie in the non-collapse region $\{p \in \Delta^{|\mathcal{C}|-1}|\min_c p_c \geq \delta > 0\}$.

### B.1  ASSUMPTIONS

**Assumption 1.** *(**Lipschitz continuity of the encoder**) Let $f(\cdot; \theta_f)$ be an encoder parameterized by $\theta_f$. We assume $f$ is Lipschitz continuous with respect to its parameters. Specifically, there exists a constant $L_f > 0$ such that for any $\theta_1, \theta_2$ and any input $\mathbf{x}$ in the domain,*

$$\|f_{\theta_1}(\mathbf{x}) - f_{\theta_2}(\mathbf{x})\|_2 \leq L_f\|\theta_1 - \theta_2\|.$$

Assumption 1 bounds the sensitivity of the encoder $f$ to parameter perturbations, thereby ensuring stability for subsequent optimization analysis, and is commonly employed in prior theoretical studies on representation learning and model convergence.

**Assumption 2.** *(**Standard optimization assumptions**) Let $\mathcal{L}(\cdot)$ denote the objective function, $\phi = (\theta_h, \theta_f)$ the set of optimization parameters, and $\eta_h, \eta_f$ the learning rates for the predictor and encoder, respectively. Our analysis relies on the following standard assumptions, which are common in the optimization literature:*
*(1) Smoothness: $\mathcal{L}$ is $\rho$-smooth. Specifically, there exists $\rho > 0$ such that for all $\phi_1, \phi_2$,*

$$\|\nabla_\phi \mathcal{L}(\phi_1) - \nabla_\phi \mathcal{L}(\phi_2)\| \leq \rho\|\phi_1 - \phi_2\|;$$

*(2) Descent condition: The learning rates satisfy $\max(\eta_h, \eta_f) < \frac{1}{\rho}$, ensuring monotonic descent.*

Assumption 2 constrains the appropriate range of learning rates to guarantee monotonic descent of $\mathcal{L}(\cdot)$, which is a common assumption in optimization research.

### B.2  PROOF OF LEMMA 1

**Lemma 1.** *(**Gradient dominance in entropy descent**) Let $H(\cdot)$ denote the entropy function, $g(\cdot; \theta_g), h(\cdot; \theta_h)$ and $f(\cdot; \theta_f)$ denote the classifier, the predictor and the encoder, $\phi = (\theta_h, \theta_f)$ the set of optimization parameters, and $\eta_h, \eta_f$ the learning rates for the predictor and encoder, respectively. Then the discrete change in entropy of the target branch satisfies:*

$$\|\nabla_\phi H(p^o(t))\| \geq \frac{1}{L_H\,\|g\|_2\,L_f \cdot \max(\eta_h, \eta_f)} \cdot |H(p^r(t+1)) - H(p^r(t))|,$$

*where $L_H, L_f$ are the Lipschitz constants of $H(\cdot)$ and $f$, $\|g\|_2$ denotes the spectral norm of the classifier weight vector, and $t$ is the optimization iteration step.*

*Proof.* Consider the online and target branches:

$$\begin{cases} p^o = \text{softmax}(g(h(f(\mathbf{x}; \theta_f); \theta_h); \theta_g)) & \text{(online branch)}, \\ p^r = \text{softmax}(g(f(\mathbf{x}; \theta_f); \theta_g)) & \text{(target branch)}. \end{cases}$$

By design, the encoder $f$ of the target branch is synchronized with the online branch after each update:

$$\theta_f^r(t+1) = \theta_f^o(t+1) = \theta_f^o(t) - \eta_f \nabla_{\theta_f} H(p^o(t)).$$

Thus, we bound the entropy change of the target branch $|H(p^r(t+1)) - H(p^r(t))|$ by tracing the propagation of parameter updates through the online branch.

According to the Assumption 1, the encoder $f$ is $L_f$-Lipschitz continuous w.r.t. $\theta_f$, we have

$$\|\mathbf{z}(t+1) - \mathbf{z}(t)\|_2 = \|f(\mathbf{x}; \theta_f(t+1)) - f(\mathbf{x}; \theta_f(t))\|_2 \leq L_f\|\theta_f(t+1) - \theta_f(t)\|. \quad (5)$$

where $\mathbf{z} = f(\mathbf{x}; \theta_f)$ is the output feature of the encoder.

Note that in the non-collapse region $\{p \in \Delta^{|\mathcal{C}|-1} | \min_c p_c \geq \delta > 0\}$, the entropy function $H(\cdot)$ is Lipschitz continuous with constant $L_H = \sqrt{|\mathcal{C}|}(|\log \delta| + 1)$, thus:

$$|H(p(t+1)) - H(p(t))| \leq L_H \|p(t+1) - p(t)\|_2. \tag{6}$$

Since the pre-trained classifier $g$ is fixed, and the softmax function is 1-Lipschitz, combining Eqn. (5) and Eqn. (6), we have the following chain:

$$\begin{aligned}
|H(p^r(t+1)) - H(p^r(t))| &\leq L_H \|p^r(t+1) - p^r(t)\|_2 \\
&\leq L_H \|g\|_2 \|\mathbf{z}^r(t+1) - \mathbf{z}^r(t)\|_2 \\
&\leq L_H \|g\|_2 L_f \|\theta_f^r(t+1) - \theta_f^r(t)\| \\
&= L_H \|g\|_2 L_f \eta_f \|\nabla_{\theta_f} H(p^o(t))\|,
\end{aligned}$$

where $\|g\|_2$ is the spectral norm of the classifier weight vector.

Let $\phi = (\theta_h, \theta_f)$ denote the set of optimization parameters, and $\eta_h, \eta_f$ the learning rates for the predictor and encoder. Since:

$$\eta_f \leq \max(\eta_h, \eta_f),$$

$$\|\nabla_{\theta_f} H(p^o(t))\| \leq \left\| \begin{bmatrix} \nabla_{\theta_f} H(p^o(t)) \\ \nabla_{\theta_h} H(p^o(t)) \end{bmatrix} \right\| = \|\nabla_\phi H(p^o(t))\|,$$

we obtain:

$$\|\nabla_\phi H(p^o(t))\| \geq \frac{1}{L_H \|g\|_2 L_f \cdot \max(\eta_h, \eta_f)} \cdot |H(p^r(t+1)) - H(p^r(t))|.$$

$\square$

### B.3 PROOF OF LEMMA 2

**Lemma 2.** *(Collapse Direction with Maximal Negative Curvature) Let $H(\cdot)$ denote the entropy function, $u = g(h(f(\mathbf{x}; \theta_f); \theta_h); \theta_g)$ the uncertainty logits, and $p = softmax(u)$ the induced probability distribution. For any class $c \in \mathcal{C}$, there exists a direction $\mathbf{v} = (\mathbf{v}_h, \mathbf{v}_f) \in \mathbb{R}^{\dim(\phi)}$ such that:*
*(1) $p_c$ increases monotonically along $\mathbf{v}$;*
*(2) As $p_c \to 1$, it holds that $\mathbf{v}^\top \nabla_\phi^2 H \mathbf{v} \to \lambda_{\min}(\nabla_\phi^2 H)$;*
*(3) For any $\mathbf{w} \perp \mathbf{v}$, we have $\mathbf{v}^\top \nabla_\phi^2 H \mathbf{v} \leq \mathbf{w}^\top \nabla_\phi^2 H \mathbf{w}$.*

*Proof.* (1) Let $\mathbf{z} = f(\mathbf{x}; \theta_f) \in \mathbb{R}^d$ be the encoder output, $u = g(h(\mathbf{z}; \theta_h); \theta_g) \in \mathbb{R}^{|\mathcal{C}|}$ the uncertainty logits, and $\mathbf{W}, \mathbf{P}$ the weight vector of $g, h$, respectively. Define $\mathbf{v} = (\mathbf{v}_h, \mathbf{v}_f)$ such that:

$$\begin{cases} \mathbf{v}_h = a \cdot \mathbf{W}_k^\top r, \\ \mathbf{v}_f = \nabla_{\theta_f}(r^\top \mathbf{z}); \end{cases}$$

where $r$ is a feature direction satisfying $r^\top \mathbf{z} \neq 0$ and $a > 0$. $\mathbf{v}_f$ is the direction in normalization layer parameters that amplifies the feature component along $r$.

This joint direction $\mathbf{v} = (\mathbf{v}_h, \mathbf{v}_f)$ induces a change in logits:

$$\Delta u = \mathbf{W}(\mathbf{v}_h)\mathbf{z} + \mathbf{W}\mathbf{P}(\nabla_{\theta_f} \mathbf{z} \mathbf{v}_f) \approx a(r^\top \mathbf{z})\mathbf{W}\mathbf{W}_k^\top + \mathbf{W}\mathbf{P}r.$$

By choosing $r^\top \mathbf{z} > 0$ and sufficiently large $a$, we ensure:

$$[\Delta u]_c > 0, \quad [\Delta u]_j = \mathcal{O}(1) \text{ for } j \neq c.$$

Thus, moving along $\mathbf{v}$ increases $u_c$, and hence $p_c = \text{softmax}(u_c)$, monotonically.

(2) The Hessian of entropy $H(\cdot)$ w.r.t. logits $u$ is:

$$[\nabla_u^2 H]_{ij} = \frac{\partial^2 H}{\partial u_i \partial u_j} = p_i(\delta_{ij} - p_j)(H - \log p_i - 1).$$

As $p_c \to 1$, let $\mu = 1 - p_c \to 0^+$, and for $p_j \approx \frac{\mu}{|\mathcal{C}|-1}$ for $j \neq c$. Then in the Hessian matrix

$$
\begin{cases}
\frac{\partial^2 H}{\partial u_c^2} = p_c(1 - p_c)(H - \log p_c - 1), & \text{Diagonal for } c, \\
\frac{\partial^2 H}{\partial u_j^2} = p_j(1 - p_j)(H - \log p_j - 1), & \text{Diagonal for } j \neq c, \\
\frac{\partial^2 H}{\partial u_i \partial u_j} = -p_i p_j(H - \log p_i - \log p_j - 1), & \text{Off-diagonal,}
\end{cases}
$$

we obtain the following approximation:

$$
\begin{cases}
\frac{\partial^2 H}{\partial u_c^2} \approx (1 - \mu)\mu(H + \mu - 1) \approx \mu(\mu \log \frac{|\mathcal{C}|-1}{\mu} + \mu - 1) \approx -\mu, \\
\frac{\partial^2 H}{\partial u_j^2} \approx \frac{\mu}{|\mathcal{C}|-1}(\mu \log \frac{|\mathcal{C}|-1}{\mu} - \log \frac{\mu}{|\mathcal{C}|-1} - 1) \approx \frac{\mu}{|\mathcal{C}|-1} \log \frac{|\mathcal{C}|-1}{\mu} > 0, \\
\frac{\partial^2 H}{\partial u_i \partial u_j} \approx -p_i p_j(-\log p_j) > 0.
\end{cases}
$$

That is, as $p_c \to 1$, we have

$$
\mathbf{v}^\top \nabla_\phi^2 H \mathbf{v} \to \lambda_{\min}(\nabla_\phi^2 H).
$$

(3) The parameter space Hessian quadratic form along $\mathbf{v}$ is:

$$
\mathbf{v}^\top \nabla_\phi^2 H \mathbf{v} = \Delta u^\top \nabla_u^2 H \Delta u + \sum_i \frac{\partial H}{\partial u_i} \mathbf{v}^\top \nabla_\phi^2 u_i \mathbf{v}.
$$

As $p_c \to 1$, $\nabla_u H \to 0$, so the higher-order terms vanishes. Since $\Delta u = \frac{\partial u}{\partial \phi} \propto e_c$, we have:

$$
\mathbf{v}^\top \nabla_\phi^2 H \mathbf{v} \propto e_c^\top \nabla_u^2 H e_c = \frac{\partial^2 H}{\partial u_c^2} < 0.
$$

For any $\mathbf{w} \perp \mathbf{v}$, $\Delta u_\mathbf{w} = \frac{\partial u}{\partial \phi} \mathbf{w}$ is not parallel to $e_c$, so:

$$
\mathbf{w}^\top \nabla_\phi^2 H \mathbf{w} \geq \mathbf{v}^\top \nabla_\phi^2 H \mathbf{v}.
$$

$\square$

### B.4 PROOF OF THEOREM 1

**Theorem 1.** *(Optimization and Stability of ZeroSiam)* *Consider the ZeroSiam objective $\mathcal{L} = H(p^o) + \alpha D(p^o \,\|\, \mathrm{sg}[p^r])$, where $H(\cdot)$ denotes the entropy loss, $D(\cdot)$ the alignment regularizer, and $p^o, p^r \in \Delta^{|\mathcal{C}|-1}$ are the probability distributions induced by the online and target branches. Given the encoder $f$, classifier $g$ and predictor $h$. Under Assumptions 1 and 2, the following hold:*
*(1) For $\alpha = 0$, the entropy variation satisfies $|\Delta H(p^o)| > |\Delta H(p^r)|$, and the Hessian of $H(p^o)$ attains its minimal eigenvalue along collapse directions $\mathbf{v}$:*

$$
\lambda_{\min}\big(\nabla^2 H(p^o)\big) = \mathbf{v}^\top \nabla^2 H(p^o) \mathbf{v}.
$$

*(2) For $\alpha > 0$, the predictor $h$ serves as a filtering mechanism that suppresses gradient update directions corresponding to over-amplified logits, and the system converges to a stable equilibrium: there exists $h_{\min} > 0$ such that*

$$
H(p^o) > h_{\min}, \ p^o \to p^r.
$$

*Proof.* (1) By Lemma 1, the target branch entropy variation is determined by the gradient of the online branch:

$$
\|\nabla_\phi H(p^o(t))\| \geq \frac{1}{L \cdot \eta_{\max}} \cdot |H(p^r(t+1)) - H(p^r(t))|, \tag{7}
$$

where $L = L_H \|g\|_2 L_f$ and $\eta_{\max} = \max(\eta_h, \eta_f)$.

Under the Assumption 2 that $\mathcal{L} = H(\cdot)$ is $\rho$-smoothness and $\eta_{\max} < \frac{1}{\rho}$, the descent lemma gives:

$$
H(p^o(t+1)) \leq H(p^o(t)) - \nabla_\phi H^\top D_\eta \nabla_\phi H + \frac{\rho}{2}\|D_\eta \nabla_\phi H\|^2,
$$

where $D_\eta = \mathrm{diag}(\eta_f \mathbf{I}_{d_f}, \eta_h \mathbf{I}_{d_h})$. So we have:

$$
|\Delta H^o| = H(p^o(t+1)) - H(p^o(t)) \geq \eta_{\min}\|\nabla_\phi H\|^2, \ \eta_{\min} = \min(\eta_h, \eta_f). \tag{8}
$$

Combining Eqn. (7) and Eqn. (8), we obtain:

$$\frac{|\Delta H^o|}{|\Delta H^r|} \geq \frac{\eta_{\min}\|\nabla_\phi H\|^2}{L \cdot \eta_{\max}\|\nabla_\phi H\|} = \frac{\eta_{\min}}{\eta_{\max}} \cdot \frac{\|\nabla_\phi H\|}{L}.$$

Since $L, \|\nabla_\phi H\| > 0$ and bounded in non-collapse regions, achieving $\frac{|\Delta H^o|}{|\Delta H^r|} > 1$ requires:

$$\frac{\eta_{\min}}{\eta_{\max}} > \frac{L}{\|\nabla_\phi H\|}.$$

In practice, we set $\eta_h = k \cdot \eta_f$ with $k > 1$, so there always exists a $k$ such that

$$|\Delta H(p^o)| > |\Delta H(p^r)|.$$

We further construct the direction $\mathbf{v} = (\mathbf{v}_h, \mathbf{v}_f) \in \mathbb{R}^{\dim(\phi)}$ according to Lemma 2, the corresponding directional derivative is given by:

$$D_{\mathbf{v}}H = \nabla_\phi H^\top \mathbf{v} \propto [\nabla_u H]_c,$$

where $u$ is the uncertainty logits. And the second-order directional derivative is:

$$D_{\mathbf{v}}^2 H = \mathbf{v}^\top \nabla_\phi^2 H \mathbf{v} = \lambda_{\min} < 0.$$

This implies that along $\mathbf{v}$, the loss decreases fastest at second order (Jin et al., 2017).

Moreover, by Lemma 2, $p_c$ increases monotonically along $\mathbf{v}$, thus $[\nabla_u H]_c = p_c(H - \log p_c - 1)$ increases synchronously, creating a self-reinforcing feedback loop that accelerates collapse.

(2) Consider the optimization objective:

$$\mathcal{L} = H(p^o) + \alpha\, D(p^o \,\|\, \mathrm{sg}[p^r]),$$

where $D(p^o \,\|\, \mathrm{sg}[p^r]) = D_{\mathrm{KL}}(p^o \,\|\, \mathrm{sg}[p^r]) + D_{\mathrm{KL}}(\mathrm{sg}[p^r] \,\|\, p^o)$ are symmetric KL loss.

Under the stop-gradient setting on the target branch, the gradient of the shared encoder $f$ is solely determined by the online branch. We have:

$$\nabla_{\mathbf{z}}\mathcal{L} = (\frac{\partial u^o}{\partial \mathbf{z}})\nabla_{u^o}\mathcal{L} = \mathbf{P}^\top \mathbf{W}^\top[\nabla_{u^o}H(p^o) + 2\alpha(p^o - p^r)],$$

where $\mathbf{z} = f(\mathbf{x}; \theta_f)$ is the encoder output, $\mathbf{W}, \mathbf{P}$ the weight vector of the classifier $g$ and the predictor $h$, respectively. Consider a parameter update unit direction $\mathbf{v_h}, \|\mathbf{v_h}\|_2 = 1$, then:

$$< \nabla_{\mathbf{z}}\mathcal{L}, \mathbf{v_h} >= \mathbf{v_h}^\top \nabla_{\mathbf{z}}\mathcal{L} = \mathbf{v_h}^\top(\frac{\partial u^o}{\partial \mathbf{z}})\nabla_{u^o}\mathcal{L} = \mathbf{P}^\top\mathbf{W}^\top[\nabla_{u^o}H(p^o) + 2\alpha(p^o - p^r)]$$

During the minimization of $\mathcal{L}$, the predictor $h$ encourages the update direction that ensuring

$$p^o \to p^r,$$

and suppresses gradient update directions corresponding to over-amplified logits.

We further analyze the convergence properties of the optimization system.

By the definition of entropy and KL divergence, we have:

$$H(p^o) = -\sum_{c=1}^{|\mathcal{C}|} p_c^o \log p_c^o = -\sum_{c=1}^{|\mathcal{C}|} p_c^o \log p_c^r - D_{\mathrm{KL}}(p^o\|p^r).$$

Applying Gibbs' inequality, $-\sum_{c=1}^{|\mathcal{C}|} p_c^o \log p_c^r \geq H(p^r)$, we obtain:

$$H(p^o) \geq H(p^r) - D_{\mathrm{KL}}(p^o\|p^r).$$

Since $D_{\mathrm{SKL}}(p^o\|p^r) \geq D_{\mathrm{KL}}(p^o\|p^r)$, it follows that:

$$H(p^o) \geq H(p^r) - D_{\mathrm{SKL}}(p^o\|p^r). \tag{9}$$

When the target branch is still in the non-collapse region $\{p^r \in \Delta^{|\mathcal{C}|-1}\,|\,\min_c p^r_c \geq \delta > 0\}$, the entropy satisfies:

$$H(p^r) \geq -\log(1 - (1 - |\mathcal{C}|)\delta) > 0. \tag{10}$$

Define the Lyapunov function $V(t) = \mathcal{L}(t)$. Since $H(p^o) \geq 0$ and $D_{\text{SKL}}(p^o||p^r) \geq 0$, we have

$$V(t) > 0.$$

Moreover, under the Assumption 2, the descent lemma guarantees:

$$V(t+1) \geq V(t) - \frac{\eta_{\min}}{2}\|\nabla_\phi \mathcal{L}(t)\|^2 \leq V(t).$$

Thus, $V(t)$ is monotonically decreasing and bounded below. According to the Monotone Convergence Theorem, $V(t) \to V^* > 0$. In particular:

$$D_{\text{SKL}}(p^o(t)||p^r(t)) \leq \frac{V(0)}{\alpha}. \tag{11}$$

Combining Eqn. (9), Eqn. (10), and Eqn. (11):

$$H(p^o) \geq H(p^r) - D_{\text{SKL}}(p^o||p^r) \geq -\log(1 - (1 - |\mathcal{C}|)\delta) - \frac{V(0)}{\alpha}.$$

Define

$$h_{\min} = -\log(1 - (1 - |\mathcal{C}|)\delta) - \frac{V(0)}{\alpha}.$$

Let the hyperparameter $\alpha > \frac{V(0)}{-\log(1-(1-|\mathcal{C}|)\delta)}$, then $h_{\min} > 0$ .

Overall, we have proven that the predictor $h$ serves as a filtering mechanism that suppresses gradient update directions corresponding to over-amplified logits, and the system converges to a stable equilibrium: there exists $h_{\min} > 0$ such that

$$H(p^o) > h_{\min}, \; p^o \to p^r.$$

$\square$

## C   MORE IMPLEMENTATION DETAILS

### C.1   MORE DETAILS ON DATASET

For vision adaptation, we evaluate the out-of-distribution generalization ability of all methods on a large-scale and widely used benchmark, namely **ImageNet-C**[3] (Hendrycks & Dietterich, 2019). ImageNet-C is constructed by corrupting the original ImageNet (Deng et al., 2009) test set. The corruption (as shown in Figure A) consists of 15 different types, *i.e.*, Gaussian noise, shot noise, impulse noise, defocus blur, glass blur, motion blur, zoom blur, snow, frost, fog, brightness, contrast, elastic transformation, pixelation, and JPEG compression, where each corruption type has 5 severity levels and the larger severity level means more severe distribution shift.

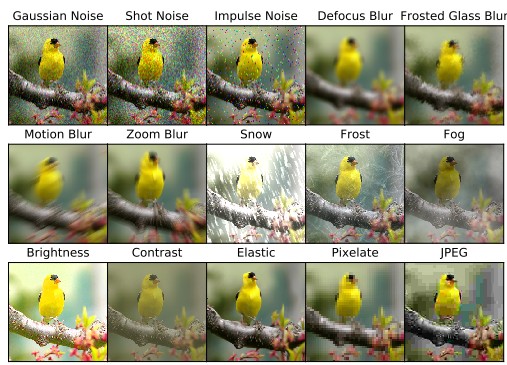

Figure A: Visualizations of different corruption types in ImageNet-C benchmark.

For natural language reasoning, we evaluate the effectiveness of online reasoning incentivization on a comprehensive mathematical reasoning benchmark, including Math-500 (Lightman et al., 2023), CollegeMath (Tang et al., 2024), AIME24 (Codeforces, 2024), and Minerva (Lewkowycz et al., 2022). Specifically, Math-500 is derived from the larger MATH benchmark (Hendrycks et al., 2021), consisting of difficult mathematics problems originally taken from high school competitions. It spans domains such as pre-algebra, algebra, number theory, and calculus, emphasizing abstract reasoning and multi-step problem solving. CollegeMath is a college-level mathematics dataset constructed from nine open-access college textbooks, covering seven core areas: algebra, pre-calculus, calculus, vector calculus, probability, linear algebra, and differential equations. The full dataset contains 2818 test questions, from which we sample 1200 for efficient evaluation. AIME24 consists of 30 advanced problems from the 2024 American Invitational Mathematics Examination (AIME) I and II, designed to test extended chain-of-thought reasoning. Minerva includes 272 undergraduate-level mathematics and science questions from MIT OpenCourseWare (OCW), aimed at assessing models' reasoning ability in scientific problem-solving contexts.

### C.2   MORE EXPERIMENTAL PROTOCOLS ON METHODS

All pre-trained models involved in our paper are publicly available, including ResNet50-GN[4], ViT-Base[5], ViT-Small[6], ConvNeXt-Tiny[7] and Swin-Tiny[8] obtained from `timm` repository (Wightman,

---

[3] https://zenodo.org/record/2235448#.YzQpq-xBxcA
[4] https://github.com/rwightman/pytorch-image-models/releases/download/v0.1-rsb-weights/resnet50_gn_a1h2-8fe6c4d0.pth
[5] https://storage.googleapis.com/vit_models/augreg/B_16-i21k-300ep-lr_0.001-aug_medium1-wd_0.1-do_0.0-sd_0.0--imagenet2012-steps_20k-lr_0.01-res_224.npz
[6] https://storage.googleapis.com/vit_models/augreg/S_16-i21k-300ep-lr_0.001-aug_light1-wd_0.03-do_0.0-sd_0.0--imagenet2012-steps_20k-lr_0.03-res_224.npz
[7] https://github.com/rwightman/pytorch-image-models/releases/download/v0.1-rsb-weights/convnext_tiny_hnf_a2h-ab7e9df2.pth
[8] https://github.com/SwinTransformer/storage/releases/download/v1.0.0/swin_tiny_patch4_window7_224.pth

2019) for image classification, and Llama3.1-8B-Instruct[9] (Dubey et al., 2024) for natural language reasoning. In the following, we provide the implementation details of our proposed method and all comparative methods evaluated in our experiments, which help reproduce our results.

**ZeroSiam (Ours).** We use symmetric KL as the divergence term in Eqn. (4), and $\alpha$ is fixed to 1 without tuning. For vision adaptation, we use SGD as the update rule, with a momentum of 0.9, batch size of 64 (except for the experiments of batch size = 1), learning rate $\eta_f$ of 0.0025 / 0.005 / 0.001 / 0.00025 / 0.0005 with learning rate $\eta_h$ of 0.025 / 0.005 / 0.01 / 0.0025 / 0.0025 for ResNet50-GN / ViT-Base / ViT-Small / ConvNeXt-Tiny / Swin-Tiny, respectively. The learning rate for batch size = 1 is scaled down by 16 for ResNet50-GN and 32 for other models following SAR. The trainable parameters are all the affine parameters of normalization layers. For natural language reasoning, following the configuration for TLM, we use AdamW as the update rule, with $\beta_1 = 0.9$, $\beta_2 = 0.999$, weight decay = 0.0, and optimize the prediction entropy of the first 8 output tokens. Both learning rates $\eta_f$ and $\eta_h$ are set to $7.5\times10^{-6}$ / $5\times10^{-6}$ / $1.25\times10^{-5}$ / $5\times10^{-6}$ for Math-500 / CollegeMath / AIME24 / Minerva. Trainable parameters are the LoRA parameters with a rank of 8.

**Tent[10] (Wang et al., 2021).** We follow all hyper-parameters that are set in Tent unless it does not provide. Specifically, for vision adaptation, we use SGD as the update rule, with a momentum of 0.9, batch size of 64 (except for the experiments of batch size = 1), and learning rate of 0.00025 / 0.001 / 0.0001 / 0.000025 / 0.00005 for ResNet50-GN / ViT-Base / ViT-Small / ConvNeXt-Tiny / Swin-Tiny, respectively. The learning rate for batch size = 1 is scaled down to (0.00025/32) for ResNet50-GN, (0.001/64) for ViT-Base, (0.0001/64) for ViT-Small, (0.000025/64) for ConvNeXt-Tiny and (0.00005/64) for Swin-Tiny. The trainable parameters are all the affine parameters of normalization layers. For natural language reasoning, we use AdamW as the update rule, with $\beta_1 = 0.9$, $\beta_2 = 0.999$, weight decay = 0.0, and optimize the prediction entropy of the first 8 output tokens. The learning rate is $7.5\times10^{-6}$ / $5\times10^{-6}$ / $1.25\times10^{-5}$ / $5\times10^{-6}$ for Math-500 / CollegeMath / AIME24 / Minerva. The trainable parameters are the LoRA parameters with a rank of 8.

**SAR[11] (Niu et al., 2023).** We follow all hyper-parameters that are set in SAR unless it does not provide. Specifically, for vision adaptation, we use SGD as the update rule, with a momentum of 0.9, batch size of 64 (except for the experiments of batch size = 1), and learning rate of 0.00025 / 0.001 / 0.0001 / 0.000025 / 0.00005 for ResNet50-GN / ViT-Base / ViT-Small / ConvNeXt-Tiny / Swin-Tiny. The learning rate for batch size = 1 is scaled down to (0.00025/16) for ResNet50-GN, (0.001/32) for ViT-Base, (0.0001/32) for ViT-Small, (0.000025/32) for ConvNeXt-Tiny and (0.00005/32) for Swin-Tiny. The entropy threshold $E_0$ is set to $0.4\times \ln C$, where $C$ is the number of task classes. The trainable parameters are the affine parameters of norm layers from layer 1 to layer 3 in ResNet50-GN, from blocks 1 to blocks 8 in ViT-Base and ViT-Small, and all affine parameters in other models. For natural language reasoning, $E_0$ is set to 0.4. We use AdamW as the update rule, with $\beta_1 = 0.9$, $\beta_2 = 0.999$, weight decay = 0.0, and optimize the prediction entropy of the first 8 output tokens. The learning rate is $7.5\times10^{-6}$ / $5\times10^{-6}$ / $1.25\times10^{-5}$ / $5\times10^{-6}$ for Math-500 / CollegeMath / AIME24 / Minerva, trainable parameters are the LoRA parameters with a rank of 8.

**EATA[12] (Niu et al., 2022).** We follow all hyper-parameters that are set in EATA unless it does not provide. Specifically, for vision adaptation, the entropy constant $E_0$ (for reliable sample identification) is set to $0.4\times \ln C$, where $C$ is the number of task classes. The $\epsilon$ for redundant sample identification is set to 0.05. The trade-off parameter $\beta$ for entropy loss and regularization loss is set to 2,000. The number of pre-collected in-distribution test samples for Fisher importance calculation is 2,000. The update rule is SGD, with a momentum of 0.9, batch size of 64 (except for the experiments of batch size = 1), and learning rate of 0.00025 / 0.001 / 0.0001 / 0.000025 / 0.00005 for ResNet50-GN / ViT-Base / ViT-Small / ConvNeXt-Tiny / Swin-Tiny. The learning rate for batch size = 1 is scaled down to (0.00025/32) for ResNet50-GN, (0.001/64) for ViT-Base, (0.0001/64) for ViT-Small, (0.000025/64) for ConvNeXt-Tiny and (0.00005/64) for Swin-Tiny. The trainable parameters are all affine parameters of normalization layers. For natural language reasoning, $E_0$ is set to 0.4, while the anti-forgetting regularizer is not applied due to a lack of knowledge for in-distribution data that the large language model trains on. We use AdamW as the update rule, with $\beta_1 = 0.9$, $\beta_2 = 0.999$, weight decay = 0.0, and optimize the prediction entropy of the first 8 output tokens. The

---

[9] https://huggingface.co/meta-llama/Llama-3.1-8B-Instruct
[10] https://github.com/DequanWang/tent
[11] https://github.com/mr-eggplant/SAR
[12] https://github.com/mr-eggplant/EATA

learning rate is $7.5 \times 10^{-6}$ / $5 \times 10^{-6}$ / $1.25 \times 10^{-5}$ / $5 \times 10^{-6}$ for Math-500 / CollegeMath / AIME24 / Minerva. The trainable parameters are the LoRA parameters with a rank of 8.

**DeYO**[13] **(Lee et al., 2024).** We follow all hyper-parameters that are set in DeYO unless it does not provide. Specifically, the entropy constant $E_0$ (for reliable sample identification) is set to $0.4 \times \ln 1000$, and the factor $\tau_{\text{Ent}}$ is set to $0.5 \times \ln 1000$. The Pseudo-Label Probability Difference (PLPD) threshold $\tau_{\text{PLPD}}$ is set to 0.2. The update rule is SGD, with a momentum of 0.9, batch size of 64 (except for the experiments of batch size = 1), and learning rate of 0.00025 / 0.001 / 0.0001 / 0.000025 / 0.00005 for ResNet50-GN / ViT-Base / ViT-Small / ConvNeXt-Tiny / Swin-Tiny. The learning rate for batch size = 1 is scaled down to (0.00025/16) for ResNet50-GN, (0.001/32) for ViT-Base, (0.0001/32) for ViT-Small, (0.000025/32) for ConvNeXt-Tiny and (0.00005/32) for Swin-Tiny. Trainable parameters are the affine parameters of norm layers from layer 1 to 3 in ResNet50-GN, from blocks 1 to 8 in ViT-Base and ViT-Small, and all affine parameters in other models.

**COME**[14] **(Zhang et al., 2025a).** For vision adaptation, we implement COME based on DeYO for comparisons, following the hyper-parameters set in DeYO unless otherwise specified. Specifically, the entropy constant $E_0$ (for reliable sample identification) is set to $0.4 \times \ln 1000$, and the factor $\tau_{\text{Ent}}$ is set to $0.5 \times \ln 1000$. The Pseudo-Label Probability Difference (PLPD) threshold $\tau_{\text{PLPD}}$ is set to 0.2. The uncertainty mass is set to $C$, where $C$ is the number of task classes. The update rule is SGD, with a momentum of 0.9, batch size of 64 (except for the experiments of batch size = 1), and learning rate of 0.00025 / 0.001 / 0.0001 / 0.000025 / 0.00005 for ResNet50-GN / ViT-Base / ViT-Small / ConvNeXt-Tiny / Swin-Tiny. The learning rate for batch size = 1 is scaled down to (0.00025/16) for ResNet50-GN, (0.001/32) for ViT-Base, (0.0001/32) for ViT-Small, (0.000025/32) for ConvNeXt-Tiny and (0.00005/32) for Swin-Tiny. The trainable parameters are the affine parameters of norm layers from layer 1 to layer 3 in ResNet50-GN, from blocks 1 to blocks 8 in ViT-Base and ViT-Small, and all affine parameters in other models. For natural language reasoning, COME is implemented based on Tent, and uncertainty mass is set to $|Z|$, where $|Z|$ denotes the vocabulary size. We use AdamW as the update rule, with $\beta_1 = 0.9$, $\beta_2 = 0.999$, weight decay = 0.0, and optimize the prediction entropy of the first 8 output tokens. The learning rate is $7.5 \times 10^{-6}$ / $5 \times 10^{-6}$ / $1.25 \times 10^{-5}$ / $5 \times 10^{-6}$ for Math-500 / CollegeMath / AIME24 / Minerva. The trainable parameters are the LoRA parameters with a rank of 8.

**TLM (Hu et al., 2025a)**[15]**.** We follow all hyper-parameters that are set in TLM unless it does not provide. Specifically, we compare with TLM on the natural language reasoning task, the perplexity threshold $\mathcal{P}_t$ is set to $e^3$. The update rule is AdamW, with $\beta_1 = 0.9$, $\beta_2 = 0.999$, weight decay = 0.0. The learning rate is $7.5 \times 10^{-6}$ / $5 \times 10^{-6}$ / $1.25 \times 10^{-5}$ / $5 \times 10^{-6}$ for Math-500 / CollegeMath / AIME24 / Minerva. The trainable parameters are the LoRA parameters with a rank of 8.

---

[13] https://github.com/Jhyun17/DeYO
[14] https://github.com/BlueWhaleLab/COME
[15] https://github.com/Fhujinwu/TLM

# D   MORE EXPERIMENTAL RESULTS

In this section, we provide the detailed results of Tables 3-6 in the main paper and include additional experiments under both mild and wild testing scenarios to enable a comprehensive comparison.

## D.1   DETAILED AND ADDITIONAL RESULTS UNDER WILD TEST SCENARIOS

We first provide the detailed results of TTA under label shifts and a single sample. From Tables A & B, ZeroSiam achieves superior performance across nearly all cases, *e.g.*, 51.6% (Ours) *vs.* 43.9% (DeYO) on ResNet50-GN under label shifts, and 51.3% (Ours) *vs.* 42.3% (COME) on ViT-Small under the batch size of 1. Similar results are observed even when the data stream exhibits both label shifts and data scarcity, as shown in Table C, suggesting our effectiveness across test settings.

## D.2   DETAILED RESULTS FOR TTA ON THE BLIND-SPOT SUBSET

From Table D, prior methods tend to collapse when adapting on the blind-spot subset, *e.g.*, 18.9% (DeYO) *vs.* 30.6% (NoAdapt) w.r.t. the average accuracy on R50-GN. In contrast, even under such a challenging setting, ZeroSiam achieves consistent improvement across all domains and models, suggesting that ZeroSiam substantially expands the scope and reliability of TTA in the real world.

## D.3   DETAILED RESULTS UNDER THE MILD TEST SCENARIO

ZeroSiam is not only effective under challenging test scenarios, but also enhances TTA performance significantly on the mild test scenario (Wang et al., 2021), where data comes with shuffled labels. As shown in Table E, ZeroSiam demonstrates superior performance on 4 out of 5 models and also achieves comparative results on ViT-Base, *e.g.*, the average accuracy of 48.6% (Ours) *vs.* 41.7% (DeYO) on ResNet50-GN, and 62.6% (Ours) *vs.* 62.8% (DeYO) on ViT-Base. This suggests that our ZeroSiam's design helps improve the stability and efficacy of TTA across a wide range of scenarios.

Table A: Detailed results of TTA under **IMBALANCED LABEL SHIFTS**, Table 3 in the main paper.

| Model+Method | Noise | | | Blur | | | | Weather | | | | Digital | | | | Avg. |
|---|---|---|---|---|---|---|---|---|---|---|---|---|---|---|---|---|
| | Gauss. | Shot | Impul. | Defoc. | Glass | Motion | Zoom | Snow | Frost | Fog | Brit. | Contr. | Elastic | Pixel | JPEG | |
| ResNet50-GN | 18.0 | 19.8 | 17.9 | 19.8 | 11.4 | 21.4 | 24.9 | 40.4 | 47.3 | 33.6 | 69.3 | 36.3 | 18.6 | 28.4 | 52.3 | 30.6 |
| • Tent | 2.6 | 3.3 | 2.7 | 13.9 | 7.9 | 19.5 | 17.0 | 16.5 | 21.9 | 1.8 | 70.5 | 42.2 | 6.6 | 49.4 | 53.7 | 22.0 |
| • SAR | 33.1 | 36.5 | 35.5 | 19.2 | 19.5 | 33.3 | 27.7 | 23.9 | 45.3 | 50.1 | 71.9 | 46.7 | 7.1 | 52.1 | 56.3 | 37.2 |
| • EATA | 27.0 | 28.3 | 28.1 | 14.9 | 17.1 | 24.4 | 25.3 | 32.2 | 32.0 | 39.8 | 66.7 | 33.6 | 24.5 | 41.9 | 38.4 | 31.6 |
| • COME | 17.7 | 19.7 | 17.7 | 19.7 | 11.2 | 21.2 | 24.8 | 40.2 | 47.8 | 33.7 | 68.9 | 35.6 | 18.6 | 27.7 | 51.8 | 30.4 |
| • DeYO | 42.5 | 44.9 | 43.8 | 22.2 | 16.3 | 41.0 | 13.2 | 52.2 | 51.5 | 39.7 | 73.4 | 52.6 | 46.9 | 59.3 | 59.3 | 43.9 |
| • ZeroSiam (ours) | 42.9 | 45.1 | 43.1 | 35.1 | 36.1 | 44.5 | 49.2 | 58.1 | 55.1 | 62.9 | 72.3 | 54.8 | 52.5 | 61.7 | 60.1 | **51.6** |
| ViT-Base | 9.4 | 6.7 | 8.3 | 29.1 | 23.4 | 34.0 | 27.0 | 15.8 | 26.3 | 47.4 | 54.7 | 43.9 | 30.5 | 44.5 | 47.6 | 29.9 |
| • Tent | 32.7 | 1.4 | 34.6 | 54.4 | 52.3 | 58.2 | 52.2 | 7.7 | 12.0 | 69.3 | 76.1 | 66.1 | 56.7 | 69.4 | 66.4 | 47.3 |
| • SAR | 46.5 | 43.1 | 48.9 | 55.3 | 54.3 | 58.9 | 54.8 | 53.6 | 46.2 | 69.7 | 76.2 | 66.2 | 60.9 | 69.6 | 66.6 | 58.0 |
| • EATA | 35.9 | 34.6 | 36.7 | 45.3 | 47.2 | 49.3 | 47.7 | 56.5 | 55.4 | 62.2 | 72.2 | 21.7 | 56.2 | 64.7 | 63.7 | 50.0 |
| • COME | 45.3 | 52.1 | 52.0 | 56.1 | 57.4 | 62.5 | 60.7 | 66.6 | 64.0 | 71.6 | 77.1 | 65.9 | 61.9 | 72.9 | 69.2 | 62.4 |
| • DeYO | 53.5 | 36.0 | 54.6 | 57.6 | 58.7 | 63.7 | 46.2 | 67.6 | 66.0 | 73.2 | 77.9 | 66.7 | 69.0 | 73.5 | 70.3 | 62.3 |
| • ZeroSiam (ours) | 52.3 | 52.6 | 53.4 | 57.7 | 58.7 | 62.7 | 61.1 | 67.6 | 66.0 | 73.3 | 78.0 | 67.0 | 67.9 | 73.5 | 70.1 | **64.1** |
| ViT-Small | 2.0 | 2.0 | 1.5 | 24.2 | 17.1 | 30.3 | 22.0 | 9.5 | 19.2 | 37.9 | 44.4 | 30.1 | 24.8 | 38.2 | 41.0 | 22.9 |
| • Tent | 0.3 | 0.4 | 0.2 | 43.1 | 37.0 | 45.4 | 36.3 | 5.5 | 24.2 | 54.8 | 65.8 | 54.6 | 26.6 | 56.6 | 54.9 | 34.0 |
| • SAR | 1.6 | 2.0 | 1.3 | 44.3 | 39.0 | 46.5 | 39.0 | 16.6 | 45.9 | 58.6 | 65.8 | 54.7 | 44.4 | 56.9 | 55.2 | 38.1 |
| • EATA | 15.6 | 15.5 | 17.8 | 42.4 | 40.3 | 44.8 | 41.6 | 46.2 | 48.7 | 58.9 | 65.3 | 52.9 | 50.5 | 57.2 | 56.2 | 43.6 |
| • COME | 0.1 | 0.2 | 0.1 | 47.4 | 47.2 | 52.8 | 10.5 | 35.5 | 53.8 | 63.9 | 70.7 | 58.1 | 56.7 | 63.4 | 60.2 | 41.4 |
| • DeYO | 0.1 | 0.2 | 0.1 | 48.4 | 47.7 | 53.4 | 16.2 | 47.0 | 54.6 | 65.0 | 71.0 | 59.3 | 58.5 | 64.1 | 61.0 | 43.1 |
| • ZeroSiam (ours) | 25.7 | 25.9 | 27.9 | 48.2 | 48.3 | 53.5 | 51.5 | 55.7 | 55.6 | 65.4 | 71.2 | 58.4 | 59.5 | 64.2 | 61.2 | **51.5** |
| ConvNeXt-Tiny | 21.4 | 27.7 | 22.1 | 24.5 | 11.0 | 32.5 | 31.2 | 44.5 | 52.5 | 39.5 | 72.0 | 44.8 | 23.1 | 17.7 | 55.8 | 34.7 |
| • Tent | 18.1 | 23.8 | 26.6 | 24.3 | 4.2 | 33.9 | 29.2 | 41.6 | 46.3 | 39.7 | 73.2 | 51.6 | 18.4 | 10.7 | 56.9 | 33.2 |
| • SAR | 34.8 | 36.7 | 34.9 | 27.1 | 4.1 | 35.0 | 30.8 | 44.3 | 47.6 | 8.4 | 73.1 | 51.1 | 19.4 | 24.7 | 56.7 | 35.2 |
| • EATA | 34.4 | 36.8 | 35.0 | 26.7 | 19.1 | 37.0 | 33.3 | 47.7 | 48.4 | 50.6 | 73.0 | 51.9 | 28.7 | 31.4 | 56.9 | 40.7 |
| • COME | 42.6 | 44.5 | 42.4 | 38.7 | 1.2 | 46.0 | 11.8 | 57.1 | 56.9 | 61.8 | 75.6 | 60.6 | 5.3 | 51.9 | 61.9 | 43.9 |
| • DeYO | 33.2 | 38.0 | 36.7 | 27.9 | 2.6 | 36.4 | 9.2 | 52.7 | 45.3 | 52.3 | 74.4 | 56.6 | 4.7 | 19.5 | 58.4 | 33.2 |
| • ZeroSiam (ours) | 42.3 | 44.5 | 42.1 | 37.4 | 33.5 | 45.3 | 44.0 | 56.5 | 55.4 | 62.9 | 74.7 | 59.4 | 44.3 | 51.7 | 60.0 | **50.3** |
| Swin-Tiny | 26.6 | 27.9 | 22.8 | 21.7 | 13.2 | 26.5 | 25.4 | 41.0 | 45.8 | 47.7 | 67.6 | 39.1 | 22.6 | 8.4 | 33.3 | 31.3 |
| • Tent | 16.1 | 14.1 | 10.5 | 17.1 | 16.9 | 23.5 | 20.0 | 30.4 | 28.3 | 11.3 | 69.7 | 47.8 | 9.3 | 1.1 | 43.2 | 24.0 |
| • SAR | 29.8 | 28.0 | 29.8 | 19.8 | 10.6 | 24.4 | 21.2 | 33.1 | 34.3 | 22.3 | 67.9 | 45.9 | 13.2 | 4.8 | 42.9 | 28.5 |
| • EATA | 36.2 | 37.1 | 33.6 | 26.3 | 29.8 | 38.2 | 37.3 | 47.3 | 44.5 | 51.4 | 70.3 | 42.4 | 40.9 | 34.8 | 46.5 | 41.1 |
| • COME | 41.8 | 43.6 | 43.2 | 36.5 | 9.8 | 46.7 | 2.4 | 54.6 | 52.3 | 60.0 | 72.0 | 47.4 | 4.3 | 3.1 | 55.8 | 38.2 |
| • DeYO | 35.3 | 38.5 | 35.6 | 18.9 | 24.6 | 37.8 | 9.4 | 43.7 | 41.8 | 52.3 | 70.6 | 50.7 | 21.5 | 15.6 | 50.8 | 36.5 |
| • ZeroSiam (ours) | 44.4 | 45.8 | 45.7 | 36.4 | 38.3 | 47.0 | 47.0 | 54.3 | 51.9 | 58.7 | 72.4 | 54.2 | 52.7 | 51.8 | 55.5 | **50.4** |

Table B: Detailed results of TTA with **BATCH SIZE=1**, *i.e.*, Table 4 in the main paper.

| | Noise | | | Blur | | | | Weather | | | | Digital | | | | |
|---|---|---|---|---|---|---|---|---|---|---|---|---|---|---|---|---|
| Model+Method | Gauss. | Shot | Impul. | Defoc. | Glass | Motion | Zoom | Snow | Frost | Fog | Brit. | Contr. | Elastic | Pixel | JPEG | Avg. |
| ResNet50-GN | 18.0 | 19.8 | 17.9 | 19.8 | 11.4 | 21.4 | 24.9 | 40.4 | 47.3 | 33.6 | 69.3 | 36.3 | 18.6 | 28.4 | 52.3 | 30.6 |
| • Tent | 2.5 | 2.9 | 2.5 | 13.5 | 3.6 | 18.6 | 17.6 | 15.3 | 23.0 | 1.4 | 70.4 | 42.2 | 6.2 | 49.2 | 53.8 | 21.5 |
| • SAR | 23.4 | 26.6 | 23.0 | 18.4 | 15.4 | 28.4 | 30.4 | 44.9 | 44.7 | 25.7 | 72.3 | 44.5 | 14.8 | 47.0 | 56.1 | 34.5 |
| • EATA | 24.8 | 28.3 | 25.7 | 18.1 | 17.3 | 28.5 | 29.3 | 44.5 | 44.3 | 41.6 | 70.9 | 44.6 | 27.0 | 46.8 | 55.7 | 36.5 |
| • COME | 17.8 | 19.7 | 17.7 | 19.7 | 11.3 | 21.4 | 24.9 | 40.3 | 47.6 | 33.6 | 68.9 | 35.7 | 18.5 | 27.9 | 51.8 | 30.5 |
| • DeYO | 41.8 | 44.7 | 43.0 | 22.5 | 24.7 | 41.8 | 24.4 | 54.5 | 52.2 | 20.7 | 73.5 | 53.5 | 48.5 | 60.2 | 59.8 | 44.4 |
| • ZeroSiam (ours) | 41.7 | 44.4 | 41.8 | 32.8 | 35.8 | 45.2 | 49.5 | 58.4 | 55.5 | 63.8 | 73.7 | 54.8 | 55.1 | 63.6 | 61.0 | **51.8** |
| ViT-Base | 9.5 | 6.7 | 8.2 | 29.0 | 23.4 | 33.9 | 27.1 | 15.9 | 26.5 | 47.2 | 54.7 | 44.1 | 30.5 | 44.5 | 47.8 | 29.9 |
| • Tent | 42.2 | 1.0 | 43.3 | 52.4 | 48.2 | 55.5 | 50.5 | 16.5 | 16.9 | 66.4 | 74.9 | 64.7 | 51.6 | 67.0 | 64.3 | 47.7 |
| • SAR | 40.8 | 36.4 | 41.5 | 53.7 | 50.7 | 57.5 | 52.8 | 59.1 | 50.7 | 68.1 | 74.6 | 65.7 | 57.9 | 68.9 | 65.9 | 56.3 |
| • EATA | 29.7 | 25.1 | 34.6 | 44.7 | 39.2 | 48.3 | 42.4 | 37.5 | 45.9 | 60.0 | 65.9 | 61.2 | 46.4 | 58.2 | 59.6 | 46.6 |
| • COME | 51.7 | 51.4 | 52.1 | 57.6 | 58.2 | 63.3 | 41.2 | 67.1 | 64.8 | 72.8 | 77.7 | 68.1 | 67.5 | 73.0 | 69.9 | 62.4 |
| • DeYO | 54.0 | 52.1 | 55.1 | 58.8 | 59.5 | 64.2 | 53.5 | 68.2 | 66.4 | 73.7 | 78.3 | 68.2 | 68.9 | 73.8 | 70.8 | **64.4** |
| • ZeroSiam (ours) | 52.1 | 52.7 | 52.8 | 57.8 | 58.3 | 63.0 | 60.6 | 67.0 | 65.7 | 73.0 | 77.9 | 67.6 | 67.9 | 72.8 | 69.9 | 63.9 |
| ViT-Small | 2.0 | 1.9 | 1.5 | 24.3 | 17.1 | 30.2 | 21.8 | 9.5 | 19.3 | 37.9 | 44.4 | 29.9 | 24.9 | 38.2 | 41.3 | 22.9 |
| • Tent | 21.3 | 27.7 | 22.0 | 24.5 | 10.6 | 32.3 | 30.9 | 44.6 | 52.4 | 39.6 | 72.3 | 44.5 | 23.4 | 17.6 | 55.6 | 34.6 |
| • SAR | 26.9 | 27.7 | 23.0 | 21.7 | 13.2 | 26.4 | 25.3 | 41.2 | 45.8 | 47.7 | 67.8 | 39.2 | 22.5 | 8.4 | 33.2 | 31.3 |
| • EATA | 2.4 | 2.5 | 1.8 | 32.8 | 25.4 | 36.3 | 28.2 | 17.1 | 28.2 | 48.7 | 53.5 | 46.1 | 34.1 | 46.7 | 47.7 | 30.1 |
| • COME | 0.5 | 3.5 | 0.4 | 48.4 | 47.3 | 52.9 | 15.4 | 38.2 | 53.8 | 64.3 | 70.7 | 58.8 | 55.8 | 63.6 | 60.4 | 42.3 |
| • DeYO | 0.4 | 0.8 | 0.3 | 49.0 | 47.4 | 53.0 | 20.1 | 46.4 | 54.6 | 64.7 | 70.9 | 59.3 | 57.3 | 64.1 | 61.0 | 43.3 |
| • ZeroSiam (ours) | 26.0 | 26.9 | 28.3 | 48.6 | 48.2 | 52.7 | 50.7 | 54.4 | 55.1 | 65.0 | 71.0 | 59.1 | 58.6 | 63.4 | 60.8 | **51.3** |
| ConvNeXt-Tiny | 21.3 | 27.7 | 22.0 | 24.5 | 10.6 | 32.3 | 30.9 | 44.6 | 52.4 | 39.6 | 72.3 | 44.5 | 23.4 | 17.6 | 55.6 | 34.6 |
| • Tent | 30.3 | 33.5 | 32.0 | 24.7 | 7.3 | 33.6 | 30.0 | 43.7 | 47.3 | 43.7 | 72.9 | 49.7 | 21.4 | 18.5 | 56.3 | 36.3 |
| • SAR | 30.3 | 33.6 | 29.8 | 25.9 | 11.1 | 33.4 | 30.8 | 44.9 | 47.8 | 38.3 | 72.9 | 46.2 | 22.6 | 18.1 | 56.2 | 36.1 |
| • EATA | 26.6 | 30.5 | 26.9 | 25.2 | 11.4 | 33.1 | 31.0 | 44.8 | 50.1 | 40.0 | 72.2 | 47.8 | 23.5 | 18.5 | 55.8 | 35.8 |
| • COME | 41.1 | 43.2 | 41.1 | 37.3 | 2.1 | 44.5 | 17.9 | 56.4 | 55.8 | 60.7 | 75.2 | 59.6 | 8.6 | 44.7 | 61.1 | 43.3 |
| • DeYO | 36.8 | 39.4 | 37.0 | 29.5 | 2.9 | 38.0 | 14.6 | 51.6 | 46.9 | 4.5 | 74.0 | 55.9 | 7.8 | 19.5 | 57.9 | 34.4 |
| • ZeroSiam (ours) | 41.5 | 43.6 | 41.5 | 37.2 | 32.5 | 44.8 | 42.7 | 55.6 | 54.7 | 62.2 | 74.3 | 59.0 | 42.3 | 50.5 | 59.3 | **49.4** |
| Swin-Tiny | 26.9 | 27.7 | 23.0 | 21.7 | 13.2 | 26.4 | 25.3 | 41.2 | 45.8 | 47.7 | 67.8 | 39.2 | 22.5 | 8.4 | 33.2 | 31.3 |
| • Tent | 23.8 | 26.2 | 22.5 | 19.4 | 18.4 | 26.9 | 27.3 | 40.4 | 36.0 | 22.8 | 69.2 | 46.5 | 16.5 | 2.4 | 41.4 | 29.3 |
| • SAR | 25.4 | 27.1 | 23.0 | 21.6 | 14.8 | 27.3 | 27.5 | 38.9 | 37.5 | 44.1 | 67.7 | 38.4 | 18.9 | 8.3 | 39.5 | 30.7 |
| • EATA | 32.8 | 34.5 | 31.7 | 23.4 | 17.5 | 32.0 | 28.2 | 43.5 | 43.6 | 45.4 | 69.7 | 45.8 | 27.2 | 10.5 | 39.6 | 35.0 |
| • COME | 38.6 | 41.4 | 39.0 | 34.4 | 14.9 | 44.6 | 7.3 | 52.9 | 51.0 | 56.9 | 72.4 | 51.7 | 15.7 | 33.8 | 53.3 | 40.5 |
| • DeYO | 35.5 | 38.5 | 35.3 | 22.0 | 23.9 | 37.1 | 15.3 | 42.4 | 40.4 | 47.6 | 70.2 | 50.8 | 23.7 | 18.0 | 48.8 | 36.6 |
| • ZeroSiam (ours) | 44.0 | 45.4 | 45.2 | 34.7 | 39.0 | 46.3 | 46.0 | 53.9 | 51.3 | 57.8 | 72.1 | 55.0 | 51.7 | 50.3 | 54.6 | **49.8** |

Table C: Additional results of TTA under **LABEL SHIFTS** with **BATCH SIZE=1** w.r.t. **Accuracy(%)**.

| | Noise | | | Blur | | | | Weather | | | | Digital | | | | |
|---|---|---|---|---|---|---|---|---|---|---|---|---|---|---|---|---|
| Model+Method | Gauss. | Shot | Impul. | Defoc. | Glass | Motion | Zoom | Snow | Frost | Fog | Brit. | Contr. | Elastic | Pixel | JPEG | Avg. |
| ResNet50-GN | 18.0 | 19.8 | 17.9 | 19.8 | 11.4 | 21.4 | 24.9 | 40.4 | 47.3 | 33.6 | 69.3 | 36.3 | 18.6 | 28.4 | 52.3 | 30.6 |
| • Tent | 1.2 | 1.5 | 1.3 | 10.0 | 2.1 | 14.3 | 11.2 | 8.2 | 11.7 | 0.8 | 70.1 | 41.6 | 3.4 | 49.5 | 52.3 | 18.6 |
| • SAR | 23.4 | 26.5 | 23.8 | 18.3 | 15.4 | 28.4 | 29.5 | 44.3 | 44.5 | 31.6 | 72.3 | 44.5 | 14.9 | 46.8 | 56.1 | 34.7 |
| • EATA | 19.2 | 21.7 | 19.4 | 17.9 | 13.0 | 23.5 | 25.7 | 39.7 | 43.6 | 34.5 | 69.4 | 38.4 | 20.0 | 34.9 | 53.2 | 31.6 |
| • COME | 17.8 | 19.7 | 17.8 | 19.7 | 11.2 | 21.3 | 24.9 | 40.3 | 47.7 | 33.6 | 69.0 | 35.7 | 18.6 | 27.9 | 51.7 | 30.5 |
| • DeYO | 41.0 | 43.8 | 42.2 | 22.9 | 23.3 | 41.4 | 15.9 | 54.0 | 52.3 | 20.7 | 73.4 | 53.6 | 48.1 | 60.1 | 59.8 | 43.5 |
| • ZeroSiam (ours) | 40.3 | 42.4 | 40.5 | 32.6 | 33.2 | 41.2 | 44.3 | 53.6 | 53.1 | 59.5 | 72.7 | 52.1 | 46.8 | 58.7 | 58.7 | **48.6** |
| ViT-Base | 9.4 | 6.7 | 8.3 | 29.1 | 23.4 | 34.0 | 27.0 | 15.8 | 26.3 | 47.4 | 54.7 | 43.9 | 30.5 | 44.5 | 47.6 | 29.9 |
| • Tent | 29.2 | 7.9 | 21.1 | 49.3 | 46.9 | 53.7 | 47.5 | 18.0 | 19.2 | 63.7 | 71.1 | 61.4 | 51.8 | 63.6 | 62.2 | 44.4 |
| • SAR | 24.8 | 18.3 | 24.1 | 38.2 | 31.9 | 42.1 | 35.5 | 31.1 | 37.5 | 52.3 | 61.0 | 52.3 | 36.3 | 52.0 | 52.4 | 39.3 |
| • EATA | 30.7 | 26.8 | 31.3 | 42.9 | 39.5 | 47.8 | 35.6 | 38.0 | 43.7 | 60.2 | 65.8 | 57.7 | 46.9 | 59.4 | 58.0 | 45.6 |
| • COME | 51.1 | 47.9 | 52.5 | 57.2 | 58.0 | 63.1 | 60.5 | 67.3 | 64.8 | 72.8 | 77.6 | 67.6 | 67.4 | 73.4 | 69.8 | **63.4** |
| • DeYO | 53.2 | 36.2 | 54.5 | 58.2 | 59.3 | 64.3 | 41.8 | 68.6 | 66.7 | 73.8 | 78.3 | 67.4 | 69.3 | 74.1 | 71.0 | 62.4 |
| • ZeroSiam (ours) | 49.9 | 50.2 | 51.0 | 56.0 | 56.4 | 60.8 | 58.4 | 65.5 | 64.4 | 71.9 | 77.4 | 66.1 | 65.9 | 71.5 | 68.7 | 62.3 |
| ViT-Small | 2.0 | 2.0 | 1.5 | 24.2 | 17.1 | 30.3 | 22.0 | 9.5 | 19.2 | 37.9 | 44.4 | 30.1 | 24.8 | 38.2 | 41.0 | 22.9 |
| • Tent | 0.3 | 0.3 | 0.2 | 43.4 | 37.2 | 45.7 | 36.6 | 4.6 | 22.3 | 58.4 | 66.0 | 54.7 | 27.0 | 56.8 | 55.3 | 33.9 |
| • SAR | 2.2 | 2.3 | 1.7 | 34.5 | 26.0 | 39.2 | 27.4 | 19.7 | 32.1 | 45.0 | 53.7 | 43.4 | 31.9 | 44.2 | 45.9 | 29.9 |
| • EATA | 2.1 | 2.3 | 1.7 | 29.1 | 21.8 | 34.6 | 25.0 | 14.7 | 24.8 | 42.8 | 49.5 | 39.4 | 28.7 | 42.7 | 44.2 | 26.9 |
| • COME | 0.5 | 3.5 | 0.4 | 47.8 | 46.9 | 52.7 | 16.9 | 37.3 | 53.6 | 64.3 | 70.6 | 58.8 | 55.8 | 63.5 | 60.4 | 42.2 |
| • DeYO | 0.4 | 0.8 | 0.3 | 48.7 | 47.3 | 53.2 | 21.2 | 46.6 | 54.2 | 64.9 | 70.9 | 59.4 | 57.5 | 64.1 | 61.0 | 43.4 |
| • ZeroSiam (ours) | 23.7 | 23.6 | 25.0 | 46.0 | 44.9 | 49.6 | 46.8 | 50.8 | 52.8 | 62.7 | 69.1 | 56.8 | 54.8 | 61.1 | 58.7 | **48.4** |
| ConvNeXt-Tiny | 21.4 | 27.7 | 22.1 | 24.5 | 11.0 | 32.5 | 31.2 | 44.5 | 52.5 | 39.5 | 72.0 | 44.8 | 23.1 | 17.7 | 55.8 | 34.7 |
| • Tent | 18.2 | 24.8 | 26.0 | 24.4 | 4.3 | 34.0 | 29.1 | 41.8 | 46.3 | 42.4 | 73.2 | 51.8 | 18.2 | 10.4 | 56.9 | 33.5 |
| • SAR | 30.2 | 33.6 | 30.2 | 25.8 | 11.1 | 33.3 | 31.0 | 44.9 | 47.6 | 37.8 | 73.0 | 46.2 | 22.7 | 18.2 | 56.3 | 36.1 |
| • EATA | 24.0 | 28.9 | 24.5 | 24.7 | 11.2 | 32.5 | 31.0 | 44.6 | 51.4 | 39.7 | 72.1 | 45.9 | 23.2 | 18.0 | 55.8 | 35.2 |
| • COME | 41.4 | 43.4 | 41.3 | 37.3 | 2.4 | 44.7 | 18.4 | 56.4 | 55.7 | 61.1 | 75.3 | 59.6 | 11.0 | 44.3 | 61.2 | 43.6 |
| • DeYO | 36.9 | 39.4 | 37.0 | 29.2 | 3.2 | 37.7 | 14.8 | 51.6 | 47.6 | 5.4 | 74.0 | 55.8 | 8.3 | 23.3 | 58.0 | 34.8 |
| • ZeroSiam (ours) | 41.3 | 43.3 | 41.1 | 36.3 | 31.4 | 43.8 | 42.2 | 54.9 | 53.7 | 61.5 | 74.2 | 58.2 | 41.0 | 49.8 | 59.2 | **48.8** |
| Swin-Tiny | 26.6 | 27.9 | 22.8 | 21.7 | 13.2 | 26.5 | 25.4 | 41.0 | 45.8 | 47.7 | 67.6 | 39.1 | 22.6 | 8.4 | 33.3 | 31.3 |
| • Tent | 14.6 | 13.1 | 11.1 | 17.4 | 14.3 | 21.2 | 20.9 | 31.4 | 24.2 | 12.1 | 69.7 | 48.1 | 9.1 | 1.0 | 42.7 | 23.4 |
| • SAR | 30.5 | 32.2 | 29.2 | 23.0 | 17.1 | 29.9 | 27.9 | 41.9 | 39.9 | 47.5 | 68.6 | 40.4 | 21.5 | 8.9 | 40.4 | 33.3 |
| • EATA | 27.7 | 29.1 | 25.0 | 21.7 | 14.2 | 27.7 | 25.9 | 40.7 | 43.5 | 47.4 | 68.1 | 39.9 | 23.4 | 9.1 | 34.3 | 31.9 |
| • COME | 38.6 | 41.0 | 39.2 | 34.3 | 21.6 | 44.6 | 5.0 | 53.1 | 51.1 | 57.2 | 72.2 | 51.5 | 21.3 | 18.3 | 53.4 | 40.2 |
| • DeYO | 35.7 | 37.6 | 35.8 | 21.0 | 26.0 | 36.8 | 14.7 | 42.3 | 42.0 | 49.6 | 70.4 | 50.9 | 24.4 | 26.2 | 48.8 | 37.5 |
| • ZeroSiam (ours) | 42.8 | 44.4 | 43.7 | 33.3 | 36.1 | 44.2 | 43.9 | 52.1 | 49.6 | 55.9 | 71.6 | 53.5 | 49.3 | 47.7 | 53.4 | **48.1** |

Table D: Detailed results of TTA on the **BLIND-SPOT SUBSET**, *i.e.*, Table 6 in the main paper.

| Model+Method | Noise | | | Blur | | | | Weather | | | | Digital | | | | Avg. |
|---|---|---|---|---|---|---|---|---|---|---|---|---|---|---|---|---|
| | Gauss. | Shot | Impul. | Defoc. | Glass | Motion | Zoom | Snow | Frost | Fog | Brit. | Contr. | Elastic | Pixel | JPEG | |
| ResNet50-GN | 18.0 | 19.8 | 17.9 | 19.8 | 11.4 | 21.4 | 24.9 | 40.4 | 47.3 | 33.6 | 69.3 | 36.3 | 18.6 | 28.4 | 52.3 | 30.6 |
| ● Tent | 0.2 | 0.2 | 0.2 | 10.5 | 2.3 | 5.8 | 2.3 | 1.6 | 2.3 | 0.2 | 63.3 | 16.7 | 0.5 | 48.3 | 46.8 | 13.4 |
| ● SAR | 17.0 | 19.6 | 16.9 | 15.8 | 11.4 | 22.2 | 24.0 | 5.8 | 42.0 | 25.4 | 69.0 | 38.9 | 1.3 | 31.9 | 54.2 | 26.4 |
| ● EATA | 18.6 | 21.2 | 18.6 | 15.6 | 13.4 | 22.7 | 24.4 | 37.1 | 38.8 | 33.2 | 67.8 | 41.6 | 19.1 | 47.1 | 53.0 | 31.5 |
| ● COME | 18.0 | 19.8 | 17.9 | 19.8 | 11.3 | 21.4 | 24.9 | 40.4 | 47.3 | 33.6 | 69.2 | 36.2 | 18.6 | 28.3 | 52.2 | 30.6 |
| ● DeYO | 0.4 | 0.6 | 0.4 | 7.9 | 0.2 | 28.1 | 1.6 | 5.5 | 7.1 | 0.1 | 71.2 | 47.8 | 1.3 | 59.7 | 51.4 | 18.9 |
| ● ZeroSiam (ours) | 38.0 | 42.7 | 39.0 | 27.9 | 34.0 | 41.1 | 43.3 | 52.1 | 44.3 | 59.1 | 60.2 | 51.2 | 53.8 | 60.3 | 45.4 | **46.2** |
| ViT-Base | 9.5 | 6.7 | 8.2 | 29.0 | 23.4 | 33.9 | 27.1 | 15.9 | 26.5 | 47.2 | 54.7 | 44.1 | 30.5 | 44.5 | 47.8 | 29.9 |
| ● Tent | 0.2 | 0.2 | 0.1 | 56.5 | 54.0 | 60.4 | 0.5 | 1.4 | 1.0 | 0.2 | 77.5 | 67.4 | 0.3 | 71.6 | 68.7 | 30.7 |
| ● SAR | 42.2 | 7.1 | 43.9 | 55.3 | 51.3 | 59.4 | 54.0 | 19.7 | 47.6 | 71.5 | 77.4 | 67.1 | 19.4 | 71.3 | 67.8 | 50.3 |
| ● EATA | 26.4 | 20.3 | 32.2 | 45.7 | 37.5 | 48.6 | 42.3 | 37.2 | 47.5 | 62.2 | 65.7 | 63.1 | 46.8 | 60.1 | 59.8 | 46.4 |
| ● COME | 0.1 | 0.1 | 0.1 | 56.9 | 58.4 | 63.5 | 0.5 | 68.3 | 64.7 | 73.4 | 77.1 | 67.1 | 69.1 | 73.4 | 69.5 | 49.5 |
| ● DeYO | 0.1 | 0.3 | 0.2 | 59.4 | 60.4 | 65.1 | 3.8 | 69.7 | 67.6 | 74.5 | 78.5 | 68.7 | 70.8 | 75.0 | 71.2 | 51.0 |
| ● ZeroSiam (ours) | 52.6 | 53.2 | 53.4 | 57.7 | 58.6 | 63.4 | 61.7 | 67.8 | 66.5 | 73.8 | 78.2 | 67.8 | 69.2 | 73.7 | 70.2 | **64.5** |
| ViT-Small | 2.0 | 1.9 | 1.5 | 24.3 | 17.1 | 30.2 | 21.8 | 9.5 | 19.3 | 37.9 | 44.4 | 29.9 | 24.9 | 38.2 | 41.3 | 22.9 |
| ● Tent | 0.1 | 0.2 | 0.1 | 44.6 | 36.6 | 45.6 | 33.2 | 1.3 | 16.5 | 59.5 | 68.0 | 56.5 | 0.5 | 58.2 | 56.2 | 31.8 |
| ● SAR | 2.1 | 2.2 | 1.6 | 31.1 | 23.7 | 39.2 | 27.5 | 9.8 | 40.0 | 51.8 | 66.4 | 46.7 | 23.4 | 53.2 | 49.8 | 31.2 |
| ● EATA | 2.2 | 2.2 | 1.6 | 28.4 | 20.3 | 33.2 | 24.2 | 11.8 | 23.0 | 44.8 | 51.4 | 47.4 | 28.3 | 45.4 | 45.6 | 27.3 |
| ● COME | 0.1 | 2.6 | 0.1 | 47.3 | 48.0 | 52.4 | 0.2 | 31.2 | 55.6 | 65.0 | 71.7 | 59.5 | 0.4 | 64.4 | 60.6 | 37.3 |
| ● DeYO | 0.1 | 0.3 | 0.1 | 49.4 | 48.9 | 54.0 | 0.5 | 45.9 | 56.3 | 66.2 | 72.3 | 60.6 | 58.6 | 65.5 | 61.9 | 42.7 |
| ● ZeroSiam (ours) | 25.1 | 25.4 | 27.6 | 48.2 | 47.9 | 52.5 | 50.6 | 54.3 | 55.5 | 65.6 | 71.8 | 59.2 | 59.4 | 64.6 | 61.2 | **51.3** |
| ConvNeXt-Tiny | 21.3 | 27.7 | 22.0 | 24.5 | 10.6 | 32.3 | 30.9 | 44.6 | 52.4 | 39.6 | 72.3 | 44.5 | 23.4 | 17.6 | 55.6 | 34.6 |
| ● Tent | 15.5 | 12.9 | 25.2 | 21.3 | 2.1 | 30.0 | 25.3 | 30.3 | 40.4 | 1.8 | 72.6 | 50.1 | 14.3 | 3.2 | 56.4 | 26.8 |
| ● SAR | 24.3 | 30.2 | 24.8 | 24.7 | 10.7 | 32.5 | 30.6 | 42.9 | 50.1 | 37.1 | 72.7 | 45.1 | 21.8 | 17.7 | 55.7 | 34.7 |
| ● EATA | 23.7 | 29.1 | 24.5 | 24.6 | 11.1 | 32.5 | 30.7 | 44.3 | 50.4 | 39.9 | 72.0 | 46.2 | 23.2 | 17.9 | 55.7 | 35.0 |
| ● COME | 41.1 | 42.7 | 40.4 | 29.4 | 0.4 | 33.3 | 2.4 | 25.1 | 24.9 | 0.2 | 74.2 | 57.7 | 0.6 | 27.6 | 58.3 | 30.6 |
| ● DeYO | 34.9 | 35.5 | 35.3 | 22.0 | 0.9 | 23.0 | 3.0 | 38.6 | 29.1 | 0.3 | 72.1 | 52.3 | 1.8 | 22.1 | 55.6 | 28.4 |
| ● ZeroSiam (ours) | 42.4 | 44.9 | 42.5 | 37.2 | 33.9 | 45.5 | 44.5 | 56.3 | 53.3 | 64.4 | 73.7 | 60.5 | 44.7 | 56.3 | 58.5 | **50.6** |
| Swin-Tiny | 26.9 | 27.7 | 23.0 | 21.7 | 13.2 | 26.4 | 25.3 | 41.2 | 45.8 | 47.7 | 67.8 | 39.2 | 22.5 | 8.4 | 33.2 | 31.3 |
| ● Tent | 0.2 | 0.4 | 0.1 | 14.0 | 5.9 | 7.2 | 5.0 | 9.6 | 8.9 | 2.6 | 67.7 | 16.1 | 1.4 | 0.3 | 34.4 | 11.6 |
| ● SAR | 27.3 | 29.0 | 24.8 | 21.6 | 14.9 | 26.7 | 25.7 | 36.6 | 45.3 | 47.5 | 67.8 | 39.4 | 10.5 | 8.6 | 36.5 | 30.8 |
| ● EATA | 27.9 | 29.0 | 25.3 | 20.7 | 14.6 | 28.4 | 25.7 | 40.1 | 40.7 | 47.4 | 68.1 | 40.8 | 23.6 | 9.2 | 33.8 | 31.7 |
| ● COME | 38.1 | 41.3 | 42.0 | 26.2 | 0.5 | 10.8 | 0.5 | 10.3 | 8.0 | 10.2 | 72.8 | 52.0 | 0.1 | 0.5 | 55.1 | 24.6 |
| ● DeYO | 0.2 | 0.1 | 0.2 | 13.8 | 6.4 | 11.4 | 3.7 | 11.6 | 16.9 | 4.3 | 70.2 | 44.4 | 3.7 | 0.4 | 47.1 | 15.6 |
| ● ZeroSiam (ours) | 45.1 | 47.5 | 47.0 | 35.2 | 40.3 | 46.1 | 46.9 | 56.1 | 51.9 | 58.5 | 68.6 | 54.8 | 54.7 | 52.9 | 56.3 | **50.8** |

Table E: Detailed results of TTA under the **MILD SCENARIO**, *i.e.*, Table 7 in the main paper.

| Model+Method | Noise | | | Blur | | | | Weather | | | | Digital | | | | Avg. |
|---|---|---|---|---|---|---|---|---|---|---|---|---|---|---|---|---|
| | Gauss. | Shot | Impul. | Defoc. | Glass | Motion | Zoom | Snow | Frost | Fog | Brit. | Contr. | Elastic | Pixel | JPEG | |
| ResNet50-GN | 18.0 | 19.8 | 17.9 | 19.8 | 11.4 | 21.4 | 24.9 | 40.4 | 47.3 | 33.6 | 69.3 | 36.3 | 18.6 | 28.4 | 52.3 | 30.6 |
| ● Tent | 5.1 | 6.1 | 5.4 | 15.0 | 10.8 | 22.1 | 22.9 | 25.8 | 33.2 | 3.1 | 70.4 | 42.7 | 11.0 | 48.1 | 54.2 | 25.1 |
| ● SAR | 28.5 | 31.4 | 29.7 | 18.7 | 19.5 | 30.2 | 30.1 | 43.3 | 43.7 | 7.0 | 70.8 | 44.0 | 19.0 | 48.8 | 55.2 | 34.7 |
| ● EATA | 37.3 | 39.1 | 39.4 | 28.1 | 27.1 | 36.8 | 39.1 | 50.8 | 49.0 | 55.8 | 72.0 | 50.1 | 41.8 | 55.7 | 58.1 | 45.4 |
| ● COME | 17.9 | 19.7 | 17.7 | 19.7 | 11.3 | 21.4 | 24.9 | 40.4 | 47.5 | 33.6 | 69.1 | 35.9 | 18.6 | 27.9 | 52.1 | 30.5 |
| ● DeYO | 39.6 | 42.1 | 40.6 | 22.2 | 23.3 | 38.3 | 37.7 | 50.4 | 49.4 | 3.0 | 73.0 | 50.3 | 41.9 | 55.5 | 57.7 | 41.7 |
| ● ZeroSiam (ours) | 40.5 | 42.4 | 41.7 | 31.4 | 31.8 | 40.8 | 44.4 | 53.5 | 53.3 | 59.1 | 72.9 | 51.5 | 47.4 | 58.9 | 58.8 | **48.6** |
| ViT-Base | 9.4 | 6.7 | 8.3 | 29.1 | 23.4 | 34.0 | 27.0 | 15.8 | 26.3 | 47.4 | 54.7 | 43.9 | 30.5 | 44.5 | 47.6 | 29.9 |
| ● Tent | 42.4 | 1.4 | 43.3 | 52.2 | 47.6 | 55.4 | 49.9 | 19.7 | 18.1 | 66.1 | 74.9 | 64.7 | 52.5 | 66.8 | 64.1 | 47.9 |
| ● SAR | 44.3 | 14.0 | 45.7 | 52.9 | 49.8 | 56.0 | 51.1 | 58.3 | 50.6 | 66.5 | 74.6 | 64.3 | 55.5 | 66.5 | 64.0 | 54.3 |
| ● EATA | 49.7 | 49.4 | 51.3 | 55.9 | 55.5 | 60.5 | 57.8 | 63.4 | 63.0 | 70.4 | 76.0 | 67.0 | 64.7 | 69.3 | 67.9 | 61.5 |
| ● COME | 50.9 | 51.6 | 52.1 | 56.9 | 57.4 | 62.2 | 59.2 | 66.1 | 64.3 | 72.5 | 77.4 | 67.9 | 66.3 | 72.4 | 69.2 | **63.1** |
| ● DeYO | 52.9 | 53.1 | 53.8 | 58.2 | 58.6 | 63.0 | 40.2 | 67.4 | 65.7 | 73.2 | 78.0 | 68.0 | 67.7 | 73.1 | 69.8 | 62.8 |
| ● ZeroSiam (ours) | 50.5 | 51.0 | 51.5 | 57.1 | 56.8 | 61.2 | 58.6 | 64.9 | 64.5 | 71.9 | 77.2 | 67.6 | 65.6 | 71.5 | 68.5 | 62.6 |
| ViT-Small | 2.0 | 2.0 | 1.5 | 24.2 | 17.1 | 30.3 | 22.0 | 9.5 | 19.2 | 37.9 | 44.4 | 30.1 | 24.8 | 38.2 | 41.0 | 22.9 |
| ● Tent | 0.5 | 0.6 | 0.4 | 39.5 | 33.0 | 42.5 | 34.0 | 10.3 | 37.1 | 54.5 | 63.6 | 51.7 | 31.5 | 53.1 | 52.3 | 33.6 |
| ● SAR | 1.4 | 2.1 | 1.1 | 40.8 | 34.9 | 43.2 | 35.5 | 16.3 | 41.8 | 54.7 | 63.7 | 51.8 | 38.5 | 53.4 | 52.5 | 35.4 |
| ● EATA | 22.4 | 20.6 | 23.6 | 45.0 | 43.0 | 47.2 | 43.5 | 48.5 | 49.9 | 60.7 | 66.4 | 56.6 | 52.7 | 58.3 | 57.2 | 46.4 |
| ● COME | 0.2 | 0.4 | 0.2 | 47.7 | 46.3 | 51.3 | 19.6 | 37.9 | 52.7 | 63.6 | 69.8 | 58.3 | 54.1 | 61.8 | 59.2 | 41.6 |
| ● DeYO | 0.2 | 0.3 | 0.2 | 47.9 | 45.8 | 51.4 | 33.6 | 44.5 | 53.0 | 63.5 | 70.1 | 58.5 | 55.3 | 62.4 | 59.8 | 43.1 |
| ● ZeroSiam (ours) | 23.8 | 23.4 | 25.2 | 47.6 | 46.0 | 50.6 | 48.1 | 51.9 | 53.3 | 63.2 | 69.9 | 58.1 | 55.7 | 61.8 | 59.4 | **49.2** |
| ConvNext-Tiny | 21.4 | 27.7 | 22.1 | 24.5 | 11.0 | 32.5 | 31.2 | 44.5 | 52.5 | 39.5 | 72.0 | 44.8 | 23.1 | 17.7 | 55.8 | 34.7 |
| ● Tent | 30.8 | 33.6 | 32.0 | 24.7 | 7.5 | 33.6 | 30.0 | 43.7 | 47.3 | 43.6 | 72.9 | 49.5 | 21.5 | 19.3 | 56.3 | 36.4 |
| ● SAR | 33.4 | 35.6 | 33.7 | 27.1 | 8.1 | 34.1 | 30.7 | 44.6 | 47.3 | 17.0 | 72.8 | 48.8 | 22.0 | 24.4 | 56.1 | 35.7 |
| ● EATA | 35.3 | 37.4 | 35.8 | 31.0 | 20.7 | 37.1 | 33.7 | 48.5 | 49.1 | 51.3 | 73.2 | 53.0 | 29.5 | 31.3 | 56.6 | 41.6 |
| ● COME | 40.4 | 42.6 | 40.4 | 36.2 | 2.3 | 43.2 | 23.4 | 54.6 | 54.2 | 58.1 | 75.1 | 58.6 | 10.6 | 44.7 | 59.9 | 43.0 |
| ● DeYO | 36.3 | 39.0 | 36.4 | 29.1 | 2.7 | 37.1 | 19.6 | 50.1 | 48.3 | 4.8 | 73.8 | 54.9 | 8.6 | 23.7 | 57.4 | 34.8 |
| ● ZeroSiam (ours) | 39.6 | 41.7 | 39.3 | 33.8 | 27.3 | 42.0 | 38.7 | 53.2 | 51.8 | 59.3 | 73.8 | 57.0 | 36.5 | 44.2 | 58.2 | **46.4** |
| Swin-Tiny | 26.6 | 27.9 | 22.8 | 21.7 | 13.2 | 26.5 | 25.4 | 41.0 | 45.8 | 47.7 | 67.6 | 39.1 | 22.6 | 8.4 | 33.3 | 31.3 |
| ● Tent | 23.7 | 26.2 | 22.7 | 19.4 | 18.6 | 27.1 | 27.4 | 40.5 | 36.3 | 24.4 | 69.2 | 46.3 | 16.9 | 2.8 | 41.5 | 29.5 |
| ● SAR | 31.8 | 32.8 | 32.5 | 22.3 | 20.2 | 29.6 | 28.4 | 42.6 | 39.8 | 37.6 | 69.2 | 46.8 | 19.9 | 6.0 | 43.2 | 33.5 |
| ● EATA | 40.9 | 42.9 | 41.6 | 33.5 | 33.6 | 42.6 | 40.1 | 49.1 | 54.9 | 54.9 | 71.8 | 52.6 | 44.8 | 40.0 | 51.1 | 46.0 |
| ● COME | 41.2 | 43.2 | 42.1 | 35.2 | 15.0 | 44.7 | 7.0 | 52.9 | 51.9 | 57.8 | 72.1 | 51.1 | 23.8 | 25.2 | 54.3 | 41.2 |
| ● DeYO | 35.2 | 37.0 | 36.0 | 22.2 | 25.0 | 35.8 | 15.4 | 42.0 | 43.8 | 48.0 | 70.1 | 50.3 | 31.6 | 34.5 | 48.0 | 38.3 |
| ● ZeroSiam (ours) | 42.4 | 44.0 | 43.0 | 31.7 | 34.9 | 43.6 | 42.6 | 51.3 | 48.9 | 54.7 | 71.5 | 53.4 | 47.5 | 45.8 | 52.3 | **47.2** |

## E  ADDITIONAL DISCUSSIONS

### E.1  ZEROSIAM'S EFFICACY BEYOND COLLAPSE PREVENTION

ZeroSiam regularize test-time entropy minimization from favoring non-generalizable shortcuts (c.f. Section 3.2), thereby improving its performance even in cases when no collapse occurs. As shown in Figure B, Tent drastically reduces entropy and eventually closes it to zero by inflating the logit norms and aligning all logits toward a dominant mode. Despite the reduction of entropy, it does not involve meaningful learning, and Tent thus fails to improve the accuracy on the first 250 batches of samples, where such biased signals can further degrade performance during a prolonged adaptation, as shown in Figure B. In contrast, ZeroSiam maintains a stable logit norm and center dominance ratio during entropy minimization, which successfully enhances TTA efficacy beyond collapse prevention, as shown in Figure B (a). Moreover, from Figure B (d), ZeroSiam does not greedily minimize the entropy loss. Instead, ZeroSiam converges to a low but non-zero entropy loss, while still enabling consistent accuracy improvement, *i.e.*, from 250 to 750 batches as in Figure B (a). Such convergence toward non-zero loss is also aligned with Theory 1, which shows that ZeroSiam inherently defines a lower bound of $h_{min}$ for entropy optimization, preventing the model from degenerating into collapsed constant one-hot outputs that trivially minimize (*i.e.*, zero out) the entropy loss.

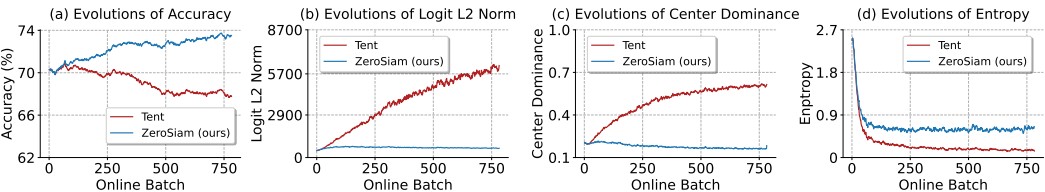

Figure B: Empirical evidence of ZeroSiam for boosting stability and efficacy. (a-d) record the ODD accuracy, logits $L_2$ norm, center dominance, and entropy in model predictions under a mild test scenario (Wang et al., 2021). Experiments are run on ImageNet-C (Bright, level 5) with ViT-Base.

### E.2  EVOLUTIONS OF DIVERGENCE LOSS IN ZEROSIAM

We provide additional results of how the divergence loss evolves during adaptation to supplement Figure 2. As shown in Figure C, the divergence loss increases more rapidly and substantially under a more imbalanced stream. Such a phenomenon is also aligned with changes of the predictor as shown in Figure 2 (a), where a larger divergence from an identity mapping results in a larger similarity loss. Overall, these results suggest *an adaptive regularization strength* according to the degree of collapse risk in the scenario, indicating the efficacy of our asymmetric entropy optimization design.

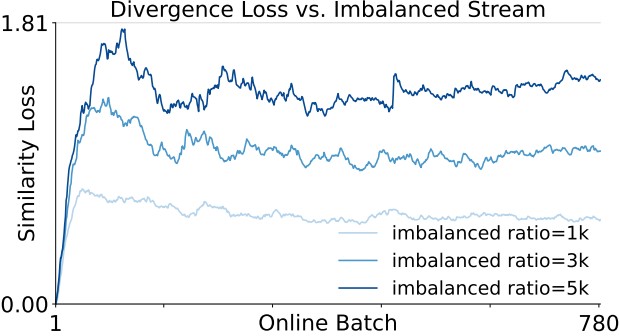

Figure C: The evolution of divergence loss during TTA . Results are reported on ImageNet-C (Snow, level 5) with ResNet50-GN under online streams with varying imbalanced ratios (Niu et al., 2023).

### E.3    ZeroSiam for Saving a Collapsed Model

Interestingly, we further demonstrate that ZeroSiam can sometimes save a model that has already collapsed. To illustrate this, we first run Tent on an imbalanced data stream until collapse occurs, and then apply ZeroSiam. As shown in Figure D, ZeroSiam helps restore strong accuracy after 300 batches of adaptation. This recovery is driven by the asymmetric alignment in ZeroSiam, which makes collapsed solutions no longer a stable minimum, encouraging the model to escape the collapsed mode and re-cluster samples. However, this does not fully explain how class-wise clusters that are consistent with pre-training re-emerge. We hypothesize this is because only the affine parameters in the normalization layers are updated during testing, which introduces a small adjustment to the model that makes performance recovery possible. Notably, we observe that the predictor in ZeroSiam has to be randomly initialized (*e.g.*, $\theta_h = \mathbf{I} + 0.1\,\mathbf{W}$, $\mathbf{W} \sim \mathcal{N}(0, \mathbf{I})$) so as to derive from the identity and also remain learnable during TTA to enable the collapse recovering effects. Even so, recovery succeeds in only 4 out of 7 domains, suggesting that while promising, collapse recovery with ZeroSiam remains an open question, and we leave it for future work.

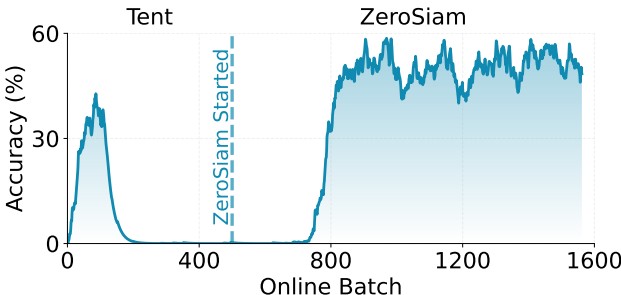

Figure D: Efficacy of ZeroSiam for saving a collapsed model. Results are reported on ImageNet-C (Shot, level 5) with ViT-Base under **ONLINE IMBALANCED LABEL SHIFTS** (imbalance ratio = $\infty$).

### E.4    Comparisons between the Entropy and Divergence Term during TTA

As shown in Figure E, the alignment loss $D(p^o\|\mathrm{sg}[p^r])$ does not grow infinitely. Instead, the alignment loss quickly increases then converges, while the entropy term $H(p^o)$ decreases and stabilizes. Throughout adaptation, the alignment term exhibits a smaller loss value than the entropy term, which provides an appropriate regularization strength that prevents the risk of collapse, without hindering the effectiveness of the entropy-based adaptation.

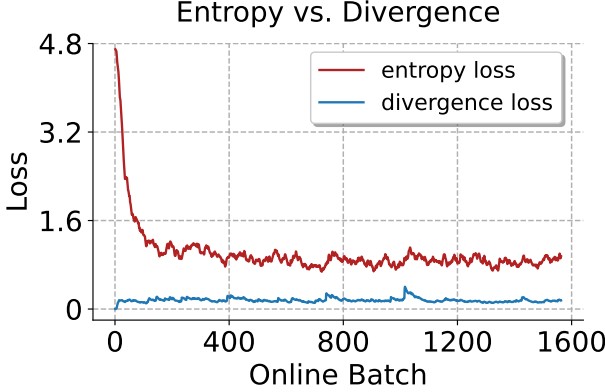

Figure E: Comparisons between entropy and divergence during TTA on ImageNet-C (Gaussian, level 5) under **ONLINE IMBALANCED LABEL SHIFTS** (imbalance ratio = $\infty$) with ViT-Base.

### E.5 MORE SENSITIVITY ANALYSIS OF LEARNING RATES IN ZEROSIAM

To demonstrate ZeroSiam's robustness, we further supplement the sensitivity analysis of learning rates on Swin-Tiny, a challenging architecture on which Tent frequently suffers from collapse.

From Figure F, ZeroSiam consistently outperforms Tent across a wide range of encoder learning rates, confirming the stability trends observed in Figure 4 across architectures. Moreover, on smaller and more collapse-prone models, ZeroSiam can benefit from a larger predictor learning rate $\eta_h$, $e.g.$, yielding an accuracy of 53.3% with $\eta_f = 10 \times 5e-5$ and $\eta_h = 40 \times \eta_f$; while still outperforming previous methods across broad learning rate configurations, $e.g.$, achieving an accuracy above 45% when $\eta_f \in [5, 20] \times 5e-5$ and $\eta_h \in [1, 40] \times \eta_f$, showing a substantial gain compared to the accuracy of 38.2% in COME and 37.7% in DeYO. These results collectively demonstrate the efficacy of ZeroSiam across learning rate choices. In practice, we did not extensively tune the learning rates, where we simply use $\eta_f = 10 \times 5e-5$ and $\eta_h = 5 \times \eta_f$ on ZeroSiam, which yields an accuracy of 50.4% for comparisons with other methods.

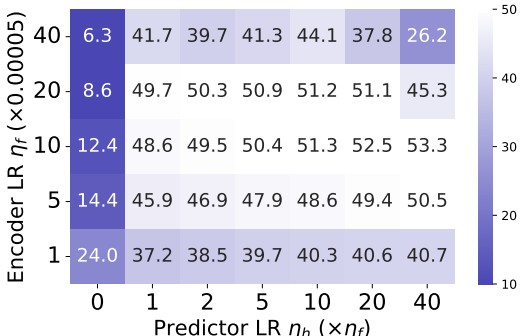

Figure F: Sensitivity to learning rates on Swin-Tiny. Results are reported on ImageNet-C (Gaussian, level 5) under **ONLINE IMBALANCED LABEL SHIFTS** (imbalance ratio = $\infty$).

### E.6 DISCUSSIONS ON A *Data-Free* APPROACH FOR SELECTING $\eta_h$ IN ZEROSIAM

While the predictor lr $\eta_h$ in ZeroSiam is simply set the same as $\eta_f$ on larger models and around $10\times$ larger than $\eta_f$ on smaller and more collapse-prone models, we further briefly introduce **a data-free method** to help select $\eta_h$. The core idea is to leverage random Gaussian noise as the proxy to track TTA dynamics, where ideally, noisy learning signals should be totally absorbed and suppressed in ZeroSiam. To this end, we track the changes in prediction, $e.g.$, prediction entropy $\Delta_e$, before and after adaptation on Gaussian noise, and select the best-performing $\eta_h$ that leads to a minimal change, $e.g.$, $\min_{\eta_h} |\Delta_e|$, which implies an appropriate regularization strength for bias filtration. As shown in Table V, on ResNet50-GN, we can find $\eta_h = 10$ by $\min_{\eta_h} |\Delta_e|$ using only generated Gaussian before TTA deployment, which also delivers the highest TTA accuracy of 51.4% under label shift.

Table F: Illustration of data-free selection for $\eta_h$ on ResNet50-GN based on entropy difference $|\Delta_e|$ before and after adaptation on 6,400 random Gaussian inputs. TTA accuracy is measured under the label shift setting. Encoder learning rate $\eta_f$ is set to 0.00125, and $\eta_h$ is set to $k \times 0.00125$.

| Metric | $\eta_h = 0$ | $\eta_h = 2$ | $\eta_h = 5$ | $\eta_h = 10$ | $\eta_h = 20$ | $\eta_h = 40$ |
|---|---|---|---|---|---|---|
| prediction $|\Delta_e|$ | 0.750 | 0.085 | 0.053 | 0.001 | 0.061 | 0.083 |
| TTA Acc. (%) | 14.3 | 49.6 | 51.0 | 51.4 | 50.3 | 45.3 |

We acknowledge that some architectures, such as ConvNeXt, can consistently output the highest prediction entropy on Gaussian noise regardless of adaptation and may require tracking other prediction metrics for data-free $\eta_h$ selection. We leave this for future work, and believe that our current focus on collapse prevention and asymmetric entropy-based architecture already makes meaningful contributions, as this significantly reduces the risk of TTA across adaptation scenarios while maintaining efficiency for practical and reliable deployment.

### E.7 EFFICACY OF ZEROSIAM ACROSS DIVERSE ADAPTATION EPOCHS

In our main experiments, all methods are evaluated with only 1 epoch over the test set, following the evaluation settings established by prior works. Here, we further evaluate Tent and ZeroSiam using more adaptation epochs under Tent's default mild setting. As shown in Table G, Tent's performance gradually degrades with more epochs (47.9% → 41.3%), reflecting its known overfitting tendency. In contrast, ZeroSiam remains stable and even benefits moderately from additional epochs (62.6% → 65.7%). This confirms that ZeroSiam improves generalization while mitigating overfitting in Tent, which maintains a stable adaptation across different epoch budgets.

Table G: Accuracy of Tent and ZeroSiam under the $k$-th epoch of TTA over the entire test set. Results are reported with ViT-Base under Tent's default mild setting.

| Method | No adapt | Tent (k=1) | Tent (k=3) | Tent (k=5) | ZeroSiam (k=1) | ZeroSiam (k=3) | ZeroSiam (k=5) |
|---|---|---|---|---|---|---|---|
| Acc. (%) | 29.9 | 47.9 | 45.5 | 41.3 | 62.6 | 65.7 | 65.7 |

### E.8 NECESSITY OF STOP-GRADIENT FOR ASYMMETRY

As verified in Theory 1, the online branch in ZeroSiam converges more rapidly towards the collapse solutions during TTA. The stop-gradient on the target branch prevents the target branch from also drifting toward the collapse modes, while providing a stable reference for the online branch. As shown in Table H, removing stop-gradient leads to severe collapse, *e.g.*, reducing the accuracy from 51.6%→20.5% on ResNet50-GN. Similar results are also observed in SimSiam (Chen & He, 2021) and BYOL (Grill et al., 2020), underscoring stop-gradient as a key component in asymmetry.

Table H: Importance of stop-gradient operation. Results are reported on ImageNet-C (severity level 5) under **ONLINE IMBALANCED LABEL SHIFTS** (imbalance ratio = ∞) regarding **Accuracy (%)**.

| | Noise | | | Blur | | | | Weather | | | | Digital | | | | |
|---|---|---|---|---|---|---|---|---|---|---|---|---|---|---|---|---|
| ResNet50-GN | Gauss. | Shot | Impul. | Defoc. | Glass | Motion | Zoom | Snow | Frost | Fog | Brit. | Contr. | Elastic | Pixel | JPEG | Avg. |
| ZeroSiam (ours) | 42.9 | 45.1 | 43.1 | 35.1 | 36.1 | 44.5 | 49.2 | 58.1 | 55.1 | 62.9 | 72.3 | 54.8 | 52.5 | 61.7 | 60.1 | 51.6 |
| - w/o stop-grad | 2.1 | 2.7 | 2.0 | 15.3 | 2.0 | 15.2 | 11.1 | 11.9 | 17.7 | 1.2 | 71.9 | 43.5 | 3.8 | 50.9 | 55.5 | 20.5 |
| ViT-Base | Gauss. | Shot | Impul. | Defoc. | Glass | Motion | Zoom | Snow | Frost | Fog | Brit. | Contr. | Elastic | Pixel | JPEG | Avg. |
| ZeroSiam (ours) | 52.3 | 52.6 | 53.4 | 57.7 | 58.7 | 62.7 | 61.1 | 67.6 | 66.0 | 73.3 | 78.0 | 67.0 | 67.9 | 73.5 | 70.1 | 64.1 |
| - w/o stop-grad | 2.8 | 0.8 | 12.8 | 54.1 | 51.1 | 57.6 | 33.7 | 6.3 | 8.0 | 68.7 | 76.1 | 65.9 | 4.4 | 69.2 | 66.3 | 38.5 |

### E.9 STATISTICAL COMPARISON

We re-run Table 3 in the main paper with 5 different random seeds. As shown in Table I, ZeroSiam achieves both higher adaptation accuracy and lower standard deviation compared to prior methods. Specifically, ZeroSiam increases the average accuracy from 42.6% (DeYO) to 52.9%, while reducing the standard deviation by over *10*-fold. This reduced standard deviation further underscores our reliability for robust adaptation in real-world applications.

Table I: Statistical comparison. Experiments follow the settings of Table 3 in the main paper. Results are reported with 5 random seeds.

| Method | ResNet50-GN | ViT-Base | ViT-Small | ConvNeXt-Tiny | Swin-Tiny | Average |
|---|---|---|---|---|---|---|
| DeYO | $39.7_{\pm2.05}$ | $61.8_{\pm2.19}$ | $40.8_{\pm0.36}$ | $34.3_{\pm0.40}$ | $36.5_{\pm0.71}$ | $42.6_{\pm0.82}$ |
| ZeroSiam | $51.2_{\pm0.06}$ | $63.5_{\pm0.10}$ | $50.0_{\pm0.06}$ | $49.8_{\pm0.11}$ | $50.2_{\pm0.16}$ | $52.9_{\pm0.06}$ |

### E.10 MORE DISCUSSIONS ON THE IMPACTS OF UNCONTROLLED LOGIT NORM INFLATION

During test-time entropy minimization, uncontrolled inflation of logit norm can induce the following issues that hinders robust and reliable model adaptation: 1) *Overconfidence*: regarding individual predictions, increasing the logit magnitude $||u||_2$ (where $u$ is the predicted logit) artificially

inflates predictive confidence without improving accuracy, since the predicted class $\arg\max_k u_k$ depends solely on the direction $u' = \frac{u}{||u||_2}$, not its magnitude. This leads to overly confident but inaccurate predictions that undermine trust in AI systems. 2) *Degraded generalization from gradient interference*: while inflating the logit norm does not benefit a sample's own generalization, it adversely affects others through gradient interference during batch updates. Empirically, test-time adaptation with only the objective of $-||u||_2$ to inflate the logit norm would lead to a catastrophic collapse of zero accuracy for all models. Theoretically, define the decision margins as $m(x) = \max_k u_k - \max_{j \neq k} u_j$ and the sharpness of relative boundaries $\frac{||\nabla_x m(x)||}{m(x)}$, given that both $m(x)$ and $||\nabla_x m(x)||$ scale proportionally with $||u||_2$, inflating the logit norm can not improve the sample's own relative boundary. However, since per-example gradients are generally non-orthogonal, a sample's update can *add irrelevant noise to the boundaries of other samples while having no benefit on its current prediction*. Such interference accumulates during online TTA, adversely affecting generalization and ultimately causing collapse. ZeroSiam mitigates this by aligning with a stable target branch, preventing uncontrolled norm growth, and improving TTA efficacy.

### E.11 MORE DISCUSSIONS WITH MULTI-BRANCH ADAPTATION METHODS

ZeroSiam advances the entropy-based, self-training TTA by adding both the mechanisms of bias learning signals filtration (c.f. Section 3.2) and asymmetric optimization (c.f. Section 3.1 through a lightweight predictor with theoretical insights (Theorem 1). We further detail our distinctions with multi-branch-based TTA methods, such as SPA (Niu et al., 2025), REM (Han et al., 2025), and TTE (Kim et al., 2025) from three aspects.

(1) *Distinct inspirations from self-supervised learning (objective vs. architecture):* SPA and REM are augmentation-based methods that exploit consistency learning—an objective similar to self-supervised approaches—across different views to promote adaptation to the target domain. To this end, their efforts focus on devising appropriate augmentations or deteriorations on images for information masking and creating a weak-to-strong consistency learning. In contrast, ZeroSiam is entirely augmentation-free and focuses on the asymmetric architecture design—orthogonal to efforts in SPA and REM—for the single-branch, entropy-based adaptation, not being limited to images.

(2) *Advantages of model collapse prevention (model-, test data stream- & modality-agnostic):* REM prevents collapse by depending on the assumption that masked images should exhibit higher prediction entropy, which develops architecture-specific masking strategies for the vision transformer equipped with a classification token and explicitly fixes the rank of prediction entropy of masked inputs. However, the assumption and the masking strategy may fail to generalize to broader architectures and modalities, while the objective can still be minimized with constant one-hot outputs. TTE prevents collapse by subtracting a moving average center in the predicted logit for self-alignment, which explicitly reduces center dominance in a straightforward manner. However, it assumes that the prediction center truly reflects the trends of overfitting, which, however, violates under a long-tail data stream where the center actually reflects the dominant class. In this case, when subtracting the center, TTE can significantly risk flipping correct predictions and producing misguided alignment. In contrast, ZeroSiam develops an assumption-free and theoretically grounded asymmetric architecture that self-induces collapse-resistant alignment to disfavor shortcuts in entropy minimization without relying on heuristic thresholds, making it generally applicable across diverse models, data streams, and modalities (Tables 2-7) and requiring fewer hyperparameters for deployment.

(3) *Unique strengths of adaptive bias filtration (improving generalization):* Beyond the widely applicable collapse prevention mechanism in ZeroSiam compared to prior methods, we also empirically and theoretically demonstrate that ZeroSiam further improves generalization with a bias learning signals filtration mechanism during TTA. Specifically, we show that the predictor in the online branch purposely absorbs and converts biased shortcut signals (*e.g.*, norm inflation) into explicit discrepancies, where biased signals are then penalized by the alignment loss. As shown in Theorem 1 and Figure 2 (b-c), any shortcut representable by the predictor is naturally suppressed, while useful signals flow into the backbone encoder update. This bias filtration effect even helps improve the generalization of a large language model during test-time reasoning incentivization, demonstrating a promising research direction to enhance future TTA methods.

Finally, we provide further empirical comparisons on both TTA performance and efficiency. As shown in Table J, existing methods improve performance at a substantial cost of computation and

memory efficiency, due to using additional augmentations and branches, *e.g.*, SPA increases the memory of Tent from 7,760MB to 28,147MB (which can not be fitted into an RTX 3090), and the latency from 113s to 275s on ViT-Base. In contrast, ZeroSiam is the only method that maintains almost the same memory and latency as Tent, while achieving the highest stability, *e.g.*, increasing the accuracy of SPA from 60.3% to 64.5%, while reducing the memory from 28,147MB to 7,806MB on ViT-Base, which reveals a novel and practical stability–efficiency Pareto frontier that has not been achieved in prior TTA methods.

Table J: comparisons on performance and efficiency with multi-branch TTA methods. Accuracy is reported under the blind-spot adaptation scenario. Efficiency is measured by processing 50,000 images under a batch size of 64 via an A100 GPU.

| Method | ResNet50-GN | | | ViT-Base | | |
|---|---|---|---|---|---|---|
| | Acc. (%) | Mem. (MB) | Time (s) | Acc. (%) | Mem. (MB) | Time (s) |
| Tent | 13.4 | 5,519 | 89 | 30.7 | 7,760 | 113 |
| TTE | 40.8 | 6,303 | 136 | 55.2 | 8,574 | 164 |
| REM | - | - | - | 56.9 | 27,438 | 346 |
| SPA | 45.0 | 21,327 | 220 | 60.3 | 28,147 | 275 |
| ZeroSiam (ours) | 46.2 | 5,584 | 89 | 64.5 | 7,806 | 113 |

## F    MORE DISCUSSIONS WITH REINFORCEMENT LEARNING

Entropy collapse is also a common challenge within reinforcement learning, where policies can degenerate (*e.g.*, become deterministic and lose diversity) due to biased or noisy advantage estimation, insufficient exploration, or overly aggressive updates (Mnih et al., 2016; Haarnoja et al., 2018; Ahmed et al., 2019). To address this, trust-region methods such as TRPO (Schulman et al., 2015) and PPO (Schulman et al., 2017) constrain each update step in policy space through KL-divergence penalties or clipping, ensuring gradual and stable improvement. Similar principles appear in value-based methods (Mnih et al., 2015; Schwarzer et al., 2023; Nauman et al., 2024), where target networks provide a stable anchor to stabilize bootstrapped updates against rapidly changing estimates. These approaches share with ZeroSiam the key idea of structurally constraining updates to counter collapse. Specifically, ZeroSiam's online branch with gradient updates resembles a fast-updating policy (*i.e.*, with extra updates in the predictor), while the stop-gradient target branch serves as an anchor, effectively forming a lightweight trust region in the entropy minimization landscape. This connection situates ZeroSiam as a general collapse-prevention mechanism, linking stabilization strategies across TTA, self-supervised learning, and reinforcement learning.

## G    LARGE LANGUAGE MODEL USAGE STATEMENT

In accordance with the ICLR 2026 policy on the responsible use of LLMs, we confirm that our study did not use any LLM to generate scientific content or conduct substantive experiments. The only use of an LLM (ChatGPT-5) was to polish the English writing and improve presentation quality; all core methodology, experiments, and analyses were authored and verified by the human authors.

