# OpenReview forum: "ZeroSiam: An Efficient Asymmetry for Test-Time Entropy Optimization without Collapse"
_ICLR.cc/2026/Conference — ICLR 2026 Poster_

### Official Review · Reviewer_UrZX · 2025-10-27

**Soundness:** 3
**Presentation:** 2
**Contribution:** 3
**Rating:** 6
**Confidence:** 3

**Summary:**

Entropy minimization in test-time adaptation often leads collapse into one-hot outputs without learning meaningful features. To alleviate this issue, the paper proposes to employ asymmetric siamese architecture inspired by negative-free self-supervised learning such as BYOL. More specifically, it simply adds a shallow linear layers of which outputs are trained to minimize the divergence with the target branch designed to have no gradients. The effectiveness of the method was demonstrated with extensive experimental results on both image and natural language reasoning benchmark datasets.

**Strengths:**

1. The proposed method is simple yet effective.
2. It provides some theoretical support that shows convergence behavior.
3. It was validated on extensive experiments for both image and natural language datasets compared to several state-of-the-art methods.
4. Various ablation studies were provided.

**Weaknesses:**

1. It is not clear why it is beneficial to have large deviation from identity mapping and what exactly imbalance ratio is in Figure 2(a).
2. Why is the inflation of logit L2 norm bad (according to Observation 2) in Figure 2(c)? Is there any theoretical connection between increase of logit norm and generalization ability?
3. What exactly is center dominance in Figure 2(d)? How is it computed? How is it connected to generalization ability in the target domain?
4. It seems like the experiments were run with one random seed. It is important to have standard deviation to compare significance of the difference between different methods.

**Questions:**

1. How small is $L_f$ in practice? Wouldn’t it be arbitrary large for the worst case scenario? Also, I don’t quite understand the meaning of “Given that only the affine parameters of the normalization layers are optimized, this assumption is well justified.”
2. Does the theorem hold when we update only on one sample? Or does it hold for any arbitrary number of inputs with random order?

---

> ### Author Response · Authors · 2025-11-23
> **Responses to reviewer UrZX [1/3]**
>
> Thank you for taking the time to review our paper and for providing valuable feedback. We would like to address your questions below.
>
> > W1. It is not clear why it is beneficial to have large deviation from identity mapping and what exactly imbalance ratio is in Figure 2(a).
>
> A. Thank you for your thoughtful questions. Imbalance ratio quantifies how uneven the class frequencies are in the test stream—A higher imbalance ratio suggests a period contains much more samples from certain classes than others, and in the most difficult case, data streams with an infinite imbalance ratio indicate that test samples come in class order, _e.g._, samples all drawn from "Dog" in a period, and all from "Cat" in the next period. This imbalance ratio is mathematically formulated in SAR [1].
>
> Under such imbalanced streams, entropy minimization suffers from a higher risk of learning trivial collapsed outputs that assign all predictions to the dominant class. As shown in Figure 2(a), ZeroSiam helps stabilize test-time entropy minimization with an **adaptive regularization strength**, _i.e._, learning larger deviation from identity (posing a stronger alignment regularization in Eqn. (4) to stabilize entropy optimization) when the test stream is highly imbalanced, while learning smaller deviation when the test stream is balanced. This manifested property of adaptive regularization strength in ZeroSiam thus helps maintain a **reliable adaptation performance** across various, **unpredictable** test scenarios via a minimal architectural change.
>
> [1] Towards Stable Test-Time Adaptation in Dynamic Wild World. ICLR 2023.
>
> > W2. Why is the inflation of logit L2 norm bad (according to Observation 2) in Figure 2 \(c)? Is there any theoretical connection between increase of logit norm and generalization ability?
>
>
> A. Thank you for your questions. During TTA, inflating logit norm can induce the following issues that hinder robust and reliable model adaptation.
>
> 1) **Overconfidence**: regarding individual predictions, increasing the logit magnitude $||u||_2$ (where $u$ is the predicted logit) artificially inflates predictive confidence without improving accuracy, since the predicted class $\arg \max_k u_k$ depends solely on the direction $u'=\frac{u}{||u||_2}$, not its magnitude. This leads to overly confident but inaccurate predictions that undermine trust in AI systems.
> 2) **Degrading generalization from gradient interference**: while inflating logit norm does not benefit a sample’s own generalization, it adversely affects others through gradient interference during online learning. **Empirically**, test-time adaptation with only the objective of inflating logit norm would lead to a catastrophic collapse of zero accuracy for all models. **Theoretically**, define the decision margins as $m(x)=\max_ku_k-\max_{j\ne k}u_j$ and the sharpness of relative boundaries $\frac{|| \nabla_x m(x) ||}{m(x)}$, given that both $m(x)$ and $|| \nabla_x m(x) ||$ scale proportionally with $||u||_2$, inflating the logit norm can not improve the sample's own relative boundary. However, since per-example gradients are generally non-orthogonal, a sample's update can **add irrelevant noise to the boundaries of other samples while having no benefit on its current prediction**. Such gradient interference accumulates during online adaptation, adversely affecting generalization and ultimately causing collapse.
>
>
> ZeroSiam mitigates this by aligning with a stable target branch, preventing uncontrolled norm growth and improving adaptation efficacy.
>
>
> > W3. What exactly is center dominance in Figure 2(d)? How is it computed? How is it connected to generalization ability in the target domain?
>
> A. Center dominance is the ratio of the averaged predicted logit across samples, i.e., $\frac{\lVert \mathbb{E}[u] \rVert}{\mathbb{E}[\lVert u \rVert]}$ and $u$ is the vector of the normalized predicted logit, measuring the extent of non-discriminative components shared across predictions [2]. To elaborate, each normalized predicted logit $u$ is first decomposed into $u=\bar{u}+r$, where $\bar{u}=\mathbb{E}[u]$ is a shared center, and $r$ is the sample-specific discriminative component. Center dominance then quantifies the dominance of the non-discriminative center in predictions, computed by $\lVert \bar{u} \rVert / \lVert u \rVert$, where the normalized logits always give $\lVert u \rVert = 1$.
>
> In Figure 2(d), despite reducing prediction entropy through Tent, a rise in center dominance indicates that Tent amplifies the non-discriminative components in predictions, while the discriminative power on the target domain data decreases. In contrast, ZeroSiam demonstrates a descending center dominance through its bias filtration effect (*Conclusion 2* from Theorem 1), which results in better performance/generalization in the target domain.
>
> [2] A Unified Understanding with Self-supervised Contrastive Learning. ICLR 2022.

---

> ### Author Response · Authors · 2025-11-23
> **Responses to reviewer UrZX [2/3]**
>
> > W4. It seems like the experiments were run with one random seed. It is important to have standard deviation to compare significance of the difference between different methods.
>
> A. Thank you for your thoughtful suggestions. ZeroSiam achieves both higher adaptation accuracy and lower standard deviation compared to prior methods. As demonstrated in Table I, ZeroSiam increases the average accuracy from 42.6% (DeYO) to 52.9%, while reducing the standard deviation by **over 10-fold**. This reduced standard deviation further underscores its reliability for real-world applications.
>
>
> Table I. Statistical comparison. Experiments follow the settings of Table 3 in the main paper. Results are reported with 5 random seeds.
>
> |Method|ResNet50-GN|ViT-Base|ViT-Small|ConvNeXt-Tiny|Swin-Tiny|Avg.|
> |-|:-:|:-:|:-:|:-:|:-:|:-:|
> |DeYO|39.7$_{\pm 2.05}$|61.8$_{\pm 2.19}$|40.8$_{\pm 0.36}$|34.3$_{\pm 0.40}$|36.5$_{\pm 0.71}$|42.6$_{\pm 0.82}$|
> |ZeroSiam|51.2$_{\pm 0.06}$|63.5$_{\pm 0.10}$|50.0$_{\pm 0.06}$|49.8$_{\pm 0.11}$|50.2$_{\pm 0.16}$|52.9$_{\pm 0.06}$|
>
> > Q1. How small is $L_f$ in practice? Wouldn’t it be arbitrary large for the worst case scenario? Also, I don’t quite understand the meaning of “Given that only the affine parameters of the normalization layers are optimized, this assumption is well justified.”
>
>
> A. Thank you for your insightful questions. We would like to answer the question about $L_f$ and the assumption separately.
>
> Firstly, $L_f$ remains a reasonable range. $L_f$ is often interpreted as the norm of slope (gradient) for $f$, which can be understood from a Taylor expansion. Specifically, under a first-order Taylor expansion of a continuous function $f_{\theta+\Delta\theta}(\mathbf{x}) \approx f_{\theta}(\mathbf{x}) + \nabla_\theta f(x)\Delta\theta$, it implies that for $\theta_1=\theta$ and $\theta_2=\theta+\Delta\theta$, we can derive $\lVert f_{\theta_1}(\mathbf{x})-f_{\theta_2}(\mathbf{x})\rVert \approx \lVert \nabla_\theta f(x)\rVert \lVert \theta_1-\theta_2 \rVert \triangleq  L_{f}\lVert \theta_1-\theta_2 \rVert$, which shows that $L_f$ corresponds to the norm of gradient $\lVert \nabla_\theta f(x)\rVert$ and its value is determined by current inputs $\mathbf{x}$ and parameters $\theta$, thus providing a reasonable bound across diverese optimization cases and scenarios.
>
> Secondly, in our theoretical analysis, we assume that the encoder is Lipschitz continuous. Lipschitz continuity implies that we can find a gradient-related value to describe the changes between $f_{\theta_1}(\mathbf{x})$ and $f_{\theta_2}(\mathbf{x})$. Such an assumption is standard in convergence analyses of gradient-based optimization and is widely used in theoretical studies involving modern neural network architectures. In the original paper, we stated that this assumption is further justified by updating only affine parameters—a significantly smaller subset of the model parameters—because operating in a much lower-dimensional parameter space intuitively makes parameter variations more controllable and thus more compatible with Lipschitz continuity. However, we find and acknowledge that this argument is not strictly rigorous, and we have revised/deleted the text accordingly. We apologize for the confusion and sincerely appreciate the reviewer for pointing this out. Nevertheless, we would like to emphasize that this does not affect the validity of our theoretical results, as Lipschitz continuity is already a well-established and widely adopted assumption in network-based theoretical analyses.

---

> ### Author Response · Authors · 2025-11-23
> **Responses to reviewer UrZX [3/3]**
>
> > Q2. Does the theorem hold when we update only on one sample? Or does it hold for any arbitrary number of inputs with random order?
>
> A. Thank you for your thoughtful questions. Our theorem holds for an arbitrary number and order of inputs to characterize ZeroSiam's optimization behaviors under Eqn. (4).
>
> As detailed in Theorem 1, we conduct theoretical analysis for Eqn. (4) according to the _predictor state_, showing the online branch converges faster to the collapse region that creates a discrepancy regularization when predictor is an identity (c.f. $\alpha=0$ in Theorem 1), where such discrepancy helps filter biased signals and guides the system toward a stable, non-collapsing equilibrium (c.f. $\alpha>0$ in Theorem 1). While the number of inputs impacts whether the algorithm converges or not, it does not influence the correctness or applicability of the analysis.
>
> As also demonstrated empirically, ZeroSiam exhibits a much smaller standard deviation under different random seeds compared to prior methods (Table I in W4), as well as a consistent online accuracy enhancement when trained with different number of samples (Figure B (a) in the main paper), highlighting a robust model adaptation regardless of the order and number of inputs, a property that is crucial for reliable real-world deployments.
>
> We have revised Section 3.2, Appendix B.1, and included two additional sections (Appendix E.9 and E.10) to reflect the rebuttal in our paper. We sincerely hope our clarifications above have addressed your concerns and can improve your opinion of our work.

---

> > ### Comment · Reviewer_UrZX · 2025-11-27
> >
> > Thank you for the detailed clarification to my concerns. Many of them are resolved, and I will keep my current ratings.

---

> > > ### Author Response · Authors · 2025-11-28
> > > **Thank you for your positive feedback!**
> > >
> > > Dear Reviewer UrZX,
> > >
> > > Thank you very much for your positive feedback, for acknowledging our contributions, and for your invaluable comments on improving our paper. We are super glad to hear that our responses have successfully addressed your concerns.
> > >
> > > If you feel that our clarifications and revisions have strengthened the paper, we would be grateful if you could consider re-evaluating our submission and potentially upgrading your initial score. Your support means a great deal to us!
> > >
> > > Thank you again for your time and effort in reviewing our submission and your thoughtful feedback. We would be happy to continue the discussion if you have any further questions.
> > >
> > > Best regards,
> > >
> > > The Authors

---

### Official Review · Reviewer_32Pf · 2025-10-28

**Soundness:** 4
**Presentation:** 3
**Contribution:** 3
**Rating:** 8
**Confidence:** 5

**Summary:**

This paper proposes ZeroSiam, a method inspired by asymmetric self-supervised learning to address the collapse problem of entropy minimization-based approaches, where all predictions degenerate into a single one-hot vector regardless of the input. By introducing an asymmetric structure, ZeroSiam eliminates the need for additional augmentations and enhances efficiency through a single forward pass, effectively preventing model collapse.

**Strengths:**

- The paper presents extensive experimental results across a wide range of benchmarks. In particular, the experiments on the blind spot adaptation effectively demonstrate the limitations of entropy minimization methods when faced with incorrect initial predictions, thereby strengthening the motivation and necessity for the proposed method.
- The paper addresses an important issue in test-time adaptation by improving inference time through a single forward pass without any augmentation. The utilization of all samples and the reduction in inference time enhance the practicality of the method for real-world applications.
- The visualization of model collapse in entropy minimization (Figure 2) is clear, and the visualization of recovery possibility (Figure D in Sup. material) is intriguing and provides additional insight.

**Weaknesses:**

- Although the proposed method is shown to be applicable under various learning rates, the comparison may not be entirely fair since it also updates affine parameters similar to previos methods but applies a higher learning rate. For instance, in Table 5, it is unclear how the learning rate for LoRA was determined, as the optimal value for the proposed method could differ due to the influence of the projector module.
- While the efficiency achieved through a single forward pass is acknowledged, there are concerns regarding the novelty of the approach compared to prior works that also draw inspiration from self-supervised learning and apply it to TTA. For example,\
[1] Self-Bootstrapping for Versatile Test-Time Adaptation, ICML 2025\
[2] Ranked Entropy Minimization for Continual Test-Time Adaptation, ICML 2025\
[3] Test-Time Ensemble via Linear Mode Connectivity: A Path to Better Adaptation, ICLR 2025\
Furthermore, [1] employed the self-supervised learning and [2, 3] addressed the model collpase problem.
- Some typos should be revised. For example, u^r in eq(2) and u^o in eq (3) are not same to softmax(u_r) and softmax(u_o) (line 190), resepctively.

**Questions:**

- In Figure 4, the sensitivity analysis of the learning rate is conducted using ViT-Base, where collapse does not occur seriously, making it difficult to fairly assess the robustness of learning rates across different approaches.
- Moreover, different learning rates are applied to the predictor depending on the architecture, and the hyperparameters of existing methods are not fully optimized, which complicates fair comparison. If the predictor’s learning rate is treated as a tunable hyperparameter, wouldn’t it also be possible to alleviate the collapse problem in other entropy minimization methods by appropriately adjusting parameters such as the entropy threshold or learning rate?

---

> ### Author Response · Authors · 2025-11-23
> **Responses to reviewer 32Pf [1/4]**
>
> We thank you for your recognition of our work and positive feedback on further improving the quality of our paper. We would like to address your questions below.
>
> > W1. Although the proposed method is shown to be applicable under various learning rates, the comparison may not be entirely fair since it also updates the affine parameters similar to previous methods but applies a higher learning rate. For instance, in Table 5, it is unclear how the learning rate for LoRA was determined, as the optimal value for the proposed method could differ due to the influence of the projector module.
>
> A. Thank you for your valuable comments. **We strictly follow the *same baseline learning rates* whenever they are provided by the original papers (which were already tuned by the authors), and we also tested that they do not benefit from a larger learning rate.** For models without officially reported learning rates, we additionally tune them to ensure fair and near-optimal overall performance. As verified in Table I, previous methods could not consistently benefit or even suffer from degradation with larger learning rates, confirming that the baselines are already operating near optimal settings.
>
> **In contrast, ZeroSiam consistently benefits from larger learning rates.** This is primarily benefiting from its asymmetric architecture and alignment loss, which effectively absorb noisy signals and stabilize the update process, thereby permitting more aggressive learning with larger learning rates. This behavior is conceptually related to the motivations of EATA and DeYO, where reweighting mechanisms implicitly enlarge the effective update scale for reliable samples. However, their noise-suppression capabilities remain weaker than ZeroSiam, resulting in less consistent gains under larger learning rates.
>
> Table I. Effectiveness of previous methods with increased learning rates under label shifts. All methods use the base learning rates of 2.5e-5 for ConvNeXt-Tiny, and 5e-5 for Swin-Tiny.
>
> |Model|Tent  $(\mathrm{lr}\times 1)$|Tent $(\mathrm{lr}\times 2)$|EATA $(\mathrm{lr}\times 1)$|EATA $(\mathrm{lr}\times 2)$|EATA $(\mathrm{lr}\times 5)$|EATA $(\mathrm{lr}\times 10)$|SAR  $(\mathrm{lr}\times 1)$|SAR  $(\mathrm{lr}\times 2)$|SAR $(\mathrm{lr}\times 5)$|SAR $(\mathrm{lr}\times 10)$|DeYO $(\mathrm{lr}\times 1)$|DeYO $(\mathrm{lr}\times 2)$|ZeroSiam $(\mathrm{lr}\times 2)$|ZeroSiam $(\mathrm{lr}\times 5)$|ZeroSiam $(\mathrm{lr}\times 10)$|
> |-|:-:|:-:|:-:|:-:|:-:|:-:|:-:|:-:|:-:|:-:|:-:|:-:|:-:|:-:|:-:|
> |ViT_Small|34.0|33.0|43.6|42.3|36.4|19.4|38.1|39.6|40.8|40.6|43.4|43.4|46.1|50.0|51.5|
> |ConvNeXt_Tiny|33.2|27.9|40.7|42.5|43.9|40.7|35.2|34.4|34.7|32.8|33.9|32.7|42.8|47.0|50.3|
> |Swin_Tiny|24.0|18.5|41.1|40.1|32.1|23.1|28.5|30.7|28.7|27.9|37.7|35.3|43.4|47.9|50.4|

---

> ### Author Response · Authors · 2025-11-23
> **Responses to reviewer 32Pf [2/4]**
>
> > W2. While the efficiency achieved through a single forward pass is acknowledged, there are concerns regarding the novelty of the approach compared to prior works that also draw inspiration from self-supervised learning and apply it to TTA
>
> A. Thank you for your valuable comments. ZeroSiam advances the entropy-based, self-training TTA by adding both the mechanisms of bias learning signals filtration (c.f. Section 3.2) and asymmetric optimization (c.f. Section 3.1) through a lightweight predictor with theoretical insights (Theorem 1). In the following, we detail our distinctions with SPA [1], REM [2], and TTE [3] from three aspects.
>
> **(1) Distinct inspirations from self-supervision (objective *vs.* architecture):**
>
> **SPA and REM are augmentation-based methods that exploit consistency learning**—an **objective** similar to self-supervised approaches—across different views to promote adaptation to the target domain. To this end, their efforts focus on devising appropriate augmentations or deteriorations on images for information masking and creating a weak-to-strong consistency learning. In contrast, **ZeroSiam is entirely augmentation-free and focuses on the asymmetric architecture design—orthogonal to efforts in SPA and REM**—for the single-branch, entropy-based adaptation, while not being limited to images (Table 5).
>
> **(2) Advantages of model collapse prevention (model-, test data stream- & modality-agnostic):**
>
> REM prevents collapse by depending on the assumption that masked images should exhibit higher prediction entropy, which develops architecture-specific masking strategies for ViT with a classification token and explicitly fixes the rank of prediction entropy of masked inputs. However, the assumption and the masking strategy may fail to generalize to broader architectures and modalities, while the objective can still be minimized with constant one-hot outputs.
>
> TTE prevents collapse by subtracting a moving average center in the predicted logit for self-alignment, which straightforwardly reduces center dominance. However, it assumes that the prediction center truly reflects the trends of overfitting, which, however, violates under a long-tail data stream where the center actually reflects the dominant class. In this case, when subtracting the center, TTE can significantly risk flipping correct predictions and producing misguided alignment.
>
> In contrast, ZeroSiam develops an **assumption-free and theoretically grounded asymmetric architecture** (Theorem 1) that self-induces collapse-resistant alignment to disfavor shortcuts in entropy minimization without relying on heuristic thresholds, making it **generally applicable across diverse models, data streams, and modalities** (Tables 2-6) and requiring fewer hyperparameters for deployment.
>
> **(3) Unique strengths of adaptive bias filtration (improved generalization)**
>
> Beyond collapse prevention, we also empirically and theoretically demonstrate that **ZeroSiam further improves generalization with an unique biased learning signals filtration mechanism during TTA**. Specifically, we show that the predictor in the online branch purposely absorbs and converts biased shortcuts (e.g., norm inflation signals) into explicit discrepancies, where these biased signals are then penalized by the alignment loss. As shown in Theorem 1 and Figure 2(b-c), **any shortcut representable by the predictor is naturally suppressed, while useful signals flow into the backbone encoder update**. This bias filtration effect even helps improve the generalization of a large language model during test-time reasoning incentivization, demonstrating a promising research direction to enhance future TTA methods.
>
> Finally, we provide further empirical comparisons on both TTA performance and efficiency. As shown in Table II, existing methods improve performance at a substantial cost of computation and memory efficiency, due to using additional augmentations and branches, e.g., SPA increases the memory of Tent from 7,760MB to 28,147MB (can not be fitted into an RTX 3090), and the latency from 113s to 275s on ViT-B. In contrast, ZeroSiam is the only method that maintains nearly the same memory and latency as Tent, while achieving the highest stability, e.g., increasing the accuracy of SPA from 60.3% to 64.5%, while reducing the memory from 28,147MB to 7,806MB on ViT-B, which reveals a novel and practical stability–efficiency Pareto frontier that has not been achieved in prior TTA methods.
>
> Table II. comparisons on TTA performance and efficiency. Accuracy is reported under the blind-spot TTA scenario. Efficiency is measured by processing 50,000 images under a batch size of 64 via an A100 GPU.
>
> ||R50-GN|||ViT-B|||
> |-|:-:|:-:|:-:|:-:|:-:|:-:|
> |Method|Acc. (%)|Mem. (MB)|Time (s)|Acc. (%)|Mem. (MB)|Time (s)|
> |Tent|13.4|5,519|89|30.7|7,760|113|
> |TTE|40.8|6,303|136|55.2|8,574|164|
> |REM|-|-|-|56.9|27,438|346|
> |SPA|45.0|21,327|220|60.3|28,147|275|
> |ZeroSiam (Ours)|46.2|5,584|89|64.5|7,806|113|

---

> ### Author Response · Authors · 2025-11-23
> **Responses to reviewer 32Pf [3/4]**
>
> > W3. Some typos should be revised.
>
> A. Thank you for pointing out the issue. We have carefully reviewed and revised our manuscript accordingly.
>
> > Q1. In Figure 4, the sensitivity analysis of the learning rate is conducted using ViT-Base, where collapse does not occur seriously, making it difficult to fairly assess the robustness of learning rates across different approaches.
>
> A. Thank you for your expertise and constructive comments. To demonstrate ZeroSiam's robustness, **we further supplement the sensitivity analysis of learning rates on Swin-Tiny, a challenging architecture on which Tent frequently suffers from collapse, showing that ZeroSiam outperforms prior methods across broad learning rate settings.**
>
> From Table III, ZeroSiam consistently outperforms Tent across a wide range of encoder learning rates, confirming the stability trends observed on ViT-Base across architectures. Moreover, on smaller and more collapse-prone models, ZeroSiam can benefit from a larger predictor learning rate $\eta_h$, _e.g._, yielding an accuracy of 53.3% with $\eta_f=10\times5e-5$ and $\eta_h=40\times\eta_f$; while still outperforming previous methods across broad learning rate configurations, _e.g._, achieving an accuracy above 45% when $\eta_f\in[5,20]\times5e-5$ and $\eta_h \in [1,40]\times\eta_f$, showing a substantial gain compared to the accuracy of 38.2% in COME and 37.7% in DeYO. These results collectively demonstrate the efficacy of ZeroSiam across learning rate choices. In practice, we did not extensively tune the learning rates, where we simply use $\eta_f=10\times5e-5$ and $\eta_h=5\times\eta_f$ on ZeroSiam that yields an accuracy of 50.4% for comparisons with other methods.
>
>
> Table III. Sensitivity of learning rates on Swin-Tiny under label shifts.
>
> |encoder LR $\eta_f$ / predictor LR $\eta_h$|$\eta_h=0\times\eta_f$|$\eta_h=1\times\eta_f$|$\eta_h=2\times\eta_f$|$\eta_h=5\times\eta_f$|$\eta_h=10\times\eta_f$|$\eta_h=20\times\eta_f$|$\eta_h=40\times\eta_f$|
> |-:|:-:|:-:|:-:|:-:|:-:|:-:|:-:|
> |$\eta_f=1\times5e-5$|24.0|37.2|38.5|39.7|40.3|40.6|40.7|
> |$\eta_f=5\times5e-5$|14.4|45.9|46.9|47.9|48.6|49.4|50.5|
> |$\eta_f=10\times5e-5$|12.4|48.6|49.5|50.4|51.3|52.5|53.3|
> |$\eta_f=20\times5e-5$|8.6|49.7|50.3|50.9|51.2|51.1|45.3|
> |$\eta_f=40\times5e-5$|6.3|41.7|39.7|41.3|44.1|37.8|26.2|

---

> ### Author Response · Authors · 2025-11-23
> **Responses to reviewer 32Pf [4/4]**
>
> > Q2. Moreover, different learning rates are applied to the predictor depending on the architecture, and the hyperparameters of existing methods are not fully optimized, which complicates fair comparison. If the predictor’s learning rate is treated as a tunable hyperparameter, wouldn’t it also be possible to alleviate the collapse problem in other entropy minimization methods by appropriately adjusting parameters such as the entropy threshold or learning rate?
>
> A. Thank you for your questions. **We ensure a fair comparison by experimenting with both models that have been carefully tuned in prior methods (ResNet50-GN and ViT-Base), where we simply adopt their optimal hyperparameters; and models that are beyond prior evaluations (_e.g._, ConvNeXt-Tiny and Swin-Tiny), where we have searched for a near-optimal learning rate for existing methods (Table I). In this rebuttal, we further show existing methods benefit limitedly from extensive threshold tuning.**
>
> On ResNet50-GN and ViT-Base, *even when using optimal parameters*, prior methods still suffer from model collapse in the blind-spot adaptation scenario, whereas they also demonstrate inferior performance compared to ZeroSiam under the conventional Wild test settings (such as label shifts and batch size=1), indicating that the benefits of our ZeroSiam arise fundamentally from its algorithmic design rather than hyperparameter tuning.
>
> In this rebuttal, we further support the claim with carefully optimized results of DeYO on ConvNeXt-Tiny and Swin-Tiny, searching across entropy thresholds $e_0 \in \\{0.2, 0.4, 0.6\\}*\ln C$, and PLPD thresholds $\tau_{p} \in \\{0.1,0.2,0.3,0.4,0.5,0.6,0.7,0.8,0.9\\}$. As shown in Table IV, DeYO benefits limitedly from extensive hyperparameter tuning, and the performance can converge to NoAdapt that bypasses collapse by performing no adaptation, _e.g._, 31.3% (NoAdapt) _vs._ 31.4% (DeYO with $\tau_p$=0.9) under blind-spot TTA. In this case, ZeroSiam still largely outperforms the carefully optimized DeYO across scenarios, improving the accuracy from 41.4% (DeYO with $\tau_p$=0.5) to 50.4% under label shift, and from 31.4% (DeYO with $\tau_p$=0.9) to 50.8% under blind-spot adaptation, which underscores the efficacy of ZeroSiam's asymmetric design far outweighs the gains achievable via hyperparameter tuning.
>
> Table IV. Hyperparameter tunings of entropy threshold $e_0$ and PLPD threshold $\tau_{p}$ for DeYO.
>
> |Model|Scenario|NoAdapt|ZeroSiam|$e_0=0.2$|$e_0=0.4$|$e_0=0.6$|$\tau_p=0.1$|$\tau_p=0.2$|$\tau_p=0.3$|$\tau_p=0.4$|$\tau_p=0.5$|$\tau_p=0.6$|$\tau_p=0.7$|$\tau_p=0.8$|$\tau_p=0.9$|
> |-|-|:-:|:-:|:-:|:-:|:-:|:-:|:-:|:-:|:-:|:-:|:-:|:-:|:-:|:-:|
> |ConvNeXt_Tiny|label_shift|34.7|50.3|34.2|33.9|32.0|32.7|33.9|34.9|36.5|37.7|37.8|38.1|38.3|38.0|
> ||blind_spot|34.7|50.6|31.8|28.4|17.8|26.8|28.4|29.6|30.6|31.5|32.9|34.8|35.6|34.7|
> |Swin_Tiny|label_shift|31.3|50.4|35.4|37.7|30.1|35.9|37.7|39.2|40.8|41.4|41.1|41.6|36.9|35.9|
> ||blind_spot|31.3|50.8|18.3|15.6|13.0|14.3|15.6|16.4|19.3|22.2|27.0|29.3|31.1|31.4|
>
> Moreover, we further briefly introduce *a data-free method* to help select $\eta_h$. The core idea is to leverage random Gaussian noise as the proxy to track TTA dynamics, where ideally, noisy learning signals should be totally absorbed and suppressed in ZeroSiam. To this end, we track the changes in prediction, e.g., prediction entropy $\Delta_e$, before and after adaptation on Gaussian noise, and select the best-performing $\eta_h$ that leads to a minimal change, e.g., $\min_{\eta_h} |\Delta_e|$, which implies an appropriate regularization strength for bias filtration. As shown in Table V, on ResNet50-GN, we can find $\eta_h=10$ by $\min_{\eta_h} |\Delta_e|$ using only generated Gaussian before TTA deployment, which also delivers the highest TTA accuracy of 51.4% under label shift.
>
> We acknowledge that some architectures, such as ConvNeXt, can consistently output the highest prediction entropy on Gaussian noise and may require tracking other prediction metrics for data-free $\eta_h$ selection. We leave this for future work, and believe that our current focus on collapse prevention and asymmetric entropy-based architecture already makes meaningful contributions, as this significantly reduces the risk of TTA across adaptation scenarios while maintaining efficiency for practical and reliable deployment.
>
> Table V. Illustration of data-free selection for $\eta_h$ on ResNet50-GN based on $|\Delta_e|$. TTA accuracy is measured under the label shift setting. Encoder learning rate $\eta_f$ is set to 0.00125, and $\eta_h$ is set to $k\times$ 0.00125.
>
> |Metric|$\eta_h=0$|$\eta_h=2$|$\eta_h=5$|$\eta_h=10$|$\eta_h=20$|$\eta_h=40$|
> |:-:|:-:|:-:|:-:|:-:|:-:|:-:|
> |$\|\Delta_e\|$|0.750|0.085|0.053|0.001|0.061|0.083|
> |TTA Acc. (%)|14.3|49.6|51.0|51.4|50.3|45.3|
>
> We have revised Section 4.4 and included three additional sections (Appendix E.5, E.6, E.11) to reflect the rebuttal in our paper.

---

> ### Author Response · Authors · 2025-11-28
> **Thanks and a kind reminder for discussion**
>
> Dear Reviewer 32Pf,
>
> We would like to thank you again for your valuable time and effort in reviewing our submission. Based on your previous feedback, we have provided point-to-point responses and revised the manuscript accordingly. We are wondering if our response has addressed your questions appropriately and can improve your opinion of our work.
>
> As the discussion period is coming to its end, please kindly let us know if you might have further comments, and we would be glad to address them.
>
> Best regards,
>
> The Authors

---

### Official Review · Reviewer_yJuL · 2025-10-31

**Soundness:** 2
**Presentation:** 3
**Contribution:** 2
**Rating:** 2
**Confidence:** 4

**Summary:**

This paper addresses the problem of "model collapse" in test-time entropy minimization, where models adapting during inference can degenerate to predicting a single dominant class. The authors propose ZeroSiam, a lightweight, asymmetric Siamese architecture inspired by self-supervised learning methods like SimSiam. The key idea is to process a feature from a single encoder pass through two branches: an "online" branch, which includes a learnable predictor before the classifier, and a "target" branch, which uses a stop-gradient on the original classifier output. The model is then trained to minimize the entropy of the online branch while simultaneously aligning its output distribution to the stop-gradient target branch. The authors claim this approach efficiently prevents collapse without requiring augmentations or extra backbone passes5. This claim is supported by theoretical analysis and extensive empirical results showing stable performance on vision-based test-time adaptation (TTA) tasks.

**Strengths:**

S1. The primary strength of this paper is the asymmetric, predictor-stop-gradient architecture (inspired by negative-free SSL) to the problem of test-time entropy minimization. While the components (predictor, stop-gradient) are not new, adapting this mechanism to stabilize a single-branch, entropy-driven objective is an architectural contribution.

S2. The authors demonstrate the method's effectiveness beyond standard vision TTA by applying it to incentivize reasoning in large language models.

S3. The organization and the writing of the paper are good. The motivation of the proposed method is natural. The paper is easy for reader to follow.

**Weaknesses:**

W1. The most significant weakness of this paper is the misrepresentation of the primary baseline, TENT. The original TENT paper and its open-source implementation are designed for online adaptation over a limited number of epochs (e.g., 1-5). It is well-understood that adapting indefinitely on an unlabeled, out-of-distribution data stream is inherently unstable. This paper, however, appears to run TENT for an excessive number of steps to induce the very collapse it aims to solve. This creates a "strawman" argument. The plots showing TENT's accuracy dropping to near-zero are a consequence of this unfair experimental setup, not necessarily an inherent flaw of the TENT method under its intended operating conditions. This experimental choice undermines the validity of the central empirical claims.


W2. The paper's motivation states that "minimizing entropy alone does not ensure that the corresponding supervised loss... is also reduced". This is a trivial observation. The lack of ground-truth labels is the fundamental challenge of all unsupervised test-time adaptation. This framing slightly overstates the conceptual contribution by presenting a foundational premise of the subfield as a problem that this work is uniquely identifying.

W3. The alignment weight $\alpha$ is a critical hyperparameter that balances the original entropy objective with the new alignment regularizer. The paper fixes this to 1, claiming it works well without tuning24. While Table F provides a sensitivity analysis, this trade-off seems under-explored. It is likely that the optimal balance between exploration (entropy minimization) and stability (alignment) depends on the severity of the domain shift, which is hard to know beforehand.

**Questions:**

Q1. Could the authors provide a direct comparison (e.g., for Table 3 or 4) where both TENT and ZeroSiam are run for only a small, fixed number of epochs (e.g., 1, 3, and 5) over the entire test set?

Q2.  Does the alignment loss $D(p^o || sg[p^r])$ also grow indefinitely, eventually overpowering the entropy term $H(p^o)$? A plot of the individual loss components over time would be very insightful.

---

> ### Author Response · Authors · 2025-11-23
> **Responses to reviewer yJuL [1/3]**
>
> Thank you for taking the time to review our paper and for providing valuable feedback. We would like to address your questions below.
>
> > W1&Q1. The most significant weakness of this paper is the misrepresentation of the primary baseline, TENT. The original TENT paper and its open-source implementation are designed for online adaptation over a limited number of epochs (e.g., 1-5). It is well-understood that adapting indefinitely on an unlabeled, out-of-distribution data stream is inherently unstable. This paper, however, appears to run TENT for an excessive number of steps to induce the very collapse it aims to solve. This creates a "strawman" argument. The plots showing TENT's accuracy dropping to near-zero are a consequence of this unfair experimental setup, not necessarily an inherent flaw of the TENT method under its intended operating conditions. This experimental choice undermines the validity of the central empirical claims. Could the authors provide a direct comparison (e.g., for Table 3 or 4) where both TENT and ZeroSiam are run for only a small, fixed number of epochs (e.g., 1, 3, and 5) over the entire test set?
>
> A. Thank you for the thoughtful comments and for raising this concern. We would like to respectfully clarify that **our experiments do not run Tent for excessive steps**, and all results in Tables 2–4 \& 7 **are fair comparisons that strictly follow the widely adopted evaluation settings** established by prior works such as Tent, SAR, and DeYO. In these experiments, all methods (including Tent and ZeroSiam) are **evaluated with only 1 epoch over the test set—each sample is processed exactly once**. The x-axis in Figure 2  corresponds to the batch index in the online stream, not multiple epochs.
>
> (1) **Results in Tables 2–4: Wild setting (SAR, DeYO protocol).**
> The Wild protocol—characterized by small batch sizes, online imbalanced label-shifted streams, and mixture of domain shifts—is widely adopted for TTA evaluation because it more closely reflects real-world deployment scenarios. Under this challenging setting, Tent is known to exhibit instability even with a single pass (i.e., one epoch). This behavior has been consistently reported in previous works (including SAR and DeYO) and stems from Tent’s entropy-minimization objective, which can overfit the current micro-batch and drift toward biased predictions. In contrast, ZeroSiam achieves consistent improvements across model architectures and datasets under the same conditions, demonstrating stronger reliability for practical test-time model adaptation.
>
> (2) **Results in Table 7: Tent’s default mild setting.**
> Under the original Tent setup (uniform label space, larger batch size (64), one epoch), Tent is indeed more stable, improving from 29.9% (NoAdapt) to 47.9% on ViT-Base with ImageNet-C. However, as shown in Fig. B, Tent still exploits shortcut behaviors such as logit-norm inflation, limiting its generalization. In contrast, ZeroSiam mitigates these biased signals via the predictor-induced asymmetry, improving accuracy to 62.6% even when Tent does not collapse—highlighting that ZeroSiam’s benefit is not tied to collapse.
>
> (3) **Additional results for 1/3/5 epochs (requested).**
> Following your suggestion, we further evaluate Tent and ZeroSiam using 1, 3, and 5 epochs under Tent’s default mild setting. Tent’s performance gradually degrades with more epochs (47.9% → 41.3%), reflecting its known overfitting tendency. In contrast, ZeroSiam remains stable and even benefits moderately from additional epochs (62.6% → 65.7%). This confirms that ZeroSiam improves generalization while mitigating overfitting in Tent, which maintains a stable adaptation across different epoch budgets.
>
> Table I. Accuracy of Tent and ZeroSiam under the $k$-th epoch of TTA over the entire test set on ViT-Base under Tent's default mild setting.
>
> | Method | NoAdapt | Tent (k=1) | Tent (k=3) | Tent (k=5) | ZeroSiam (k=1) | ZeroSiam (k=3) | ZeroSiam (k=5) |
> | ------ | :-----: | :--------: | :--------: | :--------: | :------------: | :------------: | :------------: |
> | Acc.(%)  | 29.9 | 47.9 | 45.5 | 41.3 | 62.6 | 65.7 | 65.7 |
>
> In summary, our evaluation does not rely on running Tent for excessive steps, and all baselines follow the standard one-epoch protocol. The additional results further demonstrate that ZeroSiam improves Tent not only under instability but also under relative stable regimes (i.e., mild setting).

---

> ### Author Response · Authors · 2025-11-23
> **Responses to reviewer yJuL [2/3]**
>
> > W2. The paper's motivation states that "minimizing entropy alone does not ensure that the corresponding supervised loss... is also reduced". This is a trivial observation. The lack of ground-truth labels is the fundamental challenge of all unsupervised test-time adaptation. This framing slightly overstates the conceptual contribution by presenting a foundational premise of the subfield as a problem that this work is uniquely identifying.
>
> A. Thank you for the insightful comments. We fully agree that the absence of ground-truth labels is the fundamental challenge of unsupervised TTA, and **we do not intend to position ZeroSiam as uniquely identifying this difficulty**. As noted in the fourth paragraph of the Introduction, **we explicitly acknowledge extensive prior efforts—including DeYO, SAR and TLM**—that aim to mitigate instability in entropy-based adaptation: "*Various TTA methods have been proposed to tackle the above issue* ...", and have provided detailed discussions in the related work section.
>
> Our motivation is not to restate this foundational premise, but to highlight a specific gap that persists in existing approaches: many of their objectives can still be trivially minimized by constant one-hot predictions, which leaves them vulnerable to collapse in more realistic or challenging test-time conditions. This is empirically reflected in Tables 2–6, where even strong methods such as DeYO occasionally degrade to extremely low accuracy (e.g., 0.2%) under more difficult scenarios or weaker base models.
>
> To bridge this gap, ZeroSiam introduces an asymmetric two-branch design that inherently avoids these trivial solutions. Unlike prior entropy-based approaches, the introduced asymmetry filters out degenerate shortcut signals and stabilizes the adaptation dynamics. This design not only mitigates collapse, but also leads to improved generalization even in regimes where collapse does not occur (Tables 2–7). Our contribution lies in this new perspective and mechanism, rather than in identifying the underlying challenge.
>
>
> > W3. The alignment weight $\alpha$ is a critical hyperparameter that balances the original entropy objective with the new alignment regularizer. The paper fixes this to 1, claiming it works well without tuning. While Table F provides a sensitivity analysis, this trade-off seems under-explored. It is likely that the optimal balance between exploration (entropy minimization) and stability (alignment) depends on the severity of the domain shift, which is hard to know beforehand.
>
>
> A. Thank you for your thoughtful comments. In ZeroSiam, **we aim to find a reasonable $\alpha$ that is robust and alleviates hyperparameter tuning for practical TTA deployment**. The rationale is that it is often infeasible to pursue a single optimal hyperparameter, including the learning rate, that consistently works the best for all unpredictable test scenarios.
>
> While setting $\alpha=1.0$ may not be optimal for all cases, **it provides a practical tuning-free and interpretable default for two reasons**: 1) **comparable gradient scales**: both entropy loss and the divergence loss operate in _probability space_ and are also _per-sample normalized_, which yield comparable gradient scales for TTA, without any one dominanting optimization; 2) **simplified and interpretable formulations**: consider Eqn. (4) with $\alpha=1.0$ detailed as $H(p^{o})+D_{KL}(p^{o}||\mathrm{sg}[p^r])+D_{KL}(\mathrm{sg}[p^r]||p^{o})$. By expanding the terms, we derive an overall formulation of $-p^{o}\log \mathrm{sg}[p^r] - \mathrm{sg}[p^r]\log p^{o}$, which simplifies into a _symmetric cross-entropy_ between online and target predictions that provides a clear probabilistic interpretation of the optimization behaviors.
>
> In this rebuttal, we provide further sensitivity analysis of $\alpha$ under different domain shift severities. As shown in Table II, by aligning the gradient scale of both terms, ZeroSiam achieves consistently the best results with $\alpha=1$ across severities of domain shift, outperforming Tent's accuracy by 2.1% (at level 1) and 4.6% (at level 3). Extensive results across various test scenarios (Tables 2-6 in the main paper) also validate the efficacy of this setup, making it a reasonable default to avoid careful hyperparameter tuning in TTA.
>
> Table II. Sensitivity analysis of $\alpha$ under different severity levels of domain shift.
>
> |            | NoAdapt | Tent | $\alpha=0.5$ | $\alpha=1$ | $\alpha=1.5$ | $\alpha=2$ | $\alpha=2.5$ |
> | ---------- | :-----: | :--: | :----------: | :----------: | :----------: | :----------: | :----------: |
> | Severity_Level=1 |   67.9    | 76.8 |     78.8     |     78.9     |     78.6     |     78.0     |     77.0     |
> | Severity_Level=3 |   53.8      | 69.5 |     73.9     |     74.1     |     73.8     |     73.2     |     71.8     |
> | Severity_Level=5 | 29.9 | 47.3 |     63.9     | 64.1 | 63.9 | 63.3 | 62.4 |

---

> ### Author Response · Authors · 2025-11-23
> **Responses to reviewer yJuL [3/3]**
>
> > Q2. Does the alignment loss $D(p^o||\mathrm{sg}[p^{r}])$ also grow infinitely, eventually overpowering the entropy term $H(p^{o})$? A plot of the individual loss components over time would be very insightful.
>
> A. Thank you for your insightful questions. **The alignment loss $D(p^o||\mathrm{sg}[p^{r}])$ does not grow infinitely and maintains a reasonable range.** We have supplemented a figure (Figure E) in the revised paper, showing that the alignment loss quickly increases then converges, while the entropy term $H(p^{o})$ decreases and stabilizes. Throughout adaptation, the alignment term exhibits a smaller loss value than the entropy term, which provides an appropriate regularization strength that prevents the risk of collapse, without hindering the effectiveness of the entropy-based adaptation.
>
> We have revised Section 4.4 and included three additional sections (Appendix C.3, E.4, E.7) to reflect the rebuttal in our paper. We sincerely hope our clarifications above have addressed your concerns and can help you re-assess our work.

---

> ### Author Response · Authors · 2025-11-28
> **Thanks and a kind reminder for discussion**
>
> Dear Reviewer yJuL,
>
> We would like to thank you again for your valuable time and effort in reviewing our submission. Based on your previous feedback, we have provided point-to-point responses and revised the manuscript accordingly. We are wondering if our response has addressed your questions appropriately and can improve your opinion of our work.
>
> As the discussion period is coming to its end, please kindly let us know if you might have further comments, and we would be glad to address them.
>
> Best regards,
>
> The Authors

---

### Official Review · Reviewer_BMnf · 2025-11-04

**Soundness:** 2
**Presentation:** 3
**Contribution:** 2
**Rating:** 4
**Confidence:** 4

**Summary:**

ZeroSiam introduces a lightweight, low‑cost asymmetric architecture for test‑time adaptation by inserting a single linear predictor before the classifier and pairing it with a stop‑gradient target branch, enabling adaptation with only one encoder pass. Its test‑time loss function minimizes entropy on the online branch while aligning it to the stop‑grad branch via a symmetric‑KL term, which discourages shortcut solutions such as logit‑norm inflation and single‑class collapse. Extensive experiments across diverse backbones (ResNet, ViT, ConvNeXt, Swin) show consistent gains under mixed corruptions, label shifts, and batch‑size‑1 streams. This paper also shows improvement in LLM reasoning experimentally.

**Strengths:**

- This paper delivers stable improvements over strong TTA baselines under mixed corruptions, online label shifts, batch‑size‑1, and even blind‑spot adaptation. Furthermore, it includes LLM mathematical reasoning underscoring concrete experimental cases.

 - The network modification is minimal, instead it requiresa single linear predictor plus a stop‑grad branch. In contrast, it does not depend on augmentations or extra encoder passes. Thus, Its runtime equals to baselines.

**Weaknesses:**

- This paper lacks a comprehensive understanding of representation adaptation to test-time scalings and the preservation of pre-training knowledge. The proposed loss function, which minimizes the difference from pre-trained knowledge, is driven by KL Divergence. While the paper asserts that $\alpha = 1.0$ is applicable across all experimental cases, I believe that a deeper understanding and explanation of representations are crucial for forming more concrete and reasonable evidence, rather than elaborating numeric numbers across experimental cases.

- I believe that the theoretical analysis presented in this paper doesn’t align with the empirical contribution. The analysis focuses on entropy minimization and KL Divergence separately. Specifically, it fails to accomodate which dynamic occurs during the training process when two terms are considered at the same time. When applied to test-time adaptation experiments, these concepts serve as loss functions simultaneously. Consequently, I believe that this analysis hinders the accurate understanding of the proposed loss function.

**Questions:**

- I was wondering if ZeroSiam achieves consistant performance when it negates predictor $h$ and employ only classifier $g$. It would be more favorable to highlight the proposed loss function without any modification of networks.

---

> ### Author Response · Authors · 2025-11-23
> **Responses to reviewer BMnf [1/2]**
>
> Thank you for taking the time to review our paper and for providing valuable feedback. We would like to address your questions below.
>
> > W1-1. This paper lacks a comprehensive understanding of representation adaptation to test-time scalings and the preservation of pre-training knowledge. The proposed loss function, which minimizes the difference from pre-trained knowledge, is driven by the KL Divergence.
>
>
> A. Thank you for your question. We would like to respectfully clarify that ZeroSiam does not rely on any pre-trained knowledge, nor does it minimize the KL divergence to pre-trained knowledge. **ZeroSiam contains only one single target model, where we only introduce a lightweight predictor (a simple fully connected layer) to form two branches** (as in Figure 1 c): (1) the online branch, consisting of the **shared encoder**, predictor, and **shared classifier**; and (2) the target branch, consisting of the **shared encoder** and **shared classifier**. In this way, the two branches produce different outputs and thus we minimize the KL divergence between them for alignment.
>
> The motivation for this design comes from Figure 2(b–c): entropy-based objectives often exploit degenerate shortcuts—such as logit-norm inflation or single-class dominance—to reduce entropy, leading to collapsed representations. In ZeroSiam, the additional predictor in the online branch purposely absorbs these biased shortcut signals and converts them into explicit discrepancies, which are then penalized by the alignment loss. As shown in Theorem 1, any shortcut representable by the predictor is naturally suppressed, while useful signals flow into the backbone encoder update.
>
> This asymmetric, predictor-based bias filtration is a minimal architectural change that yields both theoretical guarantees and practical robustness. ZeroSiam maintains Tent-level efficiency, avoids collapse even under adversarial or non-semantic inputs (Figure 3), and provides stable adaptation ability under various real-world challenging test-time adaptation settings without relying on pre-training knowledge preservation.
>
>
> > W1-2. While the paper asserts that $\alpha=1.0$ is applicable across all experimental cases, I believe that a deeper understanding and explanation of representations are crucial for forming more concrete and reasonable evidence, rather than elaborating numeric numbers across experimental cases.
>
>
> A. Thank you for your thoughtful comments. Our intention in setting $\alpha = 1.0$ is not to claim that it is *universally optimal* according to the empirical results, but to provide a robust and interpretable default that **avoids heavy tuning in practical TTA use cases**.
>
> From the learning objective, $\alpha = 1.0$ strikes a balanced and interpretable interaction between the two objectives:
>
> - **comparable gradient scales**: both entropy loss and the divergence loss operate in _probability space_ and are also _per-sample normalized_, which yield comparable gradient scales for TTA, without any one dominating optimization;
> - **simplified and interpretable formulations**: consider Eqn. (4) with $\alpha=1.0$ detailed as $H(p^{o})+D_{KL}(p^{o}||\text{sg}[p^r])+D_{KL}(\text{sg}[p^r]||p^{o})$. By expanding the terms, we derive an overall formulation of $-p^{o}\log \text{sg}[p^r] - \text{sg}[p^r]\log p^{o}$, which expands to a symmetric cross-entropy between online and target branches. This symmetry encourages the two branches to exchange reliable signals while preventing either from drifting too far—an effect we find important for stable adaptation.
>
> Finally, extensive experiments across datasets, shifts, and ablation settings (Tables 2–6 and Table 11 in the revised paper) show that this default consistently yields strong adaptation without tuning, further indicating its practicality and empirically supporting our choice.

---

> ### Author Response · Authors · 2025-11-23
> **Responses to reviewer BMnf [2/2]**
>
> > W2. I believe that the theoretical analysis presented in this paper doesn’t align with the empirical contribution. The analysis focuses on entropy minimization and KL Divergence separately. Specifically, it fails to accommodate which dynamic occurs during the training process when two terms are considered at the same time. When applied to test-time adaptation experiments, these concepts serve as loss functions simultaneously. Consequently, I believe that this analysis hinders the accurate understanding of the proposed loss function.
>
> A. Thank you for your comments. Our theoretical analysis is designed to faithfully reflect the mechanisms of our ZeroSiam and support its contributions. Specifically, **when both entropy and divergence are considered at the same time (_e.g._, $\alpha=1.0$), this corresponds to our theoretical analysis for $\alpha>0$ in Theorem 1**, where we show that ZeroSiam converges toward a stable, and non-collapsing equilibrium.
>
> Additionally, we include a theoretical analysis for $\alpha=0$ in Theorem 1, which explains how the discrepancy between the online and target branches is created and why it is meaningful to stabilize adaptation. We theoretically demonstrate that the online branch converges more rapidly to the collapse region, and the target branch thus provides an anti-collapse regularization when optimizing entropy.
>
> Together, the analyses for both $\alpha=0$ and $\alpha>0$ give complementary insights into how ZeroSiam introduces meaningful discrepancies for regularization and bias filtration, while also demonstrating its convergence toward a stable, and non-collapsing equilibrium, which aligns with our empirical findings of ZeroSiam's stability across challenging adaptation scenarios.
>
> > Q1. I was wondering if ZeroSiam achieves consistent performance when it negates predictor $h$ and employs only classifier $g$. It would be more favorable to highlight the proposed loss function without any modification of networks.
>
> A. Thank you for the thoughtful question. As clarified in our response to W1-1, **the predictor $h$ is the key—and the only—component that introduces asymmetry**, ensuring that the two branches produce different outputs despite sharing the same encoder and classifier. This asymmetry is crucial for generating a meaningful alignment loss. **Without the predictor $h$**, both branches in ZeroSiam become identical, yielding a trivial zero alignment loss. In this case, **ZeroSiam actually reduces to the original Tent method**, losing both the asymmetric structure and the bias-filtration effect that are essential for stabilizing test-time adaptation.
>
> We have revised Section 4.4 and Appendix B to reflect the rebuttal in our paper. We sincerely hope our clarifications above have addressed your concerns and can improve your opinion of our work.

---

> ### Author Response · Authors · 2025-11-28
> **Thanks and a kind reminder for discussion**
>
> Dear Reviewer BMnf,
>
> We would like to thank you again for your valuable time and effort in reviewing our submission. Based on your previous feedback, we have provided point-to-point responses and revised the manuscript accordingly. We are wondering if our response has addressed your questions appropriately and can improve your opinion of our work.
>
> As the discussion period is coming to its end, please kindly let us know if you might have further comments, and we would be glad to address them.
>
> Best regards,
>
> The Authors

---

### Author Response · Authors · 2025-12-02
**Summary of Rebuttal and Clarifications**

Dear Reviewers, ACs, SACs, and PCs,

We sincerely appreciate the valuable feedback provided by all reviewers and the additional efforts made by the ACs, SACs, and PCs under these special circumstances. As no further discussion is permitted, we would like to provide a summary of our rebuttal.

**Reviewer 32Pf (score 8)** recognizes the **efficiency and practicality** of the proposed method for real-world applications, and appreciates our extensive evaluations across **a wide range of benchmarks** and the **visualization of the stabilizing effects** (Strengths 1-3). Reviewer 32Pf's questions focus on the fairness of hyperparameter choices, mainly about the learning rate (W1, Q1 & Q2). We clarify that we ensure fair comparisons with baselines by adopting their optimal hyperparameters from the original papers on ResNet50-GN and ViT-Base, and have already searched for the near-optimal hyperparameters on newly added models like ConvNeXt-Tiny and Swin-Tiny (W1 & Q2). Moreover, we show that our method is robust and consistently surpasses prior methods across broad learning rate configurations on various models (Q1).

**Reviewer UrZX (score 6)** recognizes the proposed method is **simple and effective**, and appreciates our **theoretical supports of convergence**, **extensive ablations**, and **diverse results covering vision and large language models** (Strengths 1-4). Reviewer UrZX's questions mainly seek further explanation on concepts and theory, such as the imbalanced ratio (W1), logit norm (W2), center dominance (W3), and the Lipschitz continuity assumption (Q1). We provide more clarifications and references on these concepts in our rebuttal and revised our paper accordingly. Reviewer UrZX has provided further feedback that the questions are addressed.

**Reviewer yJuL and BMnf** also give positive feedback on our work, including "*an architectural contribution*", "*does not depend on augmentations or extra encoder passes*", "*concrete experimental cases*", and "*delivers stable improvements*".

We greatly appreciate Reviewer yJuL and BMnf for their valuable and constructive feedback, but meanwhile we also would like to respectfully point out that the some main concerns raised by Reviewer yJuL (score 2) and BMnf (score 4) might **stem from biased understandings of the proposed method/TTA protocols**, which may have adversely influenced their assessments. We respectfully re-clarified this and revised the manuscript for a better understanding. Specifically,

**Reviewer yJuL (score 2)** raised concerns that we had adopted an incorrect evaluation protocol for Tent that trains for an **excessive epochs** over the test set to induce the collapse issue (W1 & Q1). In the rebuttal, we respectfully clarified that we **strictly followed the Mild and Wild evaluation settings established in Tent and SAR**, where methods such as Tent and ZeroSiam are evaluated with **only 1 epoch** over the test set. Reviewer yJuL also concerned that we position ourselves as **uniquely identifying** the difficulties in unsupervised adaptation (W2). However, we find this might stem from that the reviewer **missed our statements in our original paper**. Thus, in the rebuttal, we respectfully clarified that we explicitly **recognized extensive prior efforts**—including SAR, DeYO, TLM—on identifying this difficulty in the introduction, related work, and experiment section.

**Reviewer BMnf (score 4)** mainly concerned that the proposed method directly **minimizes the KL divergence to pre-trained knowledge** (W1 & Q1). With extreme respect, in the response, we clarified that our ZeroSiam **does not rely on any pre-trained knowledge**, nor does it minimize the KL divergence to pre-trained knowledge, and **contains only a single target model** to maintain high computational efficiency. Reviewer BMnf also concerned that we lack a theoretical analysis when using entropy and divergence simultaneously (W2). In the rebuttal, we respectfully clarified that this circumstance (_i.e._, $\alpha=1$) is included in our theoretical analysis for $\alpha>0$ in Theorem 1.

Overall, our paper aims to fundamentally **address the collapse issue** in test-time entropy optimization **without using heuristic thresholds, data augmentations, and additional forward passes**, to advance the practicality and reliability of TTA in the real world. Our main novelty lies in our **efficient asymmetric architecture** design tailored for self-training, which exploits **a single lightweight predictor** to introduce both the mechanisms of **bias signals filtration** and **asymmetric optimization** for preventing collapse and enhancing generalization. Extensive theoretical analysis, visualizations, ablations, and empirical results on both vision and large language models validate the effectiveness of the proposed method.

Once again, we extend our sincere gratitude to all reviewers, ACs, SACs, and PCs for their significant efforts and contributions.

Best regards,

The Authors

---

### Meta-Review · Area_Chair_Z8AV · 2026-01-06

**Summary:**

1. The majority of the reviewers (i.e. Reviewers 32Pf, UrZX, and BMnf) find the proposed algorithms effective and practical. In addition, they acknowledge the consistent performance improvements in extensive experiment settings across a wide range of benchmarks, including large language models and vision tasks.

2. Reviewer UrZX  also praises the theoretical guarantees on convergence behaviors.

3. The major concern raised by Reviewer yJuL, who gave the lowest score 2, is the incorrect evaluation protocol for Tent adopted by this paper. However, this seems to be a misunderstanding: the authors carefully clarified in the rebuttal that they followed the same evaluation protocol as done in the literature.

4. Reviewer BMnf (score 4) is also concerned about the lack of theoretical results when using entropy and divergence simultaneously, which also seems to be a misunderstanding: as pointed out by the authors' rebuttal, the theoretical results contain this case as a special case when alpha =1.

**Reviewer Concerns:**

The following concerns have been addressed by the rebuttal:

1. Reviewer yJuL's concern 1: "The most significant weakness of this paper is the misrepresentation of the primary baseline, TENT. "

The rebuttal has clarified that they implemented the baseline TENT in the same way as done in the literature and didn't run it for extra steps.

2. Reviewer's  yJuL's concern 2: "The lack of ground-truth labels is the fundamental challenge of all unsupervised test-time adaptation. This framing slightly overstates the conceptual contribution by presenting a foundational premise of the subfield as a problem that this work is uniquely identifying."

The rebuttal clarified that the goal of this paper is to fill in the gap in the existing methods to address the unsupervised test-time adaptation problems instead of proposing a conceptually new subfield to address this issue.

3. Reviewer yJuL's concern 3 and Reviewer BMnf's concern 1: "The alignment weight is a critical hyperparameter that balances the original entropy objective with the new alignment regularizer. The paper fixes this to 1, claiming it works well without tuning."

The rebuttal demonstrates that alpha = 1 is a simple and robust choice, consistently providing good empirical performance, even though it may not be optimal in all cases.

4. Reviewer BMnf's concern 2: "I believe that the theoretical analysis presented in this paper doesn’t align with the empirical contribution. The analysis focuses on entropy minimization and KL Divergence separately. Specifically, it fails to accommodate which dynamic occurs during the training process when two terms are considered at the same time."

The rebuttal clarifies that their theoretical results hold for alpha >0, and the case questioned by the reviewer is a special case when alpha =1, so it is covered by the theoretical results provided by this paper.

5. Reviewer UrZX's concerns were addressed successfully, as acknowledged by the reviewer themselves.

6. Reviewer 32Pf (score 8)'s concerns were also addressed successfully.

The following are outstanding concerns.
1. Despite the additional theoretical results, I think the choice of alpha, a common concern by two reviewers, deserves further attention and discussion.

**Reviewer Scores:**

1. Reviewers 32Pf (score 8): I don't think this reviewer will change their score after discussion.

2. Reviewer UrZX (score 6) didn't change their score after reviewing the rebuttal.

3. Reviewer BMnf (score 4): I think this reviewer will increase the score to 5 or keep it at 4.

4. Reviewer yJuL (score 2): I think this reviewer will increase the score to 4 or 5.

---

> ### Public Comment · ~Guohao_Chen1 · 2026-02-28
> **Request for Modifying the Title**
>
> Dear AC and reviewers,
>
> We kindly request to change the title of the ICLR submission from "ZeroSiam: An Efficient Siamese for Test-Time Entropy Optimization without Collapse" to "ZeroSiam: An Efficient Asymmetry for Test-Time Entropy Optimization without Collapse". We believe asymmetry better aligns with our motivations and helps outline our contributions to asymmetric architectural design for entropy minimization, without changing the scope of the paper.
>
> Best regards,
>
> The Authors

---

### Decision · Program_Chairs · 2026-01-26

Accept (Poster)